# Model of Propagation of VLF Beams in the Waveguide Earth-Ionosphere. Principles of Tensor Impedance Method in Multilayered Gyrotropic Waveguides.

Yuriy Rapoport[#±1], Vladimir Grimalsky[*2], Victor Fedun[~3], Oleksiy Agapitov [#⊥4], John Bonnell[⊥5], Asen Grytsai[#6], Gennadi Milinevsky[#ε7], Alex Liashchuk[±8], Alexander Rozhnoi[φ9], Maria Solovieva[φ10], Andrey Gulin[#11]

[#]Taras Shevchenko National University of Kyiv, Ukraine
[*]Autonomous University of State Morelos (UAEM), Mexico
[~]The University of Sheffield, UK
[⊥]University of California, Berkeley, USA
[ε]College of Physics, International Center of Future Science, Jilin University, Changchun, China
[±] National Space Facilities Control and Test Center, State Space Agency of Ukraine
[φ]Institute of the Earth Physics, RAS, Moscow, Russia

[1]yuriy.rapoport@gmail.com,      [2]v_grim@yahoo.com,      [3]v.fedun@sheffield.ac.uk,      [4]oleksiy.agapitov@gmail.com, [5]jwbonnell@berkeley.edu,      [6]a.grytsai@gmail.com,      [7]genmilinevsky@gmail.com,      [8]alex.liashchukk@gmail.com, [9]rozhnoi@ifz.ru, [10]MCSolovieva@gmail.com, [11]mytimetosay@gmail.com

*Correspondence to*: Yuriy Rapoport (yuriy.rapoport@gmail.com)_

**Abstract.** The modeling of very low frequency electromagnetic beam propagation in the earth-ionosphere waveguide is considered. A new tensor impedance method for modeling propagation of electromagnetic beams in a multi-layered/inhomogeneous waveguide is presented. The waveguide is assumed to possess the gyrotropy and inhomogeneity with a thick cover layer placed above the waveguide. The influence of geomagnetic field inclination and carrier beam frequency on the characteristics of the polarization transformation in the Earth-ionosphere waveguide is determined. The new method for modeling of propagation of electromagnetic beams allows us to study: (i) propagation of the very low frequency modes in the earth-ionosphere waveguide and, in perspective, their excitation by the typical Earth-ionosphere waveguide sources, such as radio wave transmitters and lightning discharges and (ii) leakage of Earth-ionosphere waveguide waves into the upper ionosphere/magnetosphere. The proposed approach can be applied to the variety of problems related to the analysis of propagation of electromagnetic waves in layered gyrotropic/anisotropic active media in a wide frequency range, e.g. from the Earth-ionosphere waveguide to optics waveband, an artificial signal propagation such as metamaterial microwave or optical waveguides.

*Keywords* — ionosphere, atmosphere, VLF, tensor impedance, gyrotropy, layered waveguide, beam, electromagnetic waves, boundary conditions, ionospheric disturbances, vertical coupling processes

# 1 Introduction

The results of analytical and numerical study of very low frequency (VLF) electromagnetic (EM) wave/beam propagation in the Lithosphere–Atmosphere–Ionosphere–Magnetosphere system (LAIM), in particular in the waveguide "Earth-Ionosphere" (WGEI) are presented. The amplitude and phase of the VLF wave propagates in the earth-ionosphere waveguide (WGEI) can change, and these changes may be observable using ground-based and/or satellite detectors. This reflects the

variations in the ionospheric electrodynamic characteristics (complex dielectric permittivity) and the influences on the ionosphere, for example, "from above" by chain Sun – Solar Wind – Magnetosphere – Ionosphere (Patra et al., 2011; Koskinen, 2011; Boudjada et al., 2012; Wu et al., 2016; Yiğit et al., 2016). Then the influence on the ionosphere "from below" comes from the most powerful meteorological, seismogenic and other sources in the lower atmosphere and lithosphere/Earth, such as cyclones and hurricanes (Nina et al., 2017; Rozhnoi et al., 2014; Chou et al., 2015) as well as from earthquakes (Hayakawa, 2015; Surkov and Hayakawa, 2014; Sanchez-Dulcet et al., 2015) and tsunamis. From inside the ionosphere the strong thunderstorms, lightning discharges, and terrestrial gamma-ray flashes or sprite streamers (Cummer et al., 1998; Qin et al., 2012; Dwyer, 2012; Dwyer and Uman, 2014; Cummer et al., 2014; Mezentsev et al., 2018) influence the ionospheric electrodynamic characteristics as well. Note that the VLF signals are very important for the merging of the atmospheric physics and space plasma physics with astrophysics and high-energy physics. The corresponding "intersection area" for these four disciplines includes cosmic rays and very popular now objects of investigation – high-altitude discharges (sprites), anomalous X-ray bursts, and powerful gamma-ray bursts. The key phenomena for the occurrence of all of these objects is the appearance of runaway avalanche in the presence of high energy seed electrons. In the atmosphere, there are cosmic ray secondary electrons (Gurevich and Zybin, 2001). Consequently, these phenomena are intensified during the air shower generating by cosmic particles (Gurevich and Zybin, 2001; Gurevich et al., 2009). The runaway breakdown and lightning discharges including **high-altitude** ones can **cause** radio emission both in HF range, which could be observed using the Low-Frequency Array (LOFAR) radio telescope network facility and other radio telescopes (Buitink et al., 2014; Scholten et al., 2017; Hare et al., 2018), and in the VLF range (Gurevich and Zybin, 2001). The corresponding experimental research includes the measurements of the VLF characteristics by the international measurement system of the pairs "transmitted-receiver" separated by a distance of a couple of thousand km (Biagi et al., 2011; 2015). The World Wide Lightning Location Network is the one more international facility for VLF measurements during thunderstorms with lightning discharges (Lu et al. 2019). Intensification of magnetospheric research, wave processes, particle distribution and wave-particle interaction in the magnetosphere including radiation belts leads to the great interest to the VLF plasma waves, in particular whistlers (Artemyev et al., 2013, 2015; Agapitov et al., 2014; Agapitov et al., 2018).

The differences of the proposed model for the simulation of VLF waves in the WGEI from others can be summarized in three main points. (i) In distinction to the impedance invariant imbedding model (Shalashov and Gospodchikov, 2010; Kim and Kim, 2016), our model provides an optimal balance between the analytical and numerical approaches. It combines analytical-numerical approaches basing on matrix sweep method (Samarskii, 2001). As a result, this model allows to obtain analytically the tensor impedance and, at the same time, provides high effectiveness and stability of the modeling. (ii) In distinction to the full-wave finite difference time domain models (Chevalier and Inan, 2006; Marshall et al., 2017; Yu et al., 2012; Azadifar et al., 2017), our method provides the physically clear lower and upper boundary conditions, in particular physically justified upper boundary conditions corresponding to the radiation of the waves propagation in the WGEI to the upper ionosphere/magnetosphere. This allows to determine the leakage modes and to interpret not only ground-based, but also satellite measurements of the VLF beam characteristics. (iii) In distinction to the

models considered in (Kuzichev and Shklyar, 2010; Kuzichev et al., 2018; Lehtinen and Inan, 2009; Lehtinen and Inan, 2008) based on the mode presentations and made in the frequency domain, we use the combined approach. This approach includes the radiation condition at the altitudes of the F-region, equivalent impedance conditions in the lower E-region and at the lower boundary of the WGEI, mode approach, and finally, the beam method. This combined approach, finally, creates the possibility to interpret adequately data of both ground-based and satellite detection on the VLF EM wave/beam propagating in the WGEI and those, which experienced a leakage from the WGEI into the upper ionosphere and magnetosphere. Some other details on the distinctions from the previously published models are given below in Sect. 3.

The methods of effective boundary conditions such as effective impedance conditions (Tretyakov, 2003; Senior and Volakis, 1995; Kurushin and Nefedov, 1983) are well-known and can be used, in particular, for the layered metal-dielectric, metamaterial and gyrotropic active layered and waveguiding media of different types (Tretyakov, 2003; Senior and Volakis, 1995; Kurushin and Nefedov, 1983; Collin, 2001; Wait, 1996) including plasma-like solid state (Ruibys, and Tolutis, 1983) and space plasma (Wait, 1996) media. The plasma wave processes in the waveguide structures metal-semiconductor-dielectric, placed into the external magnetic field, were widely investigated (Ruibys and Tolutis, 1983; Maier, 2007; Tarkhanyan and Uzunoglu, 2006) from radio to optical frequency ranges. Corresponding waves are applied in modern plasmonics and in non-destructive testing of semiconductor interfaces. It is interesting to realize the resonant interactions of volume and surface electromagnetic waves in these structures, so the simulations of the wave spectrum are important. To describe such complex layered structures, it is very convenient and effective to use impedance approach (Tretyakov, 2003; Senior and Volakis, 1995; Kurushin and Nefedov, 1983). As a rule, impedance boundary conditions are used, when the layer covering waveguide is thin (Senior and Volakis, 1995; Kurushin and Nefedov, 1983). One of the known exclusions is the impedance invariant imbedding model. The difference between our new method and that model is already mentioned above and is explained in more details in the Subsection 3.3. Our new approach, i.e a new tensor impedance method for modeling propagation of electromagnetic beams (TIMEB), includes a set of very attractive features for practical purposes. These features are: (i) the surface impedance characterizes cover layer of finite thickness, and this impedance is expressed analytically; (ii) the method allows an effective modelling of 3D beam propagating in the gyrotropic waveguiding structure; (iii) finally, if the considered waveguide can be modified by any external influence such as bias magnetic or electric fields, or by any extra wave or energy beams (such as acoustic or quasistatic fields etc.), the corresponding modification of the characteristics (phase and amplitude) of the VLF beam propagating in the waveguide structure can be modelled.

Our approach was targeting properly and is suitable for the further development which will allow to solve also the following problems: (i) the problem of the excitation of the waveguide by the waves incident on the considered structure from above could be solved as well with the slight modification of the presented model, with inclusion also ingoing waves; (ii) to consider a plasma-like system placed into the external magnetic field, such as the LAIM system (Grimalsky et al., 1999 a, b) or dielectric-magnetized semiconductor structure. The electromagnetic waves radiated outside the waveguiding structure, such as helicons (Ruibys and Tolutis, 1983) or whistlers (Wait, 1996), and the waveguide modes could be considered altogether. An adequate boundary radiation conditions on the upper boundary of the covering layer are derived.

Based on this and absence of ingoing waves, the leakage modes above the upper boundary of the structure (in other words,
upper boundary of covering layer), will be searched with the further development of the model delivered in the present
paper. Namely, it will be possible to investigate the process of the leakage of electromagnetic waves from the open
waveguide. Then their transformation into magnetized plasma waves, propagating along magnetic field lines, and the
following excitation of the waveguiding modes by the waves incident on the system from external space (Walker, 1976), can
be modeled as a whole. Combining with the proper measurements of the phases and amplitudes of the electromagnetic
waves, propagating in the waveguiding structures and leakage waves, the model can be used for searching, and even
monitoring the external influences on the layered gyrotropic active artificial or natural media, for example, microwave or
optical waveguides or the system LAIM and the earth-ionosphere waveguide, respectively.
An important effect of the gyrotropy and anisotropy is the corresponding transformation of the field polarization
during the propagation in the WGEI, absent in the ideal metal planar waveguide without gyrotropy and anisotropy. We will
search, how such an effect depends on the carrier frequency of the beam, propagating in the WGEI, inclination of the
geomagnetic field and perturbations in the electron concentration, which could vary under the influences of the powerful
enough sources placed "below", "above" and/or "inside" the ionosphere.
In Sect. 2 formulation of the problem is presented. In Sect. 3 the algorithm is discussed including the determination
of the VLF waves/beams radiation conditions into the upper ionosphere/magnetosphere at the upper boundary, placed in the
F-region at the altitude (250-400) km. The effective tensor impedance boundary conditions at the upper boundary (~ 85 km)
of the effective earth-ionosphere waveguide and the 3D model TIMEB of the propagation of the VLF beam in the WGEI are
discussed as well. The issues regarding the VLF beam leakage regimes are considered only very briefly, since the relevant
details will be presented in the following articles. In Sect. 4 the results of the numerical modeling are presented. In Sect. 5
the discussion is presented, including an example of the qualitative comparison between the results of our theory and an
experiment including the future rocket experiment on the measurements of the characteristics of VLF signal radiated from
the artificial VLF transmitter, which is propagating in the WGEI and penetrating into the upper ionosphere.

**2 Formulation of the problem**

The VLF electromagnetic (EM) waves with frequencies $f = (10 – 100)$ kHz can propagate along the Earth's surface for long
distances >1000 km. The Earth's surface of a high conductivity z = 0 ( $z$ is vertical coordinate) and the ionosphere $F$-layer $z$
= 300 km form the VLF waveguide, see Fig. 1. The propagation of the VLF electromagnetic radiation excited by a near-
Earth antenna within the WGEI should be described by the full set of Maxwell's equations in the isotropic atmosphere $0 < z$
$< 60$ km, the approximately isotropic ionosphere $D$-layer 60 km $< z < 75$ km, and the anisotropic $E$- and $F$- layers of the
ionosphere, due to the geomagnetic field $\vec{H}_0$, added by the boundary conditions at the Earth's surface and at the $F$-layer. In
Fig. 1, $\theta$ is the angle between the directions of the vertical axis $z$ and geomagnetic field $\vec{H}_0$. Note that theta $\theta$ angle is
complementary to the angle of inclination of the geomagnetic field. Geomagnetic field $\vec{H}_0$ is directed along $z'$ axis, lies in
the plane $xz$, while the planes $x'z'$ and $xz$ coincide with each other.

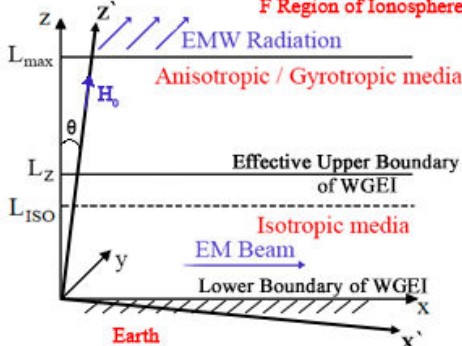


**Figure 1. The geometry of the anisotropic/gyrotropic waveguide. EM waves propagate in OX direction. $\vec{H}_0$ is the external**
**magnetic field. The effective WGEI for EM waves occupies the region $0 < z < L_z$. The isotropic medium occupies the region $0 < z <$**
**$L_{ISO}$, $L_{ISO} < L_z$. The anisotropic/gyrotropic medium occupies the region $L_{ISO} < z < L_{max}$. Covering layer occupies the region $L_z < z <$**
**$L_{max}$. WG includes isotropic region $0 < z < L_{ISO}$ and a part of anisotropic region $L_z < z < L_{max}$. It is supposed that the anisotropic**
**region is relatively small part of the WG, $(L_z - L_{ISO})/L_z \sim (0.1\text{-}0.2)$. At the upper boundary of covering layer ($z = L_{max}$) the radiation**
**of EM to the external region ($z > L_{max}$) is accounted for with the proper boundary conditions. Integration of the equations**
**describing the EM field propagation allows to obtain effective impedance boundary conditions at the upper boundary of effective**
**WG ($z = L_z$). These boundary conditions effectively includes all the effects on the wave propagation of the covering layer and the**
**radiation (at $z = L_{max}$) to the external region ($z > L_{max}$).**

**3. Algorithm**
The boundary conditions and calculations of impedance and beam propagation in the WGEI is considered in this Secion. The
other parts of the algorithm, e.g. the reflection of the EM waves from the WGEI effective upper boundary and the leakage of
EM waves from the WGEI to the upper ionosphere/magnetosphere, will be presented very briefly as they are the subjects of
the next papers.

## 3.1 Direct and inverse tensors characterizing the ionosphere

In the next subsections we derived the formulas describing the transfer of the boundary conditions at the upper boundary ($z =$
$L_{max}$), Fig. 1, resulting in the tensor impedance conditions at the upper boundary of the effective WGEI ($z = L_i$). Firstly let's
describe the tensors characterizing the ionosphere.
The algorithm's main goal is to transfer the EM boundary conditions from the upper ionosphere at the height $L_z \sim 250 - 400$
km to the lower ionosphere $L_z \sim 70 - 90$ km. All components of the monochromatic EM field is considered to be
proportional to exp$(i\omega t)$. The anisotropic medium is inhomogeneous along $OZ$ axis only and characterized by the
permittivity tensor $\hat{\varepsilon}(\omega, z)$ or by the inverse tensor $\hat{\beta}(\omega, z) = \hat{\varepsilon}^{-1}(\omega, z)$: $\vec{E} = \hat{\beta}(\omega, z) \cdot \vec{D}$, where $\vec{D}$ is the electric induction.
Below the absolute units are used. The expressions for the components of the effective permittivity of the ionosphere are in
the coordinate frame $X'YZ'$ where $OZ'$ axis is aligned along the geomagnetic field $\vec{H}_0$:

$$\hat{\varepsilon}' = \begin{pmatrix} \varepsilon_1 & \varepsilon_h & 0 \\ -\varepsilon_h & \varepsilon_1 & 0 \\ 0 & 0 & \varepsilon_3 \end{pmatrix}, \quad \varepsilon_1 = 1 - \frac{\omega_{pe}^2 \cdot (\omega - i v_e)}{((\omega - i v_e)^2 - \omega_{He}^2) \cdot \omega} - \frac{\omega_{pi}^2 \cdot (\omega - i v_i)}{((\omega - i v_i)^2 - \omega_{Hi}^2) \cdot \omega}, \quad \varepsilon_h \equiv ig;$$

$$g = -\frac{\omega_{pe}^2 \cdot \omega_{He}}{((\omega - i v_e)^2 - \omega_{He}^2) \cdot \omega} + \frac{\omega_{pi}^2 \cdot \omega_{Hi}}{((\omega - i v_i)^2 - \omega_{Hi}^2) \cdot \omega}, \quad \varepsilon_3 = 1 - \frac{\omega_{pe}^2}{(\omega - i v_e) \cdot \omega} - \frac{\omega_{pi}^2}{(\omega - i v_i) \cdot \omega};$$  (1)

$$\omega_{pe}^2 = \frac{4\pi e^2 n}{m_e}, \quad \omega_{pi}^2 = \frac{4\pi e^2 n}{m_i}, \quad \omega_{He} = \frac{eH_0}{m_e c}, \quad \omega_{Hi} = \frac{eH_0}{m_i c}$$


Here $\omega_{pe}, \omega_{pi}, \omega_{He}, \omega_{Hi}$ are the plasma and cyclotron frequencies for electrons and ions, respectively; $m_e$, $m_i$, $v_e$, $v_i$ are the
masses and collision frequencies for electrons and ions, respectively, $n$ is electron concentration. The approximations of
the three-component plasma-like ionosphere (including electron, one effective ion and one effective neutral
components) and quasi-neutrality are accepted. The expressions for the components of permittivity tensor $\hat{\varepsilon}(\omega, z)$ are
obtained from (1) by means of multiplication with the standard rotation matrices (Spiegel, 1959) dependent on angle $\theta$. For
the medium with a scalar conductivity $\sigma$, e.g. lower ionosphere or atmosphere, the effective permittivity in (1) reduces to $\varepsilon =$
1 - $4\pi i\sigma/\omega$.

**3.2 The equations for the EM field and upper boundary conditions**

The case of the VLF waveguide modes $k_x$ is slightly complex and should be calculated accounting for boundary conditions at
the Earth's surface and upper surface of the effective WGEI. The EM field depends on the horizontal coordinate $x$ as
exp($-k_x x$). Taking into account $k_x \le k_0$ ($k_0 = \omega/c$), in simulations of the VLF beam propagation, we assume $k_x = k_0$.
Therefore, Maxwell's equations are:

$$-\frac{\partial H_y}{\partial z} = ik_0 D_x, \quad \frac{\partial H_x}{\partial z} + ik_x H_z = ik_0 D_y, \quad -ik_x H_y = ik_0 D_z$$

$$-\frac{\partial E_y}{\partial z} = -ik_0 H_x, \quad \frac{\partial E_x}{\partial z} + ik_x E_z = -ik_0 H_y, \quad -ik_x E_y = -ik_0 H_z$$  (2)

Here, $E_x = \beta_{11} D_x + \beta_{12} D_y + \beta_{13} D_z$ etc. All components of the EM field can be represented through the horizontal components
of the magnetic field $H_x$, and $H_y$. The equations for these components are:
$$\frac{\partial}{\partial z}\left(\frac{\beta_{22}}{1-\beta_{22}\frac{k_x^2}{k_0^2}}\frac{\partial H_x}{\partial z}\right)-\frac{\partial}{\partial z}\left(\frac{\beta_{21}}{1-\beta_{22}\frac{k_x^2}{k_0^2}}\frac{\partial H_y}{\partial z}\right)-ik_x\frac{\partial}{\partial z}\left(\frac{\beta_{23}}{1-\beta_{22}\frac{k_x^2}{k_0^2}}H_y\right)+k_0^2H_x=0 \tag{3a}$$

$$\frac{\partial}{\partial z}\left((\beta_{11}+\frac{k_x^2}{k_0^2}\frac{\beta_{12}\cdot\beta_{21}}{1-\beta_{22}\frac{k_x^2}{k_0^2}})\frac{\partial H_y}{\partial z}\right)-\frac{\partial}{\partial z}\left(\frac{\beta_{12}}{1-\beta_{22}\frac{k_x^2}{k_0^2}}\frac{\partial H_x}{\partial z}\right)+$$

$$+ik_x\frac{\partial}{\partial z}\left((\beta_{13}+\frac{k_x^2}{k_0^2}\frac{\beta_{12}\cdot\beta_{23}}{1-\beta_{22}\frac{k_x^2}{k_0^2}})H_y\right)+ik_x(\beta_{31}+\frac{k_x^2}{k_0^2}\frac{\beta_{32}\cdot\beta_{21}}{1-\beta_{22}\frac{k_x^2}{k_0^2}})\frac{\partial H_y}{\partial z}- \tag{3b}$$

$$-ik_x\frac{\beta_{32}}{1-\beta_{22}\frac{k_x^2}{k_0^2}}\frac{\partial H_x}{\partial z}+k_0^2(1-\beta_{33}\frac{k_x^2}{k_0^2}-\frac{k_x^4}{k_0^4}\frac{\beta_{23}\cdot\beta_{32}}{1-\beta_{22}\frac{k_x^2}{k_0^2}})H_y=0$$

The expressions for the horizontal components of the electric field $E_x$, $E_y$ are:
$$E_x=\frac{i}{k_0}\left((\beta_{11}+\frac{k_x^2}{k_0^2}\frac{\beta_{12}\cdot\beta_{21}}{1-\beta_{22}\frac{k_x^2}{k_0^2}})\frac{\partial H_y}{\partial z}-\frac{\beta_{12}}{1-\beta_{22}\frac{k_x^2}{k_0^2}}\frac{\partial H_x}{\partial z}\right)-\frac{k_x}{k_0}(\beta_{13}+\frac{k_x^2}{k_0^2}\frac{\beta_{12}\cdot\beta_{23}}{1-\beta_{22}\frac{k_x^2}{k_0^2}})H_y$$

$$E_y=\frac{i}{k_0}\left(-\frac{\beta_{22}}{1-\beta_{22}\frac{k_x^2}{k_0^2}}\frac{\partial H_x}{\partial z}+\frac{\beta_{21}}{1-\beta_{22}\frac{k_x^2}{k_0^2}}\frac{\partial H_y}{\partial z}\right)-\frac{k_x}{k_0}\frac{\beta_{23}}{1-\beta_{22}\frac{k_x^2}{k_0^2}}H_y \tag{4}$$

In the region $z\geq L_{max}$ the upper ionosphere is assumed to be weakly inhomogeneous, and the geometric optics approximation
is valid in the VLF range there. We should note that due to high inhomogeneity of the ionosphere in the vertical direction
within $E$-layer (i.e. at the upper boundary of the effective VLF WGEI) such an approximation is not applicable. These
conditions determine the choice of the upper boundary $z=L_{max}\sim(250-400)$ km, where the conditions of the radiation are
formulated. The dispersion equation connected the wave numbers and the frequency of the outgoing waves is obtained from
Eqs. (3), where $H_{x,y}\sim\exp(-ik_z\tilde{z})$, while the derivatives like $\partial\beta_{11}/\partial z$ and the inhomogeneity of the media are neglected:

$$\left( \beta_{22} k_z^2 - k_0^2 (1 - \beta_{22}\frac{k_x^2}{k_0^2}) \right) \cdot \left( (\beta_{11}(1 - \beta_{22}\frac{k_x^2}{k_0^2}) + \frac{k_x^2}{k_0^2}\beta_{12} \cdot \beta_{21})k_z^2 + ((\beta_{13} + \beta_{31})(1 - \beta_{22}\frac{k_x^2}{k_0^2}) + \right.$$

$$\left. + \frac{k_x^2}{k_0^2}(\beta_{12} \cdot \beta_{23} + \beta_{32} \cdot \beta_{21})k_x k_z - k_0^2((1 - \beta_{33}\frac{k_x^2}{k_0^2})(1 - \beta_{22}\frac{k_x^2}{k_0^2}) - \frac{k_x^4}{k_0^4}\beta_{23} \cdot \beta_{32}) \right) - $$

$$- \left( \beta_{21} k_z^2 + \beta_{23} k_x k_z \right) \cdot \left( \beta_{12} k_z^2 - \beta_{32} k_x k_z \right) = 0 \qquad (5)$$

Thus, generally Eq. (5) determined the wave numbers for the outgoing waves is of the fourth order (Wait 1996). The
boundary conditions at the upper boundary $z = L_{max}$ within the ionosphere $F$-layer are the absence of the ingoing waves, i.e.
the outgoing radiated (leakage) waves are present only. Two roots should be selected that possess the negative imaginary
parts $Im(k_{z1, z2}) < 0$, i.e. the outgoing waves dissipate upwards. However, in the case of VLF waves, some simplification can
be used. Namely, the expressions for the wave numbers $k_{1,2}$ are obtained from Eqs. (3), where the dependence on $x$ is
neglected: $|k_{1,2}| \gg k_0$. This approximation is valid within $F$-layer where the first outgoing wave corresponds to the whistlers
with small dissipation, the second one is the highly dissipating slow wave. To formulate the boundary conditions for Eqs.
(3a, b) at $z \geq L_{max}$, the EM field components can be presented as:

$$H_x = A_1 e^{-ik_{z1}\tilde{z}} + \alpha_2 A_2 e^{-ik_{z2}\tilde{z}}, \quad H_y = \alpha_1 A_1 e^{-ik_{z1}\tilde{z}} + A_2 e^{-ik_{z2}\tilde{z}} \qquad (6)$$

In the relations (6) $\tilde{z} = z - L_z$. Eqs. (3) are simplified in the approximation described above:

$$\beta_{22}\frac{\partial^2 H_x}{\partial z^2} - \beta_{21}\frac{\partial^2 H_y}{\partial z^2} + k_0^2 H_x = 0, \ \beta_{11}\frac{\partial^2 H_y}{\partial z^2} - \beta_{12}\frac{\partial^2 H_x}{\partial z^2} + k_0^2 H_y = 0 \qquad (7)$$

The solution of Eqs. (7) is searched for as: $H_{x,y} \sim \exp(-ik_z\tilde{z})$. The following equation has been obtained to get the wave
numbers $k_{z1, z2}$ from Eqs. (7):

$$\kappa^4 - (\beta_{22} + \beta_{11})\kappa^2 + \beta_{11}\beta_{22} - \beta_{12}\beta_{21} = 0, \quad \kappa^2 = \frac{k_0^2}{k_z^2} \qquad (8)$$

Therefore, from Eq. (8) follows,

$$\kappa_{1,2}^2 = \frac{\beta_{11} + \beta_{22}}{2} \pm \left( (\frac{\beta_{11} - \beta_{22}}{2})^2 + \beta_{12}\beta_{21} \right)^{1/2}; \alpha_1 = \frac{\beta_{22} - \kappa_1^2}{\beta_{21}} = \frac{\beta_{12}}{\beta_{11} - \kappa_1^2}; \alpha_2 = \frac{\beta_{11} - \kappa_2^2}{\beta_{12}} = \frac{\beta_{21}}{\beta_{22} - \kappa_2^2}; k_{z1,z2}^2 = \frac{k_0^2}{\kappa_{1,2}^2} \qquad (9)$$

The signs of $k_{z1, z2}$ have been chosen from the condition $Im(k_{z1, z2}) < 0$. From Eq. (5) at the upper boundary $z = L_{max}$, the
following relations are valid:

$$H_x = A_1 + \alpha_2 A_2, \quad H_y = \alpha_1 A_1 + A_2 \qquad (10)$$

From Eq. (10) one can get
$$A_1 = \Delta^{-1}(H_x - \alpha_2 H_y); A_2 = \Delta^{-1}(H_y - \alpha_1 H_x); \Delta = 1 - \alpha_1\alpha_2 \qquad (11)$$

Thus, it is possible to exclude the amplitudes of the outgoing waves $A_{1,2}$ from Eqs. (9). As a result, at $z = L_{max}$ the boundary
conditions are rewritten in terms of $H_x$, $H_y$ only:

$$\frac{\partial H_x}{\partial z} = -i(k_{z1}A_1 + k_{z2}\alpha_2 A_2) = -\frac{i}{\Delta}\left((k_{1z} - \alpha_1\alpha_2 k_{z2})H_x + \alpha_2(k_{z2} - k_{z1})H_y\right)$$

$$\frac{\partial H_y}{\partial z} = -i(k_{z1}\alpha_1 A_1 + k_{z2}A_2) = -\frac{i}{\Delta}\left((k_{z2} - \alpha_1\alpha_2 k_{z1})H_y + \alpha_1(k_{z1} - k_{z2})H_x\right)$$

$$\qquad (12)$$

The relations (12) are the upper boundary conditions of the radiation for the boundary $z=L_{max}\sim 250 - 400$ km. These
conditions will be transformed/recalculated using the analytical numerical recurrent procedure into equivalent impedance
boundary conditions at $z = L_z \sim 70 - 90$ km.
Note that in the "whistler/VLF approximation" is valid at frequencies ~ 10 kHz for the $F$-region of the ionosphere.
In this approximation and $k_x \approx 0$, we receive the dispersion equation using Eqs. (5), (8), (9), in the form:

$$k_z^{'2}k^2 = k_0^2 g^2 \qquad (13)$$

where $k^2 = k_x^2 + k_z^2 = k_x^{'2} + k_z^{'2}$; $k_x'$ and $k_z'$ are the transverse and longitudinal components of wave number relative to
geomagnetic field. For $F$-region of the ionosphere, where $\nu_e \ll \omega \ll \omega_{He}$, Eq. (13) reduces to the standard form of the
whistler dispersion equation $|k_z'|k = k_0|g|$; $g \approx -\omega_{pe}^2/(\omega\omega_{He})$; $\omega = c^2 k|k_z'|(\omega_{He}/\omega_{pe}^2)$. In a special case of the waves
propagating along geomagnetic field, $k_x' = 0$, for the propagating whistler waves, well-known dispersion dependence is
$\omega = c^2 k_z^{'2}(\omega_{He}/\omega_{pe}^2)$ (Artcimovich and Sagdeev, 1979). For the formulated problem we can reasonably assume $k_x \approx 0$.
Therefore eq. (13) is reduced to $k_z^4 \cos^2\theta = k_0^4 g^2$. As a result, we get $k_{z1} = \sqrt{g/\cos\theta}k_0$, $k_{z2} = -i\sqrt{g/\cos\theta}k_0$, and then,
similarly to the relations (12), the boundary conditions can be presented, in terms of the tangential components of electric
field as:
$$\frac{\partial \vec{U}}{\partial z} + \hat{B}\vec{U} = 0; \vec{U} = \begin{bmatrix} E_x \\ E_y \end{bmatrix}; \hat{B} = \frac{1}{2}\sqrt{\frac{g}{\cos\theta}}k_0 \begin{bmatrix} 1+i & -1-i \\ 1+i & 1+i \end{bmatrix} \qquad (14)$$


Conditions (12) or (14) are the conditions of radiation (absence of ingoing waves) formulated at the upper boundary $z=L_{max}$
and suitable for the determination of the energy of the wave leaking from the WGEI into the upper
ionosphere/magnetosphere. Note, the equations (12), (14) expressed the boundary conditions of the radiation (more
accurately speaking, an absence of incoming waves, what is the consequence to the causality principle), are obtained as a
result of limitation by the small parameter $k_x/k_0 |k_x/k_z| \to 0$ in Eq. (5). In spite of the disappearance of the dependence of
these boundary conditions explicitly on $k_x$, the dependence of the characteristics of the wave propagation process on $k_x$, as a
whole, is accounted for, and all results are still valid for the description of the wave beam propagation in the WGEI along the
horizontal axis $x$ with finite $k_x \sim k_0$.

## 3.3 Equivalent Tensor Impedance Boundary Conditions

The tensor impedance at the upper boundary of the effective WGEI $z=L_z$ (see Fig. 1), is obtained by the conditions of radiation (12) or (14), recalculated from the level $z = L_z \sim 80 - 90$ km, placed in the F region of the ionosphere, at $z=L_{max}\sim$ 250 – 400 km.

The main idea of the effective tensor impedance method is the unification of analytical and numerical approaches and derivation of the proper impedance boundary conditions without "thin cover layer" approximation. This approximation is usually used in the effective impedance approaches, applied either for artificial or natural layered gyrotropic structures, (see e.g. Tretyakov, 2003; Senior and Volakis, 1995; Kurushin and Nefedov, 1983; Alperovich and Fedorov, 2007). There is one known exception, namely invariant imbedding impedance method (Shalashov and Gospodchikov, 2011; Kim and Kim, 2016). The comparison of our method with the invariant imbedding impedance method will be presented at the end of this subsection. Eqs. (3) jointly with the boundary conditions (12), have been solved by finite differences. The derivatives in Eqs. (3) are approximated as

$$\frac{\partial}{\partial z}\left(C(z)\frac{\partial H_x}{\partial z}\right) \approx \frac{1}{h}\left(C(z_{j+1/2})\frac{(H_x)_{j+1}-(H_x)_j}{h} - C(z_{j-1/2})\frac{(H_x)_j-(H_x)_{j-1}}{h}\right),$$

$$\frac{\partial}{\partial z}\left(F(z)H_x\right) \approx \frac{1}{2h}\left(F(z_{j+1})(H_x)_{j+1} - F(z_{j-1})(H_x)_{j-1}\right) \quad \text{etc.} \tag{15}$$

In Eq. (15) $z_{j+1/2} = h\cdot(j+0.5)$. In Eqs. (10) the approximation is $\partial H_x / \partial z \approx [(H_x)_N - (H_x)_{N-1}]/h$. Here $h$ is the discretization step along $OZ$ axis; $N$ is the total number of nodes. At each step $j$ the difference approximations of Eqs. (3) take the form:

$$\hat{\alpha}_j^{(-)} \cdot \vec{H}_{j-1} + \hat{\alpha}_j^{(0)} \cdot \vec{H}_j + \hat{\alpha}_j^{(+)} \cdot \vec{H}_{j+1} = 0 \tag{16}$$

where $\vec{H}_j = \begin{pmatrix} H_x \\ H_y \end{pmatrix}$, $j = N-1, N-2,...,1$, $z_j = h\cdot j$, $L_z = h\cdot N$. Due to the complexity of expressions for the matrix coefficients in Eq. (16) we have shown them in Appendix. The set of the matrix Eq. (16) has been solved by factorization method also known as an elimination/matrix sweep method (see Samarskii, 2001). It can be written as:

$$\vec{H}_j = \hat{b}_j \cdot \vec{H}_{j-1}, \quad j = N,...,1 \tag{17a}$$

$$H_{xj+1} = b_{11j+1}H_1 + b_{12j+1}H_2; \ H_{yj+1} = b_{21j+1}H_1 + b_{22j+1}H_2; \ H_1 \equiv H_{xj}; \ H_2 \equiv H_{yj} \tag{17b}$$

This method is a variant of the Gauss elimination method for the matrix 3-diagonal set of the Eq. (16). The value of $\hat{b}_N$ is obtained from the boundary conditions (12) as:

$$\hat{\alpha}_N^{(-)} \cdot \vec{H}_{N-1} + \hat{\alpha}_N^{(0)} \cdot \vec{H}_N = 0 \tag{18}$$

Therefore $\hat{b}_N = -(\hat{\alpha}_N^{(0)})^{-1} \cdot \hat{\alpha}_N^{(-)}$. Then the matrices $\hat{b}_j$ have been computed sequentially down to the desired value of z =
$L_z = h \cdot N_z$, where the impedance boundary conditions are assumed to be applied. At each step the expression for $\hat{b}_j$ follow
from (16), (17) as:
$$(\hat{\alpha}_j^{(0)} + \hat{\alpha}_j^{(+)} \cdot \hat{b}_{j+1}) \cdot \vec{H}_j = -\hat{\alpha}_j^{(-)} \cdot \vec{H}_{j-1} \tag{19}$$
Therefore, for (17), we obtain $\hat{b}_j = -(\hat{\alpha}_j^{(0)} + \hat{\alpha}_j^{(+)} \cdot \hat{b}_{j+1})^{-1} \cdot \hat{\alpha}_j^{(-)}$. The derivatives in Eqs. (4) have been approximated as:
$$\left(\frac{\partial H_x}{\partial z}\right)_{N_z} \approx \frac{(H_x)_{N_z+1} - (H_x)_{N_z}}{h} = \frac{(b_{N_z+1\ 11} - 1) \cdot (H_x)_{N_z} + b_{N_z+1\ 12} \cdot (H_y)_{N_z}}{h} \tag{20}$$
and similar equation can be obtained for $\left(\dfrac{\partial H_y}{\partial z}\right)_{N_z}$. Note, that as a result of this discretization, only the values at the grid
level $N_z$ are included into the numerical approximation of the derivatives $\partial H_{x,y} / \partial z$ at $z = L_z$. We determine tensor
impedance $\hat{Z}$ at $z=L_z \sim 85$ km level. The tensor values are included into the following relations, all of which are
corresponded to altitude (in other words, to the grid with number $N_z$, corresponded to this altitude):
$$\vec{n} \times \vec{E} = \hat{Z}_0 \cdot \vec{H}, \ \vec{n} = (0,0,1) \ \text{or} \ E_x = Z_{021}H_x + Z_{022}H_y; \ E_y = -Z_{011}H_x - Z_{012}H_y \tag{21}$$
The equivalent tensor impedance is obtained using a two-step procedure. (1) We obtain the matrix $\hat{b}_j$ using Eqs. (3a, b) with
the boundary conditions (12) and the procedure (17) – (19) described above. (2) Placing the expressions (21) with tensor
impedance into the left parts and the derivatives $\partial H_{x,y} / \partial z$ in the form (20) into the right parts of Eqs. (4), the analytical
expressions for the components of the tensor impedance are:
$$Z_{011} = -\frac{i}{k_0 h}\left( \frac{\beta_{21}}{1 - \beta_{22}\frac{k_x^2}{k_0^2}} \cdot b_{21} - \frac{\beta_{22}}{1 - \beta_{22}\frac{k_x^2}{k_0^2}} \cdot (b_{11} - 1) \right),$$
$$Z_{012} = -\frac{i}{k_0 h}\left( \frac{\beta_{21}}{1 - \beta_{22}\frac{k_x^2}{k_0^2}}\frac{\partial H_y}{\partial z} \cdot (b_{22} - 1) - \frac{\beta_{22}}{1 - \beta_{22}\frac{k_x^2}{k_0^2}} \cdot b_{12} - k_x h \cdot \frac{\beta_{23}}{1 - \beta_{22}\frac{k_x^2}{k_0^2}} \right),$$
$$Z_{021} = \frac{i}{k_0 h}\left( (\beta_{11} + \frac{k_x^2}{k_0^2}\frac{\beta_{12} \cdot \beta_{21}}{1 - \beta_{22}\frac{k_x^2}{k_0^2}}) \cdot b_{21} - \frac{\beta_{12}}{1 - \beta_{22}\frac{k_x^2}{k_0^2}} \cdot (b_{11} - 1) \right),$$

$$Z_{022} = \frac{i}{k_0 h}\left( (\beta_{11} + \frac{k_x^2}{k_0^2}\frac{\beta_{12}\cdot\beta_{21}}{1-\beta_{22}\frac{k_x^2}{k_0^2}})\cdot(b_{22}-1) - k_x h\cdot(\beta_{13} + \frac{k_x^2}{k_0^2}\frac{\beta_{12}\cdot\beta_{23}}{1-\beta_{22}\frac{k_x^2}{k_0^2}}) - \frac{\beta_{12}}{1-\beta_{22}\frac{k_x^2}{k_0^2}}\cdot b_{12} \right) \tag{22}$$

The proposed method of the transfer of the boundary conditions from the ionosphere $F$-layer $L_{max} = 250 - 400$ km into the lower part of the $E$-layer $L_z = 80 - 90$ km is stable and easily realizable in comparison with some alternative approaches based on the invariant imbedding methods (Shalashov and Gospodchikov, 2011; Kim and Kim, 2016). The stability of our method is due to the stability of the Gauss elimination method when the coefficients at the matrix central diagonal are dominating. The last is valid for the ionosphere with electromagnetic losses where the absolute values of the permittivity tensor are large. The application of the proposed matrix sweep method in the media without losses may require the use of the Gauss method with the choice of the maximum element, to ensure the stability. However, as our simulations (not presented here) demonstrated, for the electromagnetic problems in the frequency domain, the simple Gauss elimination and the choice of the maximal element give the same results. The accumulation of errors may occur in evolutionary problems in the time domain, when the Gauss method should be applied sequentially many times. The use of the independent functions $H_x$, $H_y$ in Eqs. (3) seems natural, as well as the transfer (17a), because the impedance conditions are the expressions of the electric $E_x$, $E_y$ through these magnetic components $H_x$, $H_y$ at the upper boundary of the VLF waveguide $80 - 90$ km. The naturally chosen direction of the recalculation of the upper boundary conditions from $z = L_{max}$ to $z = L_z$, i.e. from upper layer with large impedance value to lower altitude layer with relatively small impedance value, provides, at the same time, the stability of the simulation procedure. The obtained components of the tensor impedance $\hat{Z}_0$ are small, $|Z_{0\alpha\beta}| \leq 0.1$. This determines the choice of the upper boundary $z = L_z$ for the effective WGEI. Due to small impedance, EM waves incident from below on this boundary are reflected effectively back. Therefore, the region $0 \leq z \leq L_z$ indeed can be presented as an effective WGEI. This waveguide includes not only lower boundary at $L_{ISO} \sim 65 - 75$ km with rather small losses, but also thin dissipative and anisotropic/gyrotropic layer between 75 and $85 - 90$ km.

Finally, the main differences and advantages of the proposed tensor impedance method from other methods for impedance recalculating and in particular invariant imbedding methods (Shalashov and Gospodchikov, 2011; Kim and Kim, 2016) can be summarised as follows:

(i) in contrast to invariant imbedding method currently proposed method can be used for direct recalculation of tensor impedance as it determined analytically, see Eqs. (22).

(ii) for the media without non-locality, proposed method does not require to solve integral equation(s).

(iii) the proposed method does not require forward and reflected waves. The conditions for the radiation at the upper boundary $z = L_{max}$ (see Eqs. (12)) are determined through the total field components $H_{x,y}$, which simplify the overall calculations.

 (iv) the overall calculation procedure is very effective and computationally stable. Note, that even for the very low-

loss systems, the required level of stability can be achieved with modification based on the choice of the maximal element
for matrix inversion.

## 3.4 Propagation of Electromagnetic Waves in the Gyrotropic Waveguide and the TIMEB Method

Let's use the transverse components of electric $E_y$, and magnetic $H_y$ fields to derive equations for the slow varying
amplitudes $A(x,y,z)$, $B(x,y,z)$ of the VLF beams. These components can be represented as:

$$E_y = \frac{1}{2} A(x, y, z) \cdot e^{i\omega t - ik_0 x} + c.c., \quad H_y = \frac{1}{2} B(x, y, z) \cdot e^{i\omega t - ik_0 x} + c.c. \tag{23}$$

Here we assumed $k_x = k_0$ to reflect beam propagation in the WGEI with the main part in the atmosphere and lower
ionosphere (*D*-region) which are similar to free space by its electromagnetic parameters. The presence of a thin anisotropic
and dissipative layer belonging to the *E*-region (Guglielmi and Pokhotelov, 1996) of the ionosphere causes, altogether with
the impedance boundary condition, the proper *z* dependence of $B(x,y,z)$. Using (21) and (22), the boundary conditions are
determined at the height $z = L_z$ for the slowly varying amplitudes $A(x,y,z)$, $B(x,y,z)$ of the transverse components $E_y$, $H_y$. As it
follows from Maxwell's equations, the components $E_x$ and $H_x$ through $E_y$, $H_y$ in the method of beams have the form:

$$H_x \approx -\frac{i}{k_0} \frac{\partial E_y}{\partial z}, \quad E_x \approx \gamma_{12} E_y + i \frac{\tilde{\beta}_{33}}{k_0} \frac{\partial H_y}{\partial z} + \tilde{\beta}_{13} H_y \tag{24}$$

where $\gamma_{12} = \Delta_0^{-1}(\varepsilon_{13}\varepsilon_{32} - \varepsilon_{12}\varepsilon_{33}), \; \tilde{\beta}_{13} = \Delta_0^{-1}\varepsilon_{13}, \; \tilde{\beta}_{33} = \Delta_0^{-1}\varepsilon_{33}; \; \Delta_0 = \varepsilon_{11}\varepsilon_{33} - \varepsilon_{13}\varepsilon_{31}$. From Eqs. (21) and (24), the boundary
conditions for *A, B* can be defined as:

$$A - \frac{i}{k_0} Z_{11} \cdot \frac{\partial A}{\partial z} + Z_{12} \cdot B \approx 0, \quad \gamma_{12} \cdot A + \frac{i}{k_0} Z_{21} \cdot \frac{\partial A}{\partial z} + (\tilde{\beta}_{13} - Z_{22}) \cdot B + \frac{i}{k_0} \tilde{\beta}_{33} \cdot \frac{\partial B}{\partial z} \approx 0 \tag{25}$$

The evolution equations for the slowly varying amplitudes $A(x,y,z)$, $B(x,y,z)$ of the VLF beams are derived. The
monochromatic beams are considered, when the frequency $\omega$ is fixed and the amplitudes do not depend on time *t*. Looking
for the solutions for the EM field as $\vec{E}, \vec{H} \sim \exp(i\omega t - ik_x x - ik_y y)$, Maxwell's equations are:

$$-ik_y H_z - \frac{\partial H_y}{\partial z} = ik_0 D_x, \quad \frac{\partial H_x}{\partial z} + ik_x H_z = ik_0 D_y, \quad -ik_x H_y + ik_y H_x = ik_0 D_z$$

$$-ik_y E_z - \frac{\partial E_y}{\partial z} = -ik_0 H_x, \quad \frac{\partial E_x}{\partial z} + ik_x E_z = -ik_0 H_y, \quad -ik_x E_y + ik_y E_x = -ik_0 H_z \tag{26}$$

Here $D_x = \varepsilon_{11} E_x + \varepsilon_{12} E_y + \varepsilon_{13} E_z$. From Eqs. (21), the equations for $E_x$, $E_z$ through $E_y$, $H_y$ are:

$$E_x = \frac{1}{\Delta_y}\left\{[\varepsilon_{13}\cdot\varepsilon_{32} - (\varepsilon_{12} + \frac{k_x k_y}{k_0^2})\cdot(\varepsilon_{33} - \frac{k_y^2}{k_0^2})]E_y + \frac{i}{k_0}(\varepsilon_{33} - \frac{k_y^2}{k_0^2})\frac{\partial H_y}{\partial z} + \frac{k_x}{k_0}\varepsilon_{13}\cdot H_y + \frac{ik_y}{k_0^2}\varepsilon_{13}\frac{\partial E_y}{\partial z}\right\}$$

$$E_z = \frac{1}{\Delta_y}\left\{[\varepsilon_{31}\cdot(\varepsilon_{12} + \frac{k_x k_y}{k_0^2}) - \varepsilon_{32}\cdot(\varepsilon_{11} - \frac{k_y^2}{k_0^2})]E_y - \frac{i}{k_0}\varepsilon_{31}\frac{\partial H_y}{\partial z} - \frac{k_x}{k_0}\cdot(\varepsilon_{11} - \frac{k_y^2}{k_0^2})H_y - \frac{ik_y}{k_0^2}\cdot(\varepsilon_{11} - \frac{k_y^2}{k_0^2})\frac{\partial E_y}{\partial z}\right\}$$

(27)

In Eq. (27), $\Delta_y \equiv (\varepsilon_{11} - \frac{k_y^2}{k_0^2})\cdot(\varepsilon_{33} - \frac{k_y^2}{k_0^2}) - \varepsilon_{31}\cdot\varepsilon_{13}$. The equations for $E_y$, $H_y$ obtained from the Maxwell equations are:

$$\left(\frac{\partial^2}{\partial z^2} - k_x^2 - k_y^2\right)E_y + ik_y\left(\frac{\partial E_z}{\partial z} - ik_x E_x - ik_y E_y\right) + k_0^2 D_y = 0; \quad -ik_0\frac{\partial E_x}{\partial z} + k_x k_0 E_z + k_0^2 H_y = 0$$

(28)

After substitution of (27) for $E_x$, $E_z$ into Eqs. (28), the coupled equations for $E_y$, $H_y$ can be derived. The follow expansion should be used: $k_x = k_0 + \delta k_x$, $|\delta k_x| \ll k_0$, also $|k_y| \ll k_0$. Then, according to (Weiland and Wilhelmsson, 1977):

$$-i\cdot\delta k_x \to \frac{\partial}{\partial x}, \quad -i\cdot k_y \to \frac{\partial}{\partial y}$$

(29)

The expansions should be until the quadratic terms of $k_y$ and the linear terms of $\delta k_x$. As a result, parabolic equations (Levy 2000) for the slowly varying amplitudes $A$ and $B$ are derived. In the lower ionosphere/atmosphere, where the effective permittivity reduces to a scalar $\varepsilon(\omega,z)$, they are independent:

$$\frac{\partial A}{\partial x} + \frac{i}{2k_0}\left(\frac{\partial^2 A}{\partial y^2} + \frac{\partial^2 A}{\partial z^2}\right) + \frac{ik_0}{2}\cdot(\varepsilon - 1)A = 0$$

$$\frac{\partial B}{\partial x} + \frac{i}{2k_0}\left(\frac{1}{\beta}\frac{\partial}{\partial z}(\beta\frac{\partial B}{\partial z}) + \frac{\partial^2 B}{\partial y^2}\right) + \frac{ik_0}{2}\cdot(\varepsilon - 1)B = 0$$

(30a)

Here $\beta \equiv \varepsilon^{-1}$. Accounting for the presence of gyrotropic layer and the tensor impedance boundary conditions at the upper boundary $z = L_z$ of the VLF waveguide, the equations for the slowly varying amplitudes in the general case are:

$$\frac{\partial A}{\partial x} + \frac{i}{2k_0}\left(\frac{\partial^2 A}{\partial y^2} + \frac{\partial^2 A}{\partial z^2}\right) + \frac{ik_0}{2}\cdot(\tilde{\varepsilon}_{22} - 1)\cdot A + \frac{\gamma_{21}}{2}\frac{\partial B}{\partial z} + \frac{ik_0}{2}\cdot\gamma_{23}B = 0$$

$$\frac{\partial B}{\partial x} + \frac{i}{2k_0}\left(\frac{1}{\tilde{\beta}_{11}}\frac{\partial}{\partial z}(\tilde{\beta}_{33}\frac{\partial B}{\partial z}) + \frac{\partial^2 B}{\partial y^2}\right) + \frac{i}{2\tilde{\beta}_{11}}\frac{\partial}{\partial z}(\gamma_{12}A) + \frac{1}{2\tilde{\beta}_{11}}\frac{\partial}{\partial z}(\tilde{\beta}_{13}B) + \frac{ik_0}{2\tilde{\beta}_{11}}\gamma_{32}A + \frac{\tilde{\beta}_{31}}{2\tilde{\beta}_{11}}\frac{\partial B}{\partial z} + \frac{ik_0}{2}\cdot(\frac{1}{\tilde{\beta}_{11}} - 1)\cdot B = 0$$

(30b)

In Eqs. (30b),

$$\gamma_{12} \equiv \frac{\varepsilon_{13}\cdot\varepsilon_{32} - \varepsilon_{12}\cdot\varepsilon_{33}}{\Delta}, \gamma_{21} \equiv \frac{\varepsilon_{23}\cdot\varepsilon_{31} - \varepsilon_{21}\cdot\varepsilon_{33}}{\Delta}, \gamma_{23} \equiv \frac{\varepsilon_{21}\cdot\varepsilon_{13} - \varepsilon_{23}\cdot\varepsilon_{11}}{\Delta}, \gamma_{32} \equiv \frac{\varepsilon_{31}\cdot\varepsilon_{12} - \varepsilon_{32}\cdot\varepsilon_{11}}{\Delta}, \tilde{\beta}_{11} \equiv \frac{\varepsilon_{11}}{\Delta}, \tilde{\beta}_{13} \equiv \frac{\varepsilon_{13}}{\Delta},$$

$$\tilde{\beta}_{31} \equiv \frac{\varepsilon_{31}}{\Delta}, \tilde{\beta}_{33} \equiv \frac{\varepsilon_{33}}{\Delta}; \Delta \equiv \varepsilon_{11}\cdot\varepsilon_{33} - \varepsilon_{13}\cdot\varepsilon_{31}; \tilde{\varepsilon}_{22} \equiv \varepsilon_{22} + \frac{\varepsilon_{21}(\varepsilon_{13}\varepsilon_{32} - \varepsilon_{12}\varepsilon_{33}) + \varepsilon_{23}(\varepsilon_{31}\varepsilon_{12} - \varepsilon_{32}\varepsilon_{11})}{\Delta}$$

Eqs. (30b) are reduced to Eqs. (30a) when the effective permittivity is scalar. At the Earth's surface $z = 0$, the impedance conditions are reduced, accounting for that the medium is isotropic and the conductivity of the Earth is finite, to the form:

$$E_y = Z_{0E} H_x, \quad E_x = -Z_{0E} H_y, \quad Z_{0E} \equiv \left( \frac{i\omega}{4\pi\sigma_E} \right)^{1/2} \tag{31a}$$

Here $\sigma_E \sim 10^8$ s$^{-1}$ is the Earth's conductivity. The boundary conditions (31a) at the Earth's surface, where $Z_{022} = Z_{021} \equiv Z_{0E}$, $Z_{012} = Z_{021} = 0$, $\beta_{33} = \varepsilon^{-1}(z = 0)$, $\gamma_{12} = 0$, $\tilde{B}_{13} = 0$, can be rewritten as

$$E_y + \frac{i}{k_0} Z_{0E} \frac{\partial E_y}{\partial z} = 0, \quad \frac{i}{\varepsilon(z = 0)k_0} \frac{\partial H_y}{\partial z} + Z_{0E} H_y = 0 \tag{31b}$$

Eqs. (30), combined with the boundary conditions (25) at the upper boundary of the VLF waveguide $z = L_z$, and with the boundary conditions at the Earth's surface (31b), are used to simulate the VLF wave propagation. The surface impedance of the Earth has been calculated from the Earth's conductivity, see eq. (31a). The boundary conditions to solution of Eqs. (30), (25), (31b) are chosen in the form

$$A(x = 0, y, z) = 0, \ B(x = 0, y, z) = B_0 \exp\left(-\left((y - 0.5L_y)/y_0\right)^{2n}\right) \exp\left(-\left((z - z_1)/z_0\right)^{2n}\right), \ n = 2 \tag{32}$$

In relations (32), $z_1$, $z_0$, $y_0$ and $B_0$ are the vertical position of maximum value, the vertical and transverse characteristic dimensions of the spatial distribution and the maximum value of $H_y$, respectively at the input of the system, $x = 0$. The size of the computing region along $OY$ axis is, by the order of value, $L_y \sim 2000$ km. Because the gyrotropic layer is relatively thin and is placed at the upper part of the VLF waveguide, the beams are excited near the Earth's surface, the wave diffraction in this gyrotropic layer along $OY$ axis is quite small, i.e. the terms $\partial^2 A/dy^2$, $\partial^2 B/dy^2$ are small there as well.

Contrary to this, the wave diffraction is very important in the atmosphere in the lower part of the VLF waveguide, near the Earth's surface. To solve the problem of the beam propagation, the method of splitting with respect to physical factors has been applied (Samarskii 2001). Namely, the problem has been approximated by the finite differences:

$$\vec{C} \equiv \begin{pmatrix} A \\ B \end{pmatrix}, \quad \frac{\partial \vec{C}}{\partial x} + \hat{L}_y \vec{C} + \hat{L}_z \vec{C} = 0 \tag{33}$$

In the terms $\hat{L}_y \vec{C}$, the derivatives with respect to $y$ are included, whereas all other terms are included into $\hat{L}_z \vec{C}$. Then the following fractional steps have been applied, the first one is along $y$, the second one is along $z$:

$$\frac{\vec{C}^{p+1/2} - \vec{C}^p}{h_x} + \hat{L}_y \vec{C}^{p+1/2} = 0, \quad \frac{\vec{C}^{p+1} - \vec{C}^{p+1/2}}{h_x} + \hat{L}_z \vec{C}^{p+1} = 0 \tag{34}$$

The region of simulation is $0 < x < L_x = (1000 - 2000)$ km, $0 < y < L_y = (2000 - 3000)$ km, $0 < z < L_z = 80 - 90$ km. The numerical scheme (34) is absolutely stable. Here $h_x$ is the step along $OX$ axis, $x_p = p\,h_x$, $p = 0, 1, 2, \dots$. This step has been chosen from the conditions of the simulation results independence on the diminishing $h_x$.

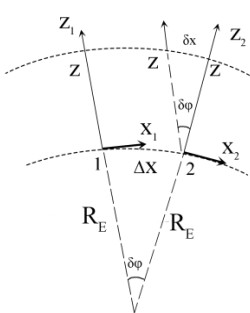

390

**Figure 2. The rotation of the local Cartesian coordinate frame at each step along the Earth's surface $h_x$ on a small angle $\delta\varphi$ $\approx$ $\approx$ $\Delta x/R_E$, radians, while $\Delta x=h_x$. The following strong inequalities are valid $h_x \ll L_z \ll R_E$. At the Earth's surface $z = 0$.**

On the simulation at each step along $OX$ axis, the correction on the Earth's curvature has been inserted in adiabatic manner applying the rotation of the local coordinate frame $XOZ$. Because the step along $x$ is small $h_x \sim 1$ km $\ll L_z$, this correction of the $\vec{C}$ results in the multiplier $exp(-ik_0\cdot\delta x)$, where $\delta x = z\cdot(h_x/R_E)$, $R_E \gg L_z$ is the Earth's radius (see Fig. 2 and the caption to this figure). At the distances $x \leq 1000$ km, the simulation results do not depend on the insertion of this correction, whereas at higher distances some quantitative difference occurs: the VLF beam propagates more closely to the upper boundary of the waveguide.

## 3.5 VLF Waveguide Modes and Reflection from the VLF Waveguide Upper Effective Boundary

In general, our model needs the consideration of the waveguide modes excitations by a current source such as dipole-like VLF radio source and lightning discharge. Then, the reflection of the waves incident on the upper boundary ($z=L_z$) of the effective WGEI can be considered. There will be possible to demonstrate that this structure has indeed good enough waveguiding properties. Then, in the model described in the present paper, the VLF beam is postulated already on the input of the system. To understand, how such a beam is excited by the, say, dipole antenna near the lower boundary $z=0$ of the WGEI, the formation of the beam structure based on the mode presentation should be searched. Then the conditions of the radiation (absence of ingoing waves) (12) can be used as the boundary conditions for the VLF beam radiated to the upper ionosphere/magnetosphere. Due to a relatively large scale of the inhomogeneity in this region, the complex geometrical optics (Rapoport et al., 2014) would be quite suitable for modeling a beam propagation, even accounting for the wave dispersion in magnetized plasma. The proper effective boundary condition, similarly to (Rapoport et al., 2014) would allow to make relatively accurate matching between the regions, described by the full wave electromagnetic approach with Maxwell's equations and complex geometrical optics (FWEM-CGO approach). All of these materials are not included in this paper, but will be delivered in the two future papers. The first paper will be dedicated to the modeling VLF waves propagating in WGEI based on the field expansion as a set of eigenmodes of the waveguide (the mode presentation

approach). The second paper will deal with the leakage of the VLF beam from the WGEI into the upper ionosphere and
magnetosphere and the VLF beam propagation in these media. Here we describe only one result, which concerns the mode
excitation in the WGEI, because this result is principally important for the justification of TIMEB method. It was shown that
more than five lowest modes of the WGEI are strongly localized in the atmosphere-lower ionosphere. Their longitudinal
wavenumbers are close to the corresponding wavenumbers of EM-waves in the atmosphere. This fact convinced that the
TIMEB method can be applied to the propagation of the VLF electromagnetic waves in the WGEI.

## 4. Modeling Results

The dependencies of the permittivity components $\varepsilon_1$, $\varepsilon_3$, $\varepsilon_h$ in the coordinate frame associated with the geomagnetic field $\vec{H}_0$
are given in Fig. 3. The parameters of the ionosphere used for modeling are taken from (Al'pert 1972; Alperovich and
Fedorov 2007; Kelley 2005; Schunk and Nagy 2010; Jursa 1985). The typical results of simulations are presented in Fig. 4.
The parameters of the ionosphere correspond to Fig. 3. The angle $\theta$ (Fig. 1) is 45°. The VLF frequency is $\omega = 10^5$ s$^{-1}$, $f =$
$\omega/2\pi \approx 15.9$ kHz. The Earth's surface is assumed as ideally conductive at the level $z = 0$. The values of EM-field are given in
absolute units. The magnetic field is measured in Oersteds (Oe), or Gauss (Gs), 1 Gs = $10^{-4}$ T, whereas the electric field is
also in Gs, 1 Gs = 300 V/cm.
Note that in the absolute (Gaussian) units the magnitudes of the magnetic field component $|H_y|$ are the same as ones of the
electric field component $|E_z|$ in the atmosphere region where the permittivity is $\varepsilon \approx 1$. Below in the Fig. 4 caption, the
correspondence between the absolute units and practical SI units is given.
It is seen that the absolute values of the permittivity components increase sharply above $z = 75$ km. The behavior of
the permittivity components is step-like, as seen from Fig. 3a. Therefore, the results of simulations are tolerant to the choice
of the upper wall position of the Earth's surface–ionosphere waveguide. The computed components of the tensor impedance
at $z = 85$ km are: $Z_{011} = 0.087 + i0.097$, $Z_{021} = 0.085 + i0.063$, $Z_{012} = -0.083 - i0.094$, $Z_{022} = 0.093 + i0.098$. So, a condition
$|Z_{0\alpha\beta}| \leq 0.15$ is satisfied there, which is necessary for the applicability of the boundary conditions (25). The maximum value
of the $H_y$ component is 0.1 Oe = $10^{-5}$ T in Fig. 4a for the initial VLF beam at $x = 0$. This corresponds to the value of $E_z$
component of 0.1 Gs = 30 V/cm. At the distance $x = 1000$ km the magnitudes of the magnetic field $H_y$ are of about $3 \cdot 10^{-5}$
Oe = 3 nT, whereas the electric field $E_y$ is of $3 \cdot 10^{-6}$ Gs $\approx 1$ mV/cm.

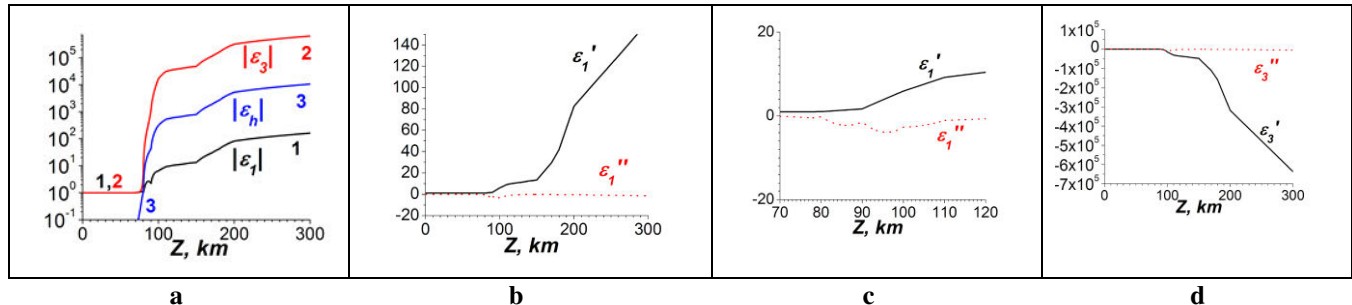

a                        b                        c                        d

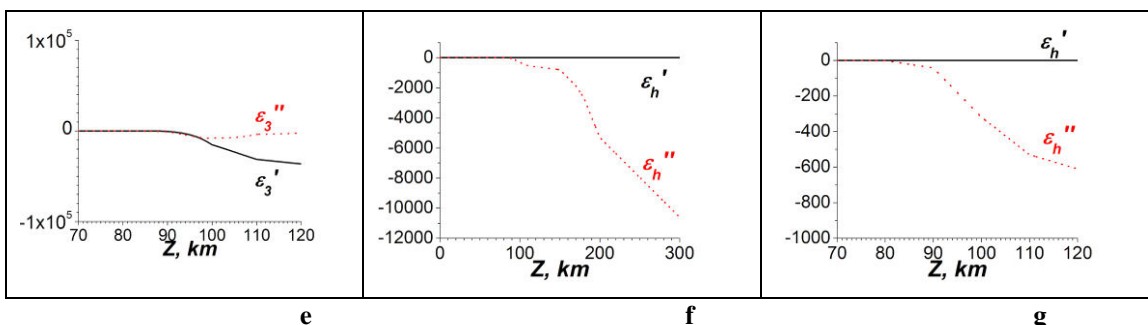


**Figure 3. (a) The vertical dependencies of the modules of components of the permittivity in the frame associated with the**
**geomagnetic field $|\varepsilon_l|$, $|\varepsilon_3|$, $|\varepsilon_h|$,, curves 1, 2, 3 correspondingly. (b) – (g) The real (corresponding lines with the values denoted by**
**one prime) and imaginary (corresponding lines with the values denoted by two primes) parts of the components $\varepsilon_l$, $\varepsilon_3$, $\varepsilon_h$, general**
**and detailed views.**

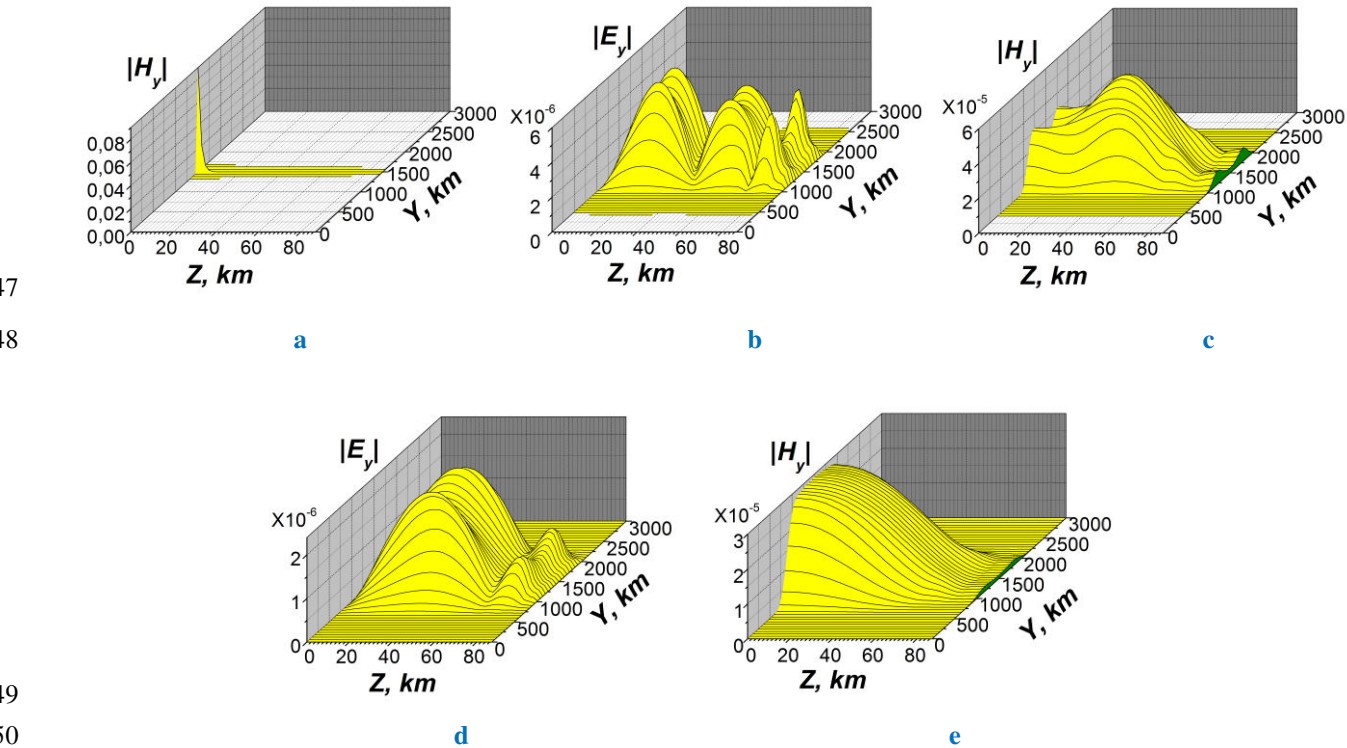


**a**                                 **b**                             **c**


**d**                             **e**

**Figure 4. Part a) is the initial distribution of $|H_y|$ at $x = 0$. Parts b), c) are $|E_y|$ and $|H_y|$ at $x = 600$ km. Parts d), e) are $|E_y|$ and $|H_y|$ at**
**$x = 1000$ km. For the electric field maximum value (Fig. d) is $3 \cdot 10^{-6}$ Gs $\approx 1$ mV/cm, for the magnetic field maximum value (Fig. e) is**
**$3 \cdot 10^{-5}$ Gs $\approx 3$ nT. At the altitudes $z < 75$ km, $|E_z| \approx |H_y|$. $\omega = 1 \cdot 10^5 \, s^{-1}$; $\theta = 45°$**
The wave beams are localized within the WGEI, $0 < z < 75$ km, mainly in the regions with the isotropic permittivity (see
Fig. 4b-e). The mutual transformations of the beams of different polarizations occur near the waveguide upper boundary
to the anisotropy of the ionosphere within the thin layer 75 km $< z <$ 85 km (Fig. 4b, d). These transformations depend on
the permittivity component values of the ionosphere at the altitude $z > 80$ km and on the components of the tensor
impedance. Therefore, the measurements of the phase and amplitude modulations of different EM components near the
Earth's surface can provide information on the properties of the lower and middle ionosphere.
In accordance with boundary condition (32), we suppose that when entering the system (at $X = 0$), only one of the two
polarization modes is excited, namely, the TM mode, i.e. at $X = 0$, $H_y \neq 0$; $E_y = 0$ (Fig. 4a). Upon further propagation of the
beam with such boundary conditions at the entrance to the system in a homogeneous isotropic waveguide, the property of the
electromagnetic field described by the relation $H_y \neq 0$; $E_y = 0$ will remain valid. The qualitative effect due to the presence
of gyrotropy (a) in a thin bulk layer near the upper boundary of WGEI and (b) in the upper boundary condition with complex
gyrotropic and anisotropic impedance is as follows. During beam propagation in the WGEI, the TE polarization mode with
the corresponding field components, including $E_y$, is also excited. This effect is illustrated in Figs. 4 b, d.
The magnitude of the $E_y$ component depends on the values of the electron concentration at the altitudes $z = 75 - 100$ km. In
Fig. 5a, b the different dependencies of the electron concentration $n(z)$ are shown (see solid (1), dash (2) and dot (3) lines).
The corresponding dependencies of the component absolute values of the permittivity are shown in Figs. 4c and 4d.

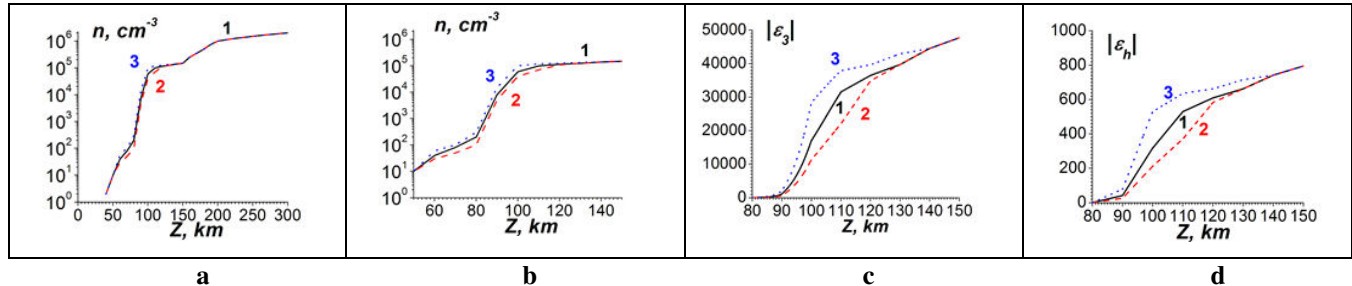

**a**               **b**               **c**               **d**

**Figure 5. Different profiles of the electron concentration $n(z)$ used in simulations: solid, dash, and dot lines correspond to**
**undisturbed, decreased and increased concentrations, respectively. (a) the detailed view; (b) general view; (c) and (d) the**
**permittivity $|\varepsilon_3|$ and $|\varepsilon_h|$ modules.**

The distributions of $|E_y|$, $|H_y|$ at $x = 1000$ km are given in Fig. 6. Results in Figs. 6a and 6b correspond to the solid (1) curve
$n(z)$ in Fig. 5; Figs. 6c and 6d correspond to the dash (2) curve; Figs. 6e and 6f correspond to the dot (3) curve in Fig. 5. The
initial $H_y$ beam is the same and is given in Fig. 4a. The values of the tensor impedance for these three cases are presented in
Table 1.

**Table 1. The values of tensor impedance components corresponded to the data shown in Fig. 5.**

| Component of the tensor impedance | $Z_{011}$ | $Z_{021}$ | $Z_{012}$ | $Z_{022}$ |
|---|---|---|---|---|
| Undisturbed concentration (curves 1 in Fig. 5) | 0.088 + i0.098 | 0.085 + i0.063 | −0.083 – i0.094 | 0.093 + i0.098 |
| Decreased concentration | 0.114 + i0.127 | 0.107 + i0.079 | −0.105 – i0.127 | 0.125 + i0.125 |

| | | | | |
|---|---|---|---|---|
| (curves 2 in Fig. 5) | | | | |
| Increased concentration (curves 3 in Fig. 5) | 0.067 + i0.0715 | 0.061 + i0.051 | −0.060 – i0.070 | 0.069 + i0.072 |


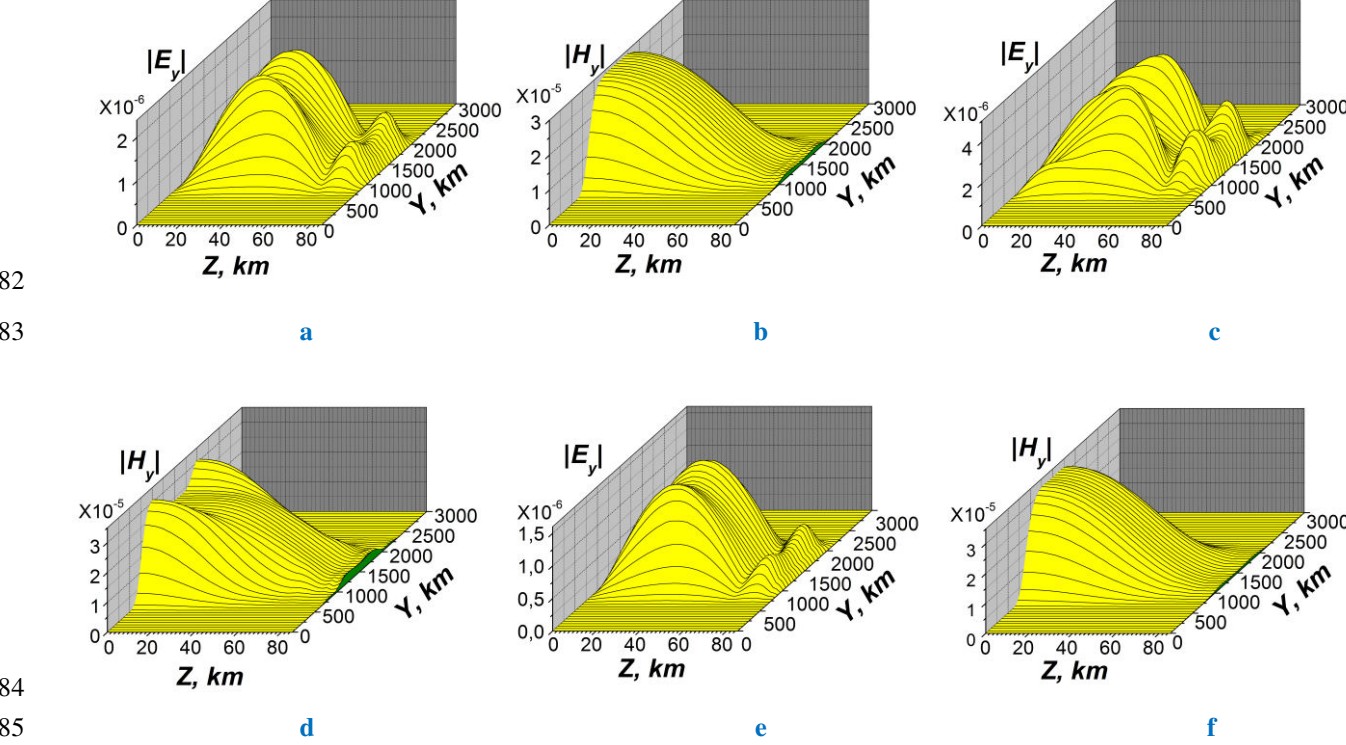


a          b          c


485          d          e          f

**Figure 6. Parts a), c), e) are dependencies of $|E_y|$, parts b), d), f) are dependencies of $|H_y|$ at $x$ = 1000 km; $\omega = 1. \cdot 10^5 s^{-1}$; $\theta = 45°$.**
**The initial beams are the same as in Fig. 4, a). Parts a), b) correspond to the solid (1) curves in Fig. 5; parts c), d) are for the dash**
**(2) curves; parts e), f) correspond to the dot (3) curves there.**

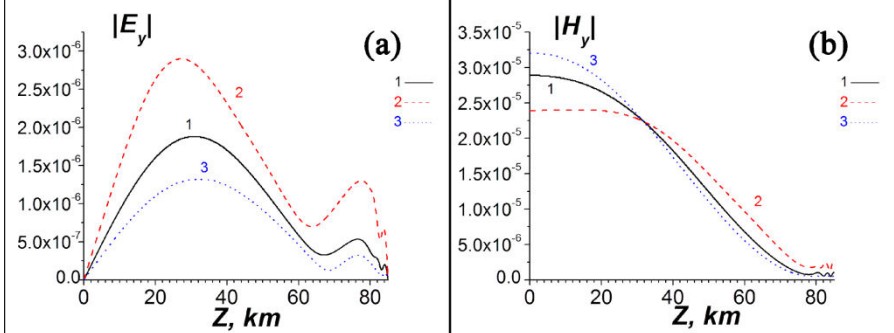


**Figure 7. The dependencies of EM components on the altitude $z$ in the center of the waveguide, $y$ = 1500 km, for the different**
**profiles of the electron concentration. The solid (1), dash (2), and dot (3) curves correspond to the different profiles of the electron**
**concentration in Fig. 5, a), b), the same kinds of curves; $\omega = 1. \cdot 10^5 s^{-1}$; $\theta = 45°$**
The distributions of $|E_y|$, $|H_y|$ on $z$ at $x$ = 1000 km in the center of the waveguide, $y$ = 1500 km, are given in Fig. 7. These
simulations show that the change in the complex tensors of both volume dielectric permittivity and impedance at the lower
and upper boundaries of effective WGEI influence remarkably on the VLF losses. The modulation of the electron
concentration at the altitudes above $z$= 120 km affects the excitation of $E_y$ component within the waveguide rather weakly.

## 5. Influence of the parameters of WGEI on the polarization transformation and losses of the propagating VLF waves

An important effect of the gyrotropy and anisotropy is the corresponding transformation of the field polarization

during the propagation in the WGEI, which is absent in the ideal metal planar waveguide without gyrotropy and anisotropy.
We will show that this effect is quite sensitive to the carrier frequency of the beam, propagating in the WGEI, inclination of
the geomagnetic field and perturbations in the electron concentration, which can vary under the influences of the powerful
sources placed "below", "above" and "inside" the ionosphere. In the real WGEI, the anisotropy and gyrotropy are connected
with the volume effect and effective surface tensor impedances at the lower and upper surfaces of the effective WGEI where
$z=0$ and $z= L_z$ (Fig. 1). For the corresponding transformation of the field polarization determination, we introduce the
characteristic polarization relation $|E_y / H_y|(z; y = L_y / 2; x = x_0)$, taken at the central plane of the beam ($y=L_y$) at the
characteristic distance ($x=x_0$) from the beam input/VLF transmitter. The choice of the characteristic polarization parameter
($|E_y/H_y|$) and its dependence on the vertical coordinate ($z$) is justified by conditions (1) – (3). (1) The WGEI is similar to the
ideal planar metallized waveguide because, first, the tensor $\hat{\varepsilon}$ is different from the isotropic $\hat{I}$ only in the relatively small
upper part of the WGEI in the altitude range from 75-80 to 85 km (see Fig. 1). Second, both the Earth and ionosphere
conductivity are quite high and corresponding impedances are quite low. In particular, the elements of the effective tensor
impedance at the upper boundary of WGEI are small, $|Z0\alpha\beta| \leq 0.1$ (see, for example, Table 1). (2) Respectively, the carrier
modes of the VLF beam are close to the modes of the ideal metallized planar waveguide. These modes are subdivided into
the sets of uncoupled (Ex,Hy,Ez) and (Hx,Ey,Hz) modes. The detailed search of the propagation of the separate eigenmodes
of the WGEI is not a goal of this paper, and respectively, will be the subject of the separate paper. (3) Because we have
adopted for the initial beam(s) the input boundary conditions in the form (32) (with $H_y \neq 0$, $E_y = 0$), the above mentioned
value $|E_y / H_y|(z; y = L_y / 2; x = x_0)$ characterizes the mode coupling and corresponding transformation of the polarization
at the distance $x_0$ from the beam input due to the presence of the volume and surface gyrotropy and anisotropy in the real
WGEI. The results presented below are obtained for $x_0$ =1000 km, that is, by the order of value, a typical distance, for
example, between the VLF transmitter and receiver of the European VLF/LF radio network (Biagi et al. 2015). Other
parameter characterizing the propagation of the beam in the WGEI, the effective total loss parameter is $|H_{ymax}(x=x_0)/$
$H_{ymax}(x=0)|$. Note that this parameter characterizes both dissipative and diffraction losses. The last are connected with beam
spreading in the transverse ($y$) direction during the propagation in the WGEI.

In Fig. 8 the polarization and loss characteristics dependencies on both the carrier beam frequency and the angle $\theta$ between the geomagnetic field and the vertical directions (see Fig. 1) are shown.

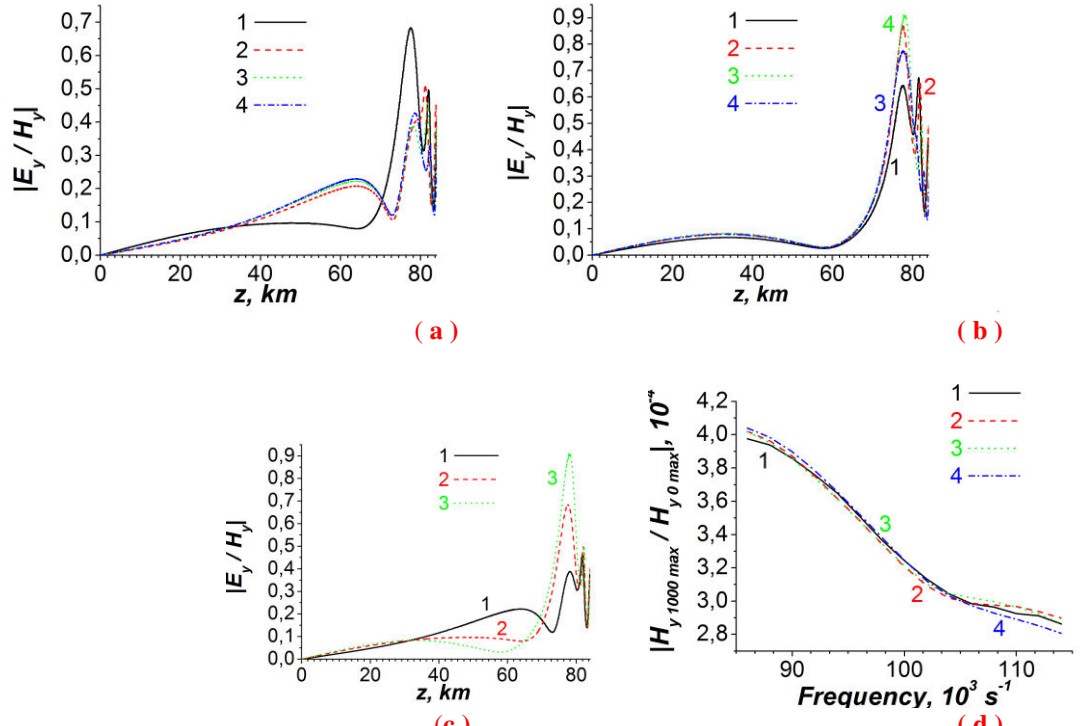

( a )                                      ( b )

( c )                                      ( d )

**Figure 8. Characteristics of the polarization transformation parameter $|E_y/H_y|$ (a-c) and effective coefficient of the total losses at the distance $x_0=1000$ km from the beam input ( d ); corresponding altitude dependence of the electron concentration is shown in Fig. 5b line (1); (a, b) and ( d ) - dependences of the polarization parameter (a, b) and total losses (d) on the vertical coordinate for different angles $\theta$ , respectively; black (1), red (2), green (3) and blue (4) curves in Figs. a, b and d correspond to $\theta$ equal to $5°, 30°, 45°$ and $60°$, respectively; (a) and (b) correspond to frequencies $\omega = 0.86 \cdot 10^5 s^{-1}$ and $\omega = 1.14 \cdot 10^5 s^{-1}$, respectively; ( c ) – dependence of the polarization parameter on the vertical coordinate for the different frequencies; black (1), red (2) and green (3) lines correspond to the frequencies $0.86 \cdot 10^5$, $1 \cdot 10^5$ and $1.14 \cdot 10^5$ s⁻¹, respectively and $\theta = 45°$ .**

In Fig. 8a-c the altitude dependence of the polarization parameter |Ey/Hy| exhibits two main maxima in the WGEI. The first one lies in the gyrotropic region above 70 km, while the second one in the isotropic region of the WGEI. As it is seen from Fig. a, b, the value of the (larger) second maximum increases, while the value of the first maximum decreases and its position shifts to the lower altitudes with increasing frequency. At the higher frequency ( $\omega = 1.14 \cdot 10^5 c^{-1}$ ), the larger maximum of the polarization parameter corresponds to the intermediate value of the angle $\theta = 45°$ (Fig. 8 b); for the lower frequency ( $\omega = 0.86 \cdot 10^5 c^{-1}$ ), the largest value of the first (higher) maximum corresponds to the almost vertical direction of the geomagnetic field ( $\theta = 5°$ , Fig. 8 a). For the intermediate value of the angle ( $\theta = 45°$ ), the largest value of the main maximum corresponds to the higher frequency ( $\omega = 1.14 \cdot 10^5 c^{-1}$ ) in the considered frequency range (Fig. 8 c). The total losses increase monotonically with increasing frequency and depend weakly on the value of $\theta$ (Fig. 8 d).

To model the effect of increasing and decreasing the electron concentration $n_e$ in the lower ionosphere on the

polarization parameter, we have used the following parameterization for the $n_e$ change $\Delta n_e = n_e(z) - n_{0e}(z)$ of the electron
concentration, where $n_{0e}(z)$ is the unperturbed altitude distribution of the electron concentration:
$$\Delta n_e(z) = n_{0e}(z)\Phi(z); \quad \Phi(z) = [F(z)] - \frac{(z-z_2)^2}{\Delta z_{12}^2}[F(z_1)] - \frac{(z-z_1)^2}{\Delta z_{12}^2}[F(z_2)]; \quad F(z) = f_{00} \cdot ch^{-2}\{[z - (\frac{z_1 + z_2}{2})]/\Delta z\} \quad (35)$$

In Eqs. (35), $\Delta z_{12} \equiv z_2 - z_1$; $\Delta z$ is the effective width of the electron concentration perturbation altitude distribution. The
perturbation $\Delta n_e$ is concentrated in the range of altitudes $z_1 \le z \le z_2$ and is equal to zero outside this region,
$\Delta n_e(z_1) = \Delta n_e(z_2) = 0$, while $\Phi(z_1) = \Phi(z_2) = 0$.

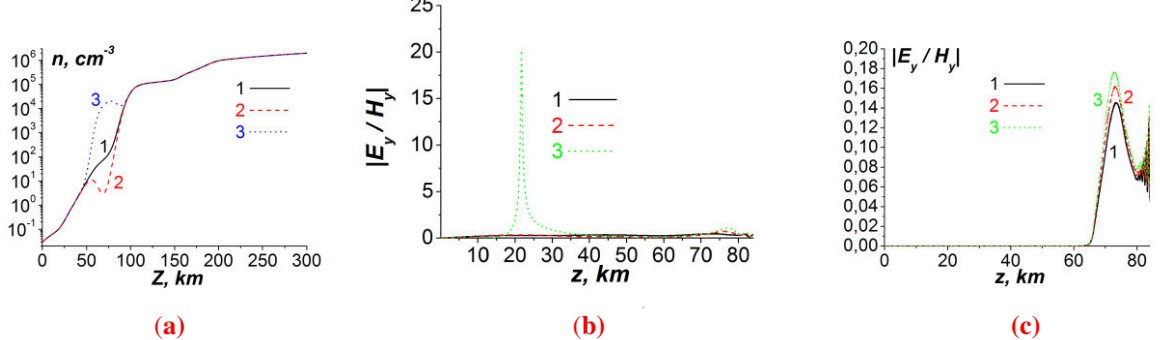

**(a)**           **(b)**           **(c)**

**Figure 9. (a) Decreased and increased electron concentration (line 2, red color) and (line 3, blue) correspond to $f_{00} = -1.25$ and**
**$f_{00} = 250$, respectively, relative to reference concentration (line 1, black) with parametrization conditions (see Eqs. (35)): $z_1 = 50$**
**km, $z_2 = 90$ km, $\Delta z = 20$ km; (b) and (c) are the polarization parameter $|E_y/H_y|$ altitude distribution, for decreased and increased**
**electron concentration, respectively. In (b) and (c) lines 1, 2 and 3 correspond to $\omega$ values $0.86 \cdot 10^5 s^{-1}$, $1 \cdot 10^5 s^{-1}$ and**
**$1.14 \cdot 10^5 s^{-1}$, respectively. Angle $\theta$ is equal to $45°$.**

The change in the concentration in the lower ionosphere causes rather nontrivial effect on the parameter of the

polarization transformation $|E_y/H_y|$, Fig. 9 a-c. Note that either increase or decrease in the ionosphere plasma concentration
have been reported as a result of seismogenic phenomena, tsunamis, particle precipitation in the ionosphere due to wave-
particle interaction in the radiation belts (Pulinets et al. 2005; Shinagawa et al. 2013; Arnoldy et al. 1989; Glukhov et al.
1992; Tolstoy et al. 1986) etc. The changes in the $|E_y/H_y|$ due to increase or decrease in electron concentration vary by
absolute values from dozens to thousands percent, as it is seen from the comparison between Figs. 9b, c (lines 3) and Fig. 8c
(line 3), which corresponds to the unperturbed distribution of the ionospheric electron concentration (see also lines 1 in Figs.
5b and 9a). It is even more interesting that in the case of decreasing (Fig. 9 a, curve 2 ) electron concentration, the main
maximum of $|E_y/H_y|$ aoppears in the lower atmosphere (at the altitude around 20 km, Fig. 9 b, curve 3, which corresponds to
$\omega \blacksquare 1.14 \bullet 10^5 c^{-1}$ ). In the case of increasing electron concentration (Fig. 9 a, curve 3) the main maximum of $|E_y/H_y|$ appears
near the E region of the ionosphere (at the altitude around 77 km, Fig. 9 c). The secondary maximum, which is placed, in the
absence of the perturbation of the electron concentration, in the lower atmosphere (Fig. 8 c, curves 2, 3), or
mesosphere/ionosphere D region ((Fig. 8 c, curve 1), practically disappears or just is not seen in the present scale, in the case
under consideration (Fig. 9 c, curves 1-3).

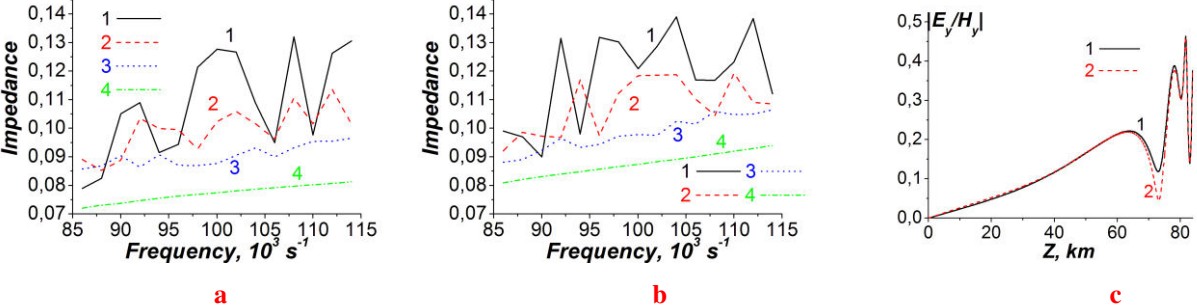


**a**                        **b**                   **c**

**Figure 10. (a, b) Frequency dependencies of the real (a) and imaginary (b) parts of the effective tensor impedance $Z_{011}$ component**
**at the upper boundary ($z=L_z$, see Fig. 1) of the WGEI. Lines 1 (black), 2 (red), 3 (blue) and 4 (green) correspond to $\theta = 5°, 30°, 45°$**
**and $60°$ degrees respectively; ( c ) polarization parameter $|E_y/H_y|$ altitude dependency at the frequency $\omega = 0.86 \cdot 10^5$ s$^{-1}$ and angle**
**$\theta = 45°$ for the isotropic surface impedance $Z_0$ at the lower surface of the WGEI equal to $10^{-4}$. Earth conductivity $\sigma$ equal to**
**$10^9$ s$^{-1}$, line 1 and $Z_0 = 10^{-2}$ ( $\sigma = 10^7 s^{-1}$ ), line 2.**

In Fig. 10, the real (a) and imaginary (b) parts of the surface impedance at the upper boundary of the WGEI have a
quasi-periodical character with the amplitude of oscillations occurring around the effective average values (not shown
explicitly in Figs. 10 a, b), which decreases with increasing the angle $\theta$. The average $Re(Z_{011})$ and $Im(Z_{011})$ values in
general, decrease with increasing angle $\theta$ (see Fig. 10a, b). The average values of $Re(Z_{011})$ at $\theta = 5°, 30°, 45°$ and $60°$ (lines 1-
4 in Fig. 10a) and $Im(Z_{011})$ at $\theta = 45°$ and $60°$ (curves 3, 4 in Fig. 10b) increase with increasing frequency in the frequency
range $(0.86-1.14) \cdot 10^5$ s$^{-1}$. The average $Im(Z_{011})$ value at $\theta = 5°$ and $30°$ changes in the frequency range $(0.86-1.14) \cdot 10^5$ s$^{-1}$
non-monotonically with the maximum at $(1-1.1) \cdot 10^5$ s$^{-1}$. The value of finite impedance at the lower Earth-atmosphere
boundary of the WGEI influences on the polarization transformation parameter minimum near the E region of the ionosphere
(lines 1, 2 in Fig. 10c). The decrease of surface impedance $Z_0$ at the lower boundary Earth-atmosphere of the WGEI by two
orders of magnitude produces the 100% increase of the corresponding $|E_y/H_y|$ minimum at $Z \sim 75$ km (Fig. 10 c).

## 6. Discussion

The observations presented in (Rozhnoi et al., 2015), shows a possibility for seismogenic increasing losses of VLF
waves in the WGEI (Fig. 11; see details in (Rozhnoi et al. 2015)). We discuss the qualitative correspondence of our results to
these experimental data.

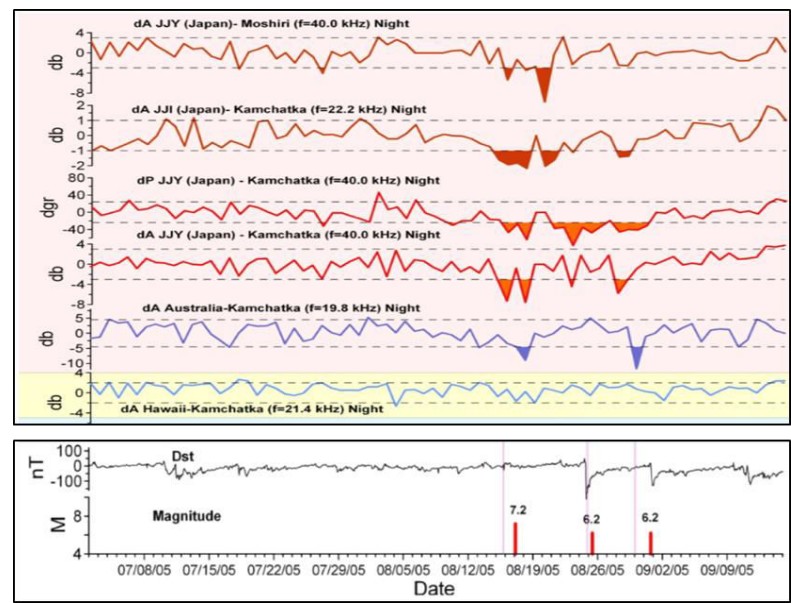

**Figure 11.** Averaged residual VLF/LF signals from ground-based observations at the wave paths: JJY-Moshiri, JJI-Kamchatka, JJY-Kamchatka, NWC-Kamchatka, and NPM-Kamchatka. Horizontal dotted lines show 2σ level. The color filled zones highlight values exceeding the -2σ level. In panel below Dst variations and earthquakes magnitude values are shown (from Rozhnoi et al., 2015, see their Fig. 1 but not including the DEMETER data; the work of Rozhnoi et al. (2015) is licensed under a Creative Commons Attribution 4.0 International License (CC BY 4.0)). See other details in (Rozhnoi et al., 2015).

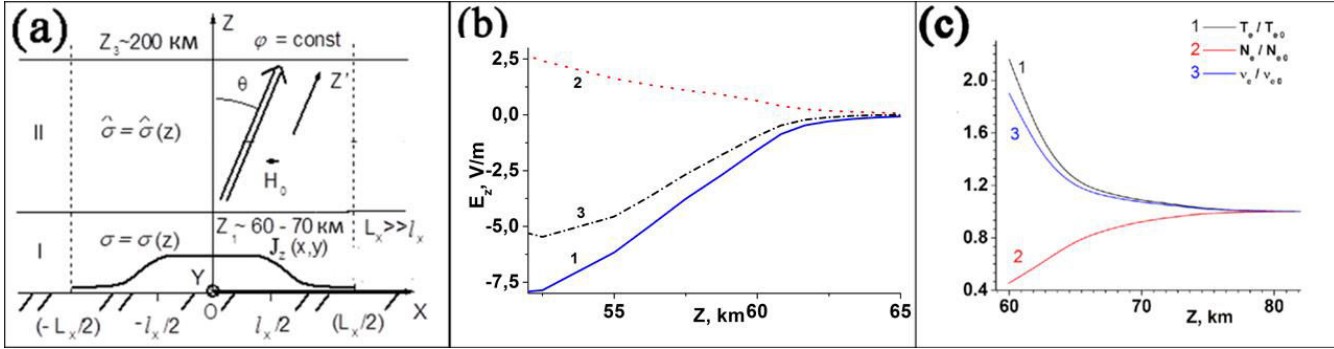

**Figure 12.** Modification of the ionosphere by electric field of seismogenic origin based on the theoretical model (Rapoport et al. 2006). (a) Geometry of the model for the determination of the electric field excited by seismogenic current source $J_z(x, y)$ and penetrated into the ionosphere with isotropic (I) and anisotropic (II) regions of the system "atmosphere-ionosphere". (b) Electric field in the mesosphere in presence of the seismogenic current sources only in the mesosphere (1); in the lower atmosphere (2); both in the mesosphere and in the lower atmosphere (3). ( c ) Relative perturbations caused by seismogenic electric field, normalized on the corresponding steady-state values in the absence of perturbing electric field, denoted by the index "0", of electron temperature ($T_e / T_{e0}$), electron concentration ($N_e / N_{e0}$), and electron collision frequency ($\nu_e / \nu_{e0}$).

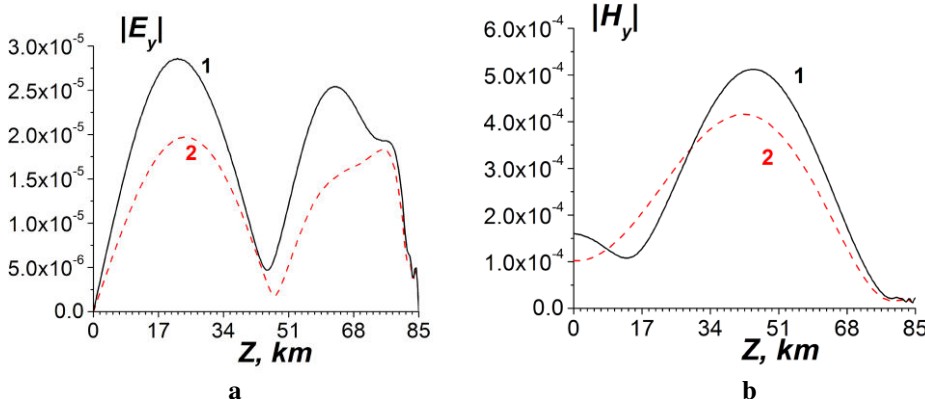

a                              b

**Figure 13.** Altitude distributions of the normalized tangential (*y*) electric (a) and magnetic (b) VLF beam field components in the central plane of the transverse beam distribution (*y=0*) at the distance *x=1000* km from the input of the system. Line 1 in (a, b) corresponds to the presence of the mesospheric electric current source only with relatively small value of $N_e$ and large $v_e$. Line 2 corresponds to the presence of both mesospheric and near-ground seismogenic electric current sources with relatively large value of $N_e$ and small $v_e$. Lines 1 and 2 in (a, b) correspond qualitatively to the lines 1 and 3, respectively, in Fig. 12b.

$\omega = 1.5 \cdot 10^5 s^{-1}; \ \theta = 45°$ .

The modification of the ionosphere due to electric field excited by the near-ground seismogenic current source has

been taken into account.  In the model (Rapoport et al., 2006), the presence of the mesospheric current source, which
followed from the observations  (Martynenko et al., 2001; Meek et al., 2004; Bragin, 1974) is also taken into account. It is
assumed that the mesospheric current has only the Z-component and is positive, which means that it is directed vertically
downward, as is the fair-weather current (curve 1, Fig. 12b). Then suppose that the surface seismogenic current is directed in
the same way as the mesospheric current. We first consider the case when the mesospheric current is zero and only the
corresponding seismogenic current is present near the earth. The corresponding mesospheric electric field under the
condition of a given potential difference between the Earth and the ionosphere (curve 2, Fig. 12 b) is directed opposite to
those excited by the corresponding mesospheric current (curve 1, Fig. 12 b). As a result, in the presence of both mesospheric
and a seismogenic surface current, the total mesospheric electric field (curve 3, Fig. 12 b) is smaller in absolute value than in
the presence of only a mesospheric current (curve 1, Fig. 12 b). It has been shown by Rapoport et al., (2006) that the
decrement of losses $|k''|$ for VLF waves in the WGEI is proportional to $|k''| \sim |\varepsilon''| \sim N_e / v_e$. $N_e$ and $v_e$ decrease and
increase, respectively, due to the appearance of a seismogenic surface electric current, in addition to the mesospheric current
(curve 3, Fig. 12, b). As a result, the losses increase compared with the case when the seismogenic current is absent and the
electric field has a larger absolute value (curve 1, Fig. 12). The increase in losses in the VLF beam, shown in Fig. 13
(compare curves 2 and 1 in Figs. 13 a, b) corresponds to an increase in losses with an increase in the absolute value of the
imaginary part of the dielectric constant, when a near-surface seismogenic current source appears (curve 3 in Fig. 12 b), in
addition to the existing mesospheric current source (curve 2 in Fig. 12 b). This seismogenic increase in losses corresponds
qualitatively to the results, presented in (Rozhnoi et al. 2015).

The TIMEB is a new method of modeling characteristics of the WGEI. The results of beam propagation in WGEI modeling presented above include the range of altitudes inside the WGEI (see Figs. 4-7). Nevertheless, the TIMEB method described by Eqs. (15)-(19), (22-24), (27), (30) and allows to determine all field components in the  range  of altitudes $0 \leq z \leq L_{max}$, where $L_{max} = 300$ km. The structure and behavior of these eigenmodes in the WGEI and leakage waves will be a subject of separate papers. We present here only the final qualitative result of the simulations.  In the range  $L_z \leq z \leq L_{max}$, where $L_z = 85$ km is the upper boundary of the effective WGEI, all field components are (1) at least one order of altitude less than the corresponding maximal field value in the WGEI and (2) field components have the oscillating character along $z$ coordinate and describe the modes, leaking from the WGEI.

Let us make a note also on the dependences of the field components in the WGEI on the vertical coordinate ($z$). The initial distribution of the electromagnetic field with altitude $z$ (Fig. 4a) is determined by the boundary conditions of the beam (see Eq. (32)). The field component includes higher eigenmodes of the WGEI. The higher-order modes experienced quite large losses and practically disappear after beam propagation on 1000 km distance. This determines the change in altitude ($z$) and transverse ($y$) distributions of the beam field during propagation along the WGEI. In particular, at the distance $x$=600 km from the beam input (Figs. 4b, c), the few lowest modes of the WGEI along $z$ and $y$ coordinates still persist. At distance $x$=1000 km (Fig. 4d, c; Fig. 6e, f; and Fig. 7a, b), only the main mode persists in the z direction. Note, the described field structure correspond to real WGEI with losses. The gyrotropy and anisotropy causes the volume effects and surface impedance, in distinction to the ideal planar metallized waveguide with isotropic filling (Collin, 2001).

The closest approach of the direct investigation of the VLF electromagnetic field profile in the Earth-Ionosphere waveguide was a series of sounding rocket campaigns at mid- and high-latitudes at Wallops Is., VA and Siple Station in Antarctica (Kintner et al., 1983; Brittain et al., 1983; Siefring and Kelly, 1991; Arnoldy and Kintner, 1989), where single-axis E-field and three-axis B-field antennas, supplemented in some cases with in situ plasma density measurements were used to detect the far-field fixed-frequency VLF signals radiated by US Navy and Stanford ground transmitters.

The most comprehensive study of the WGEI will be provided by the ongoing NASA VIPER (VLF Trans-Ionospheric Propagation Experiment Rocket) project (PI J. W. Bonnell, UC Berkeley, NASA Grant 80NSSC18K0782). The VIPER sounding rocket campaign is consist of a summer nighttime launch during quiet magnetosphere conditions from Wallops Flight Facility, VA, collecting data through the D, E, and F regions of the ionosphere with a payload carrying the following instrumentation: 2D E- and 3D B-field waveforms, DC-1 kHz; 3D ELF to VLF waveforms, 100 Hz to 50 kHz; 1D wideband E-field measurement of plasma and upper hybrid lines, 100 kHz to 4 MHz; and Langmuir probe plasma density and ion gauge neutral density measurements at a sampling rate of at least tens of Hz. The VIPER project will fly a fully 3D EM field measurement, DC through VLF, and relevant plasma and neutral particle measurements at mid-latitudes through the radiation fields of (1) an existing VLF transmitter (the VLF transmitter Cutler with call  sign NAA, the very  low frequency (VLF) shore radio station at Cutler, Maine, USA, which transmits, at a frequency of 24 kHz an input power of up to 1.8 megawatts, see Fig. 11) and (2) naturally-occurring lightning transients through and above the leaky upper boundary of the WGEI supported by a vigorous theory and modeling effort in order to explore the vertical and horizontal profile of the

observed 3D electric and magnetic radiated fields of the VLF transmitter, and the profile related to the observed plasma and
neutral densities. The VLF wave's reflection, absorption, and transmission processes as a function of altitude will be
searched making use of the data on the vertical VLF E- and B-field profile.

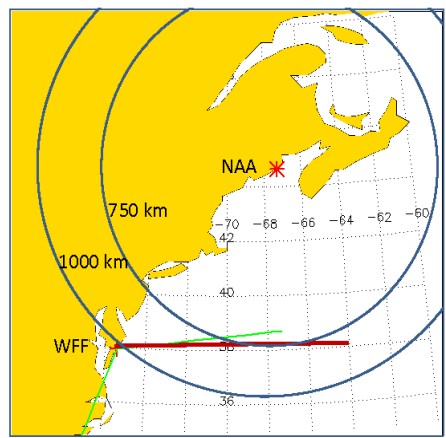

**Figure 14. Proposed VIPER Trajectory**

The aim of this experiment will be the investigation of the VLF beams launched by the near-ground source/VLF transmitter
with the known parameters and propagating both in the WGEI and leaking from WGEI into the upper ionosphere.
Characteristics of these beams will be compared with the theory proposed in the present paper and the theory on leakage of
the VLF beams from WGEI, which we will present in the next papers.

**Conclusions**
(1) We have developed the new and highly effective robust method of tensor impedance for the VLF electromagnetic
beam propagation in the inhomogeneous waveguiding media – the "tensor impedance method for modeling propagation of
electromagnetic beams (TIMEB)" in a multi-layered/inhomogeneous waveguide. The main differences/advantages of the
proposed tensor impedance method in comparison with the known method of the impedance recalculating, in particular
invariant imbedding methods (Shalashov and Gospodchikov, 2010; Kim and Kim, 2016) are the following: (i) our method is
a direct method of the recalculation of tensor impedance, and the corresponding tensor impedance is determined analytically,
(see Eq. (22)); (ii) our method applied for the media without non-locality does not need a solution of integral equation(s), as
the invariant imbedding method; (iii) the proposed tensor impedance method does not need the revealing the forward and
reflected waves. Moreover, even the conditions of the radiation in Eq. (12) at the upper boundary $z=L_{max}$ is determined
through the total field components $H_{x,y}$ that makes the proposed procedure technically less cumbersome and practically more
convenient.

(2) The waveguide includes the region for the altitudes 0 < z < 80 - 90 km. The boundary conditions are the radiation conditions at $z = 300$ km, which can be recalculated to the lower altitudes as the tensor relations between the tangential components of the EM field. In another words, the tensor impedance conditions have been used at $z = 80 - 90$ km.

(3) The application of this method jointly with the previous results of the modification of the ionosphere by seismogenic electric field gives the results, which qualitatively are in agreement with the experimental data on the seismogenic increasing losses of VLF waves/beams propagating in the WGEI.

(4) The observable qualitative effect is mutual transformation of different polarizations of the electromagnetic field occur during the propagation. This transformation of the polarization depends on the electron concentration, i.e. the conductivity, of D- and E-layers of the ionosphere at the altitudes 75 – 120 km.

(5) Changes in complex tensors of both volume dielectric permittivity and impedances at the lower and upper boundaries of effective WGEI influence remarkably on the VLF losses in the WGEI.

(6) An influence is demonstrated on the parameters characterizing the propagation of the VLF beam in the WGEI, in particular, on the parameter of the transformation polarization $|E_y/H_y|$ and tensor impedance at the upper boundary of the effective WGEI, of the carrier beam frequency, inclination of the geomagnetic field and the perturbations in the altitude distribution of the electron concentration in the lower ionosphere

(i) The altitude dependence of the polarization parameter $|E_y/H_y|$ has two main maxima in the WGEI: the higher maximum is in the gyrotropic region above 70 km, while the other is in the isotropic region of the WGEI. The value of the (larger) second maximum increases, while the value of the first maximum decreases and its position shifts to the lower altitudes with increasing frequency. In the frequency range of $\omega = (0.86 - 1.14) \cdot 10^5 s^{-1}$, At the higher frequency, the larger maximum polarization parameter corresponds to the intermediate value of the angle $\theta = 45°$; for the lower frequency, the largest value of the first (higher) maximum corresponds to the nearly vertical direction of the geomagnetic field. The total losses increase monotonically with increasing frequency and depend weakly on the value of $\theta$ (Fig. 1).

(ii) The change in the concentration in the lower ionosphere causes rather nontrivial effect on the parameter of polarization transformation $|E_y/H_y|$. This effect does include the increase and decrease of the maximum value of the polarization transformation parameter $| E_y/H_y |$. The corresponding change of this parameter has large values from dozens to thousands percent. In the case of decreasing electron concentration, the main maximum of $|E_y/H_y|$ appears in the lower atmosphere at an altitude of around 20 km. In the case of increasing electron concentration, the main maximum of $| E_y/H_y|$ appears near the E region of the ionosphere (at the altitude around 77 km), while the secondary maximum practically disappears.

(iii) The real and imaginary parts of the surface impedance at the upper boundary of the WGEI have a quasi-periodical character with the amplitude of "oscillations" occurring around some effective average values decreases with increasing the angle $\theta$. Corresponding average values of $Re(Z_{11})$ and $Im(Z_{11})$, in general, decrease with

increasing angle $\theta$. Average values of $Re(Z_{11})$ for $\theta$ equal to $5°, 30°, 45°$ and $60°$ and $Im(Z_{11})$ corresponding to $\theta$
jioequal to $45°$ and $60°$, increase with increasing frequency in the considered frequency range $(0.86\text{-}1.14) \cdot 10^5$ s$^{-1}$.
The average value of $Im(Z_{11})$ corresponds to $\theta$ equal to $5°$ and $30°$, change in the frequency range $(0.86\text{-}1.14) \cdot 10^5$ s$^{-}$
$^1$ non-monotonically, having maximum values around frequency $(1\text{-}1.1) \cdot 10^5$ s$^{-1}$.
(iv) The value of finite impedance at the lower Earth-atmosphere boundary of the WGEI make quite observable
influence on the polarization transformation parameter minimum near the E region of the ionosphere. The decrease
of surface impedance $Z$ at the lower boundary Earth-atmosphere of the WGEI in two orders causes the increase of
the corresponding minimum value of $|E_y/H_y|$ in $\sim 100\%$.
(7) In the range $L_z \leq z \leq L_{max}$, where $L_z = 85$ km is the upper boundary of the effective WGEI, all field components are
(a) at least one order of altitude less than the corresponding maximal value in the WGEI, and (b) field components have the
oscillating character (along the z coordinate) and describes the modes, leaking from the WGEI. The detail consideration of
the electromagnetic waves leaking from the WGEI will be presented in the separate paper. The initial distribution of the
electromagnetic field with z (vertical direction) is determined by the initial conditions on the beam. This field includes
higher eigenmodes of the WGEI. The higher-order modes, in distinction to the lower ones, have quite large losses and
practically disappear after a beam propagation for 1000 km distance. This circumstance determines the change in altitude (z)
distribution of the field of the beam during its propagation along the WGEI. In particular, at the distance x = 600 km from
the beam input, the few lowest modes of the WGEI along z coordinates are still survived. Further, at $x$=1000 km, practically,
only the main mode in the z direction remains. This fact reflects in a minimum number of oscillations of the beam field
components along $z$ at a given value of $x$.
(8) The proposed propagation of VLF electromagnetic beams in the WGEI model and results will be useful to explore
the characteristics of these waves as an effective instrument for diagnostics of the influences on the ionosphere "from above"
in the system of Sun-Solar Wind-Magnetosphere-Ionosphere, "from below" from the most powerful meteorological,
seismogenic and other sources in the lower atmosphere and lithosphere/Earth, such as hurricanes, earthquakes, tsunamis, and
from inside the ionosphere by the strong thunderstorms with lightning discharges, and even from the far space by as gamma-
flashes, cosmic rays events.

AUTHOR CONTRIBUTION
YuR and VG – proposed the idea and concept of the paper, made analytical calculations, have written the initial version of
the paper and took part in the revision of the paper; VG developed the code and YuR took part in its verification; YuR, VG,
A. Grytsai and A. Culin made numerical modeling; YuR and A.Grytsai provided the administration of the project; VG,
YuR, A. Grytsai and Andrey Gulin prepared the Figures; AL and. A.Grytsai took part in the writing initial version of the
paper and the preparation of the revised version.
VF, AR, MS took part in the analysis of the data and preparation the data on VLF propagation in the ionosphere.

OA. JB and GM took part in developing the concept of the paper, and writing the initial version of the paper. OA, GM, VF, AG contributed to the preparation of the revised version of the paper.

All participants took part in the analysis of the results.

COMPETING INTERESTS

The authors declare that they have no conflict of interest.

DATA AVAILABILITY

The VLF–LF data (Fig. 11) are property of IPE and TUoS groups, and they are not publicly accessible. According to an agreement between all the participants, we cannot make the data openly accessible. Data can be provided under commercial conditions via direct request to rozhnoi@ifz.ru. The ionospheric data used for the modeling the electrodynamics characteristics of the VLF waves in the ionosphere are shown in part in Fig. 5 (namely, altitude distribution of electron concentration). The other data necessary for determination the components of tensor of dielectric permittivity and then the electrodynamics modeling in the accepted simple approximation of the three-component plasma-like ionosphere (including electron, one effective ion and one effective neutral components) and quasi-neutrality are mentioned in Sect. 3.1. The corresponding ionospheric data have been taken from the well-known published handbooks, referred in the paper (Al'pert 1972; Alperovich and Fedorov 2007; Kelley 2005; Schunk and Nagy 2010; Jursa 1985).

ACKNOWLEDGMENTS

This work was supported in part by Taras Shevchenko National University of Kyiv, project 19BF051-08. The work of OA and JB was supported by the NASA grant 80NSSC18k0782. The work of OA was partially supported by the NASA LWS grant 80NSSC20K0218. VF thanks for financial support the Royal Society, International Exchanges Scheme , Standard Programme (grants no. IE170301, R1\191114), the Natural Environment Research Council (NERC) (grant no. NE/P017061/1).

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

966

**Appendix: the matrix coefficients included into eq. (16)**

Here the expressions of the matrix coefficients are presented that are used in the matrix factorization to compute the tensor

impedance, see eq. (16).

$$
\hat{\alpha}_N^{(0)} = \begin{pmatrix} 1 + \dfrac{ih_z}{\Delta}(k_1 - \alpha_1\alpha_2 k_2); & \dfrac{ih_z}{\Delta}\alpha_2(k_2 - k_1) \\[3mm] \dfrac{ih_z}{\Delta}\alpha_1(k_1 - k_2); & 1 + \dfrac{ih_z}{\Delta}(k_2 - \alpha_1\alpha_2 k_1) \end{pmatrix}, \quad \hat{\alpha}_N^{(-)} = \begin{pmatrix} -1; & 0 \\ 0; & -1 \end{pmatrix}; \quad \Delta \equiv 1 - \alpha_1\alpha_2;
$$

$$
\hat{\alpha}_j^{(-)} = \begin{pmatrix} \left(\dfrac{\beta_{22}}{1 - \beta_{22}\dfrac{k_x^2}{k_0^2}}\right)_{j-1/2}; & -\left(\dfrac{\beta_{21}}{1 - \beta_{22}\dfrac{k_x^2}{k_0^2}}\right)_{j-1/2} + \dfrac{ik_x h_z}{2}\left(\dfrac{\beta_{23}}{1 - \beta_{22}\dfrac{k_x^2}{k_0^2}}\right)_{j-1} \\[8mm] -\left(\dfrac{\beta_{12}}{1 - \beta_{22}\dfrac{k_x^2}{k_0^2}}\right)_{j-1/2} + \dfrac{ik_x h_z}{2}\left(\dfrac{\beta_{32}}{1 - \beta_{22}\dfrac{k_x^2}{k_0^2}}\right)_{j-1}; & \left(\beta_{11} + \dfrac{k_x^2}{k_0^2}\dfrac{\beta_{12}\cdot\beta_{21}}{1 - \beta_{22}\dfrac{k_x^2}{k_0^2}}\right)_{j-1/2} - \dfrac{ik_x h_z}{2}\left(\beta_{13} + \dfrac{k_x^2}{k_0^2}\dfrac{\beta_{12}\cdot\beta_{23}}{1 - \beta_{22}\dfrac{k_x^2}{k_0^2}}\right)_{j-1} - \\[8mm] & -\dfrac{ik_x h_z}{2}\left(\beta_{31} + \dfrac{k_x^2}{k_0^2}\dfrac{\beta_{32}\cdot\beta_{21}}{1 - \beta_{22}\dfrac{k_x^2}{k_0^2}}\right)_{j} \end{pmatrix}
$$

$$
\hat{\alpha}_j^{(+)} = \begin{pmatrix} \left(\dfrac{\beta_{22}}{1 - \beta_{22}\dfrac{k_x^2}{k_0^2}}\right)_{j+1/2}; & -\left(\dfrac{\beta_{21}}{1 - \beta_{22}\dfrac{k_x^2}{k_0^2}}\right)_{j+1/2} - \dfrac{ik_x h_z}{2}\left(\dfrac{\beta_{23}}{1 - \beta_{22}\dfrac{k_x^2}{k_0^2}}\right)_{j+1} \\[8mm] -\left(\dfrac{\beta_{12}}{1 - \beta_{22}\dfrac{k_x^2}{k_0^2}}\right)_{j+1/2} - \dfrac{ik_x h_z}{2}\left(\dfrac{\beta_{32}}{1 - \beta_{22}\dfrac{k_x^2}{k_0^2}}\right)_{j+1}; & \left(\beta_{11} + \dfrac{k_x^2}{k_0^2}\dfrac{\beta_{12}\cdot\beta_{21}}{1 - \beta_{22}\dfrac{k_x^2}{k_0^2}}\right)_{j+1/2} + \dfrac{ik_x h_z}{2}\left(\beta_{13} + \dfrac{k_x^2}{k_0^2}\dfrac{\beta_{12}\cdot\beta_{23}}{1 - \beta_{22}\dfrac{k_x^2}{k_0^2}}\right)_{j+1} + \\[8mm] & +\dfrac{ik_x h_z}{2}\left(\beta_{31} + \dfrac{k_x^2}{k_0^2}\dfrac{\beta_{32}\cdot\beta_{21}}{1 - \beta_{22}\dfrac{k_x^2}{k_0^2}}\right)_{j} \end{pmatrix}
$$

$$\hat{\alpha}_j^{(0)} = \begin{pmatrix} -\left(\dfrac{\beta_{22}}{1-\beta_{22}\dfrac{k_x^{\,2}}{k_0^{\,2}}}\right)_{j-1/2} - \left(\dfrac{\beta_{22}}{1-\beta_{22}\dfrac{k_x^{\,2}}{k_0^{\,2}}}\right)_{j+1/2} + k_0^{\,2}h_z^{\,2}; & \left(\dfrac{\beta_{21}}{1-\beta_{22}\dfrac{k_x^{\,2}}{k_0^{\,2}}}\right)_{j-1/2} + \left(\dfrac{\beta_{21}}{1-\beta_{22}\dfrac{k_x^{\,2}}{k_0^{\,2}}}\right)_{j+1/2} \\[4ex] \left(\dfrac{\beta_{12}}{1-\beta_{22}\dfrac{k_x^{\,2}}{k_0^{\,2}}}\right)_{j-1/2} + \left(\dfrac{\beta_{12}}{1-\beta_{22}\dfrac{k_x^{\,2}}{k_0^{\,2}}}\right)_{j+1/2}; & -\left(\beta_{11}+\dfrac{k_x^{\,2}}{k_0^{\,2}}\dfrac{\beta_{12}\cdot\beta_{21}}{1-\beta_{22}\dfrac{k_x^{\,2}}{k_0^{\,2}}}\right)_{j-1/2} - \left(\beta_{11}+\dfrac{k_x^{\,2}}{k_0^{\,2}}\dfrac{\beta_{12}\cdot\beta_{21}}{1-\beta_{22}\dfrac{k_x^{\,2}}{k_0^{\,2}}}\right)_{j+1/2} + \\[4ex] & + k_0^{\,2}h_z^{\,2}\cdot\left(1-\beta_{33}\dfrac{k_x^{\,2}}{k_0^{\,2}} - \dfrac{k_x^{\,4}}{k_0^{\,4}}\dfrac{\beta_{23}\cdot\beta_{32}}{1-\beta_{22}\dfrac{k_x^{\,2}}{k_0^{\,2}}}\right)_j \end{pmatrix}.$$