# Peer review of "Model of Propagation of VLF Beams in the Waveguide Earth-Ionosphere. Principles of Tensor Impedance Method in"

_Annales Geophysicae, 2019_

## Referee Comment (RC1) · Pier Francesco Biagi (Referee) · 15 May 2019

**Interactive comment on* "Model of Propagation of VLF Beams in the Waveguide Earth-Ionosphere. Principles of Tensor Impedance Method in Multilayered Gyrotropic Waveguides" *by* Yuriy Rapoport et al.**

---

## Referee Comment (RC2) · Anonymous Referee #2 · 6 Jun 2019

Many points need to be revised before the acceptation of this paper. The interesting goal in a model is to determine what are the main parameters for the increase/decrease of the EM field. Here in this paper we only have a variation of one parameter: the electron density. It means that you show something which is evident: when the density increases the electric field decreases. What is the effect of other parameters as the magnetic field inclination for example? The plasma frequency? . . .. Why the calculation is stopped at 80 km ? In Figure 4 why Ey is oscillating along Z ?

Minor points: The English is not fluent and there are many mistakes (or typos) which

can be easily corrected with a word processor. - Page 3 line 21 To Be Corrected - Page 3 line 25 Wait (I have not checked the references but I have seen that Ruibie & Tolutue is not correct) - Page 4 line 10 waves - Page 4 line 11 LAIM appear before and then must be explained before - Legend of Figure 1 is too long. A part must be in the text (it is also true for other figures). - Of course I have not checked the correctness of all equations but I have seen an error in the first equation (equation (1)) for the ion plasma frequency - Page 6 line 5 and line 17 the sign inside exp() is different - Page 7 the values of BETAij are not clear. What parameters they contain ? - Page 9 line 10 respectively two times - Page 9 line 15 relation - The matrix at the end of equation (14) seems strange. The left lower element is not 1-i ? - Page 13 another parameter DELTA appears here. Is the DELTA in equation (11) similar to the DELTA in equation (24) ? - Title 3.5 too long - Page 16 a lot of typos, discharges, demonstrating, speaking, present, presentation - Page 17 line 1 these - Page 17 line 9 why Figures 3_2 and 4_3 - In Figure 3 it is difficult to understand the contain of the panels b) to g) - Page 19 line 13 figure 5

---

## Author Comment (AC1) · 18 Aug 2019

We thank P.F. Biagi for his attention to our work and positive decision about the paper 'Model of Propagation of VLF Beams in the Waveguide Earth-Ionosphere. Principles of Tensor Impedance Method in Multilayered Gyrotropic Waveguides'. The referee had spent his time and had applied great efforts to read and evaluate our paper. We are very grateful for the approval of the TIMED method. Besides, text of the paper has been supplied by new results obtained using our model of VLF wave propagation in the waveguide 'Earth-ionosphere'.

Sincerely Yours,

On behalf of the Authors,

Yuriy Rapoport

Senior Research Fellow
Physics Facuty
National Taras Shevchenko University of Kyiv
UKraine

---

## Author Comment (AC2)

5   We are very grateful to the Anonymous Referee #2 and are answering directly to each of the main his comments and will answer to each of the minor comments in the final text of the revised paper as well. We believe that the paper becomes really better and more clear, informative and useful for the readers due to the improvements inspired by the Referee.

For the illustration of the improvements done in the response to the comments of Referee 2, we added the text

10   of the new draft of the paper in the end of the present document, started with **P. 11**., as additional illustration material. The section 5 and the Figures 8-10 are new, and the corresponding Conclusions are added. In our answers to the Referee's comment, we refer to this additional material, included in the present file. WE refer to corresponding pages and lines such as f.e.  "**P31**" and "**L20**". We tried to include the new material, added to the previous text of the paper in response to Referee's comments in self-consistent manner, to make clear, that the

15   material both answers to the questions and is suitable to the general context of our work. Nevertheless this is not the final version of the revised paper yet. The last will be prepared only when and if we get the corresponding permissions from the Referee and the Editors of AnGeo. The questions of the Referee are revealed below by "black bold" font.

**(1) The interesting goal in a model is to determine what are the main parameters for the**

20   **increase/decrease of the EM field. Here in this paper we only have a variation of one parameter: the electron density.**

Concerning the perturbations of the electron concentration: we modified the perturbation as follows. We include in the revised paper, besides increase or decrease in electron concentration, also definite parameters of these variations. Namely, we include parametrization of the perturbation in electron concentration, in

25   particular maximum of the perturbation, region, where it is concentrated and effective width of this distribution by the vertical coordinate. Then in response to this note of the Referee 2, we added also an influence of the change in the angle of the inclination of geomagnetic field and carrier frequency of the beam. Then besides just field spatial distributions, as an object of the influence of the parameters, we include two other very interesting and important parameters. One of them is a parameter of the  polarization transformation $|E_y/H_y|$, described in

30   new Sect. 5, **P31**, **L1-L25**. This value can be measured and characterizes the effect of the gyrotropy and anisotropy, both volume and surface, described by the tensor surface impedances at the lower and upper boundaries of the waveguide Earth-Ionosphere (WGEI). The other value, the influence on which we analyze, is the complex tensor impedance, in particular real and imaginary parts of its diagonal element $Z_{11}$. In particular, an influence on it of the inclination angle of the geomagnetic field and carrier beam frequency are included into

35   the revised paper. The details concerning these extra simulations with the influences on the propagation of a

beam in the WGEI of different parameters are presented in the Figs. 8**(P32, L1-L5),** Fig. 9**(P33, L9),** Fig.10**(P34,L5)** and the texts with explanations included after these Figures.

Now, we have an influence on the propagation losses of VLF beam in the WGEI of NOT only one parameter. Note that in the text of the paper's draft the word "Figure" and corresponding number for all the new figures, added now, reflected the new simulations, are revealed by the red color (f.e. "Figure 8") in the figure captions. Then the change of numbers of Figures concerns the Figures 11-13 (**PP 35-36**), where *only* numbers are really changed comparatively to the previous version of the paper. The Figures 11-13 describe experimentally and (qualitatively) theoretically a common effect of seismogenic changing electron concentration and collision frequency (in accordance with our previous model of the influence on the ionosphere of the seismogenic electrostatic field including the corresponding heating-photochemistry effects). Therefore this is rather the complicated mechanisms which include a set of parameters in the ionosphere (the number of which is >> 1) such as characteristics of external electrostatic sources placed in the lower atmosphere and mesosphere, photochemistry parameters, electron concentration distribution, photochemistry parameters etc, effect of which is reflected finally in Figs. 12 a- c (**P35, L11**). In the modeling beam propagation in the WGEI, we used only qualitatively the fact, that as a result of mentioned above seismogenic mechanism, electron concentration $N_e$ and collision frequency $v_e$ change simultaneously ($N_e$ and $v_e$ increase and decrease, respectively). As a result, the curve 2, comparatively to the curve 1 in Fig. 13 a,b (**P36, L3**), reflects an effect of additional losses of VLF beam, which occurs due to simultaneous change of not *one,* but *two* parameters, and, moreover, these parameters change their signs consistently. See also test in Sect. 6, **P35, L2-4; P36, L13-28. P37, L1-7.**

**(2)** **Here in this paper we only have a variation of one parameter: the electron density. It means that you show something which is evident: when the density increases the electric field decreases.**

As it is already mentioned in item (1), a variations of other parameters are included as well, such as geomagnetic field inclination angle, beam carrier frequency, **value of surface impedance/conductivity of the lower boundary of the WGEI (atmosphere-Earth boundary)**. **Corresponding results are described in more details below (see items ).** But even effects from electron variations are *not* evident! Namely:

(i)     The simultaneous effect from change of electron concentration $N_e$ and collision frequency $v_e$ is non-trivial. First, as described above, this effect is caused by the consistent change of two different parameters, and consistent change in their signs is described by rather complicated model of seismogenic electrostatic-heating-photochemistry effects, described in particular in (Rapoport et al. 2006; Grimalsky et al. 2003). The consistent change of signs of both $N_e$ and $v_e$ causes the proper change in VLF losses/amplitude of the electromagnetic field (curve 2 in Figs. 13 a, b, **P36**, **L3**), which qualitatively corresponds to the effect observed before strong earthquake (Fig. 11, **P35**, **L11**).

(ii)    As it is seen from Fig. 9 (**P33**, **L9**), the change in the concentration in the lower ionosphere causes rather nontrivial effect on the parameter of polarization transformation |Ey/Hy|. Note that either increase or decrease in the ionosphere plasma concentration have been reported as a result of

seismogenic phenomena, tsunamis, particle precipitation in the ionosphere due to wave-particle interaction in the radiation belts (Pulinets et al. 2005; Shinagawa et al. 2013; Arnoldy et al. 1989; Glukhov et al. 1992; Tolstoy et al. 1986) etc. Namely, this effect does not reduces only to increase (Fig. 9 b) or decrease (Fig. 9 c) of the maximum value of the polarization transformation parameter |Ey/Hy|. Note also that the corresponding change of this parameter has rather remarkable absolute values from dozens to thousands percent, as it is seen from the comparison between Figs. 9_new b, c and Fig. 8 c, curve 3. The last curve corresponds to the unperturbed distribution of the ionospheric electron concentration (see curve 1, Fig. 5b (**P29**, **L11**) and curve 1 in Fig. 9 a). It is even more interesting that in the case of decreasing (Fig. 9_new a, curve 2) electron concentration, the main maximum of |Ey/Hy| occurs in the lower atmosphere (at the altitude around 20 km, Fig. 9 b, curve 3, which corresponds to $\omega = 1.14 \cdot 10^5 c^{-1}$). In the case of increasing electron concentration (Fig. 9_new a, curve 3) the main maximum of |Ey/Hy| occurs near the E region of the ionosphere (at the altitude around 77 km), while the secondary maximum placed, in the absence of the perturbation of the electron concentration, in the lower atmosphere (Fig. 8 c, curves 2, 3, **P32**, **L1-5**), or mesosphere/ionospheric D region ((Fig. 8_c, curve 1), practically disappears or just is not seen in the present scale, in the case under consideration (Fig. 9 c, curves 1-3, **P32**, **L9**).

**The text in item (ii) above is included into the added draft in P33, L18-29 and P34, L 1-4. These results are summarized in the Conclusion (7), item (ii), P40, L4-12.**

**(3) What is the effect of other parameters as the magnetic field inclination for example? The plasma frequency?**

As it was already mentioned above in the paragraph (1), in response to this comment, we have added also an influence of the change in the angle of the inclination of geomagnetic field and carrier frequency of the beam. Then besides just field spatial distributions, as an object of the influence of the parameters, we include two other very interesting and important parameters, namely polarization transformation. This value can be measured and characterizes the effect of the gyrotropy and anisotropy, both volume and surface, described by the tensor surface impedances at the lower and upper boundaries of the waveguide Earth-Ionosphere (WGEI). The other value , the influence on which we analyze, is the complex tensor impedance, in particular real and imaginary parts of its diagonal element $Z_{11}$. In particular an influence on it of the inclination angle of the geomagnetic field and carrier beam frequency are included into the revised paper. The following results are obtained, in particular, (and reflected in the new Figs. 8 -10, **PP 34-34**) basing on this new modeling, performed in the response to the present Reviewer's note.

(i) As it is seen from Figs. 8 a-c (**P32**, **L1-4**), the altitude dependence of the polarization parameter $|E_y/H_y|$ has two main maxima in the WGEI, the higher of which lies in the gyrotropic region above 70 km, while the other in the isotropic region of the WGEI. As it is seen from Fig. 8 a, b, the value of the larger second maximum increases, while the position of the second maximum shifts to the lower altitudes with increasing frequency. At the higher frequency ($\omega = 1.14 \cdot 10^5 c^{-1}$), the larger maximum of the polarization parameter corresponds to the intermediate value of the angle

$\theta = 45°$ (Fig. 8_new b); for the lower frequency ($\omega = 0.86 \cdot 10^5 c^{-1}$), the largest value of the first (higher) maximum corresponds to the almost vertical direction of the geomagnetic field ($\theta = 5°$, Fig. 8_new a). For the intermediate value of the angle ($\theta = 45°$), the largest value of the main maximum corresponds to the higher frequency ($\omega = 1.14 \cdot 10^5 c^{-1}$) in the considered frequency range (Fig. 8_new c). The total losses increase monotonically with increasing frequency and depend weakly on the value of $\theta$ (Fig. 8_new d).

**These results are presented in the present draft of the paper after the caption to Figure 8 (P32, L1-4) and summarized in Conclusion (7), item (i), P39, L28-34.**

(ii)    As it is seen from Fig. 9 (**P33**, **L9**), the change in the concentration in the lower ionosphere causes rather nontrivial effect on the polarization transformation parameter |Ey/Hy |. Note that either increase or decrease in the ionosphere plasma concentration have been reported as a result of seismogenic phenomena, tsunamis, particle precipitation in the ionosphere due to wave-particle interaction in the radiation belts (Pulinets et al. 2005; Shinagawa et al. 2013; Arnoldy et al. 1989; Glukhov et al. 1992; Tolstoy et al. 1986) etc. Namely, this effect does not reduces only to increase (Fig. 9 b) or decrease (Fig. 9 c) of the maximum value of the polarization transformation parameter |Ey/Hy|. Note also that the corresponding change of this parameter has rather remarkable absolute values from dozens to thousands percent, as it is seen from the comparison between Figs. 9 b, c (**P33**, **L9**) and Fig. 8 c, curve 3, **P32**, **L1-4**. The last curve corresponds to the unperturbed distribution of the ionospheric electron concentration (see curve 1, Fig. 5b, **P29**, **L11**, and curve 1 in Fig. 9). It is even more interesting that in the case of decreasing (Fig. 9 a, curve 2 ) electron concentration, the main maximum of |Ey/Hy | occurs in the lower atmosphere (at the altitude around 20 km, Fig. 9 b, curve 3, which corresponds to $\omega = 1.14 \cdot 10^5 c^{-1}$ ). In the case of increasing electron concentration (Fig. 9 a, curve 3) the main maximum of |Ey/Hy | occurs near the E region of the ionosphere (at the altitude around 77 km), while the secondary maximum placed, in the absence of the perturbation of the electron concentration, in the lower atmosphere (Fig. 8, curves 2, 3, **P32, L1-4**), or mesosphere/ionospheric D region ((Fig. 8_new c, curve 1),  practically disappears or just is not seen in the present scale, in the case under consideration (Fig. 9 c, curves 1-3, **P33**, **L9**).

**These results are presented in the present draft of the paper after the caption to Fig. 9,  P33, L18-29, and summarized in  Conclusion (7), item (ii), P40, L4-11.**

(iii)    As it is seen from Fig. 10, **P34, L5**, the real (a) and imaginary (b) parts of the surface impedance at the upper boundary of the WGEI have a quasiperiodical character with the amplitude of "oscillations" occurring around some  effective average values (not shown explicitly in Figs. 10 a, b) decreases with increasing the angle $\theta$. Even without the determination of the exact average values for each of the curves 1-4 in Figs. 10 a, b, it is seen that corresponding average values of Re($Z_{11}$) and Im($Z_{11}$) , in general, decrease with increasing angle $\theta$. It is also seen that average values of Re($Z_{11}$) for $\theta$ equal to $5°, 30°, 45°$ and $60°$ (curves 1-4 in Fig. 10 a) and Im($Z_{11}$) corresponding to $\theta$ equal to $45°$ and $60°$ (curves 3, 4 in Fig. 10 b),  increase with increasing frequency in the considered frequency

range (0.86-1.14)·$10^5$ c$^{-1}$. The average values of Im(Z$_{11}$) corresponding to $\theta$ equal to $5°$ and $30°$, change in the frequency range (0.86-1.14)·$10^5$ c$^{-1}$ non-monotonically, having maximum values around frequency (1-1.1)·$10^5$ c$^{-1}$.

**These results are included in the present draft of the paper just after the caption to Figure 10, P34, L15-22, and summarized in Conclusion (7), item (iii), P40, L12-19.**

(iv)     It is interesting to note that the value of finite impedance at the lower (Earth-atmosphere) boundary of the WGEI make a quite observable influence on the polarization transformation parameter minimum near the E region of the ionosphere (curves 1, 2 in the Fig. 10 c, **P34, L5**). Namely, the decrease of surface impedance *Z* at the lower boundary (Earth-atmosphere) of the WGEI in two orders causes the increase of the corresponding minimum value of |*Ey/Hy*| in ~ 100% (compare minima in the curves 1 and 2 around *z*=70 km).

**These results are presented in the end of the text after the caption to Figure 10, P34, L22-26, and summarized in Conclusion (7), item (iv), P40, L20-24.**

Concerning the plasma frequency ( $\omega_{pe,i}$ ) it is proportional to the square root of the electron concentration $N_e$ and included into complex conductivity and complex tensor $\hat{\varepsilon}$ . These values determine also effective tensor of surface impedance. Therefore each and all of the electrodynamics characteristics both presented in the initial version of the paper and added and reflected in the new Figs. 8-10, **PP32-34,** added now in the response to the Reviewer's notes, are influenced by these volume tensor $\hat{\varepsilon}$ and surface impedance $Z_{ij}(i,j=1,2)$ and therefore by the altitude distributions of $N_e$ and $\omega_p$ . For example, change in the altitude distribution of *N$_e$* (Fig. 5b, **P29, L11**) causes change in $\hat{\varepsilon}$ (Figs. 5 c, d, **P29, L11**) and surface tensor impedance $\hat{Z}$ (Table 1, **P29, L21**). As a result, field spatial distributions change (Fig. 6, **P30, L1**; Fig. 7, **P30, L6** ). As a result of the change in electron concentration (and therefore $\omega_{pe,i}$ ), transformation polarization parameter |*Ey/Hy*| andt its altitude distribution change very nontrivially (Fig. 9, **P33, L9**). There are only couple of examples, but in any of the others results obtained in the paper, an influence of $N_e$ and $\omega_p$ are reflected. We do not emphasize separately $\omega_{pe,i}$ because they are determined unambiguously by $N_e$ , the influence of which is investigated rather in details and the masses of the corresponding particles, while the last are supposed to be known in the scope of the approximations accepted in this paper.

**(4)  Why the calculation is stopped at 80 km ?**

The proposed new tensor impedance method for modeling propagation of electromagnetic beams (TIMEB) and the developed model allows and we really did the simulations of (all) the electromagnetic field components both inside the WGEI (*0<z<L$_z$* , *L$_z$*=85 km) and above the WGEI (*L$_z$<z<L$_{max,}$* *L$_{max}$*=300 km). Nevertheless this paper is the first of the planned set of the papers, and this paper is devoted, besides the new method in general, also

to the propagation of the bean *inside* the WGEI, i.e. in the range of altitudes . *0<z<L$_z$* , *L$_z$*=85 km. The calculations of the field above the WGEI are performed in this context only to establish /confirm the present approximation of the propagation of the beam *inside* the WGEI . In other words the results of the field calculations above the WGEI, in the region *L$_z$<z<L$_{max,}$* *L$_{max}$*=300 km  confirms that the region *0<z<L$_z$* , *L$_z$*=85 km is really a good

5    waveguide for the VLF field accounting for the effects of the gyrotropy and anisotropy for the plasma-like media placed in the inclined geomagnetic field. This fact is really confirmed by the calculations of (all) field components performed for the range of the altitudes from 0 to 300 km. Nevertheless because, as it was already mentioned above, this paper is devoted to the beam propagation in the WGEI, only final qualitative conclusion based on the calculation above the WGEI is presented in the paper. Namely, **in the response to the present**

10  **Reviewer's question, the following text is added into the paper.**

"Note   the following. The present paper is devoted to the new method of modelling characteristics of the WGEI, namely TIMEB and to the illustration of this method by the examples of the beam propagation in WGEI presented above. Respectively the field shown in Figs. 4 (**P28, L5**), 6 (**P30, L1**), 7 (**P30, L6**), 13 (**P36, L3**) include the range of altitudes inside the WGEI. Nevertheless the present method, in particular the application of the

15  formulas (30) (**P34, L10**), (24) (**P23, L12**), (23)(**P23, L4**), (27) (**P23, L21**)  and (15)-(19) (**P20, L 15-26**; **P 21, L 1-5**) allows to determine all the field components in the  range  of altitudes $0 \leq z \leq L_{\max}$ , where $L_{\max} = 300 \text{ km}$ . We will present here only the final qualitative result of such simulations.  Namely, it is shown that in the range $L_z \leq z \leq L_{\max}$ where $L_z = 85 \text{ km}$ is the upper boundary of the effective WGEI, all the field components are (1) at least one order of altitude less than the corresponding maximal value in the WGEI and (2) field components

20  have the oscillating character (along $z$ coordinate) and describes the modes, leaking from the WGEI. The detail consideration of the electromagnetic waves leaking from the WGEI will be presented in the special paper."

**This text is included into the second paragraph after the caption to the Fig. 13, P27, L8-18, and these results are summarized in the paragraph (8) of the Conclusions, P40, L25-34; P41, L1-3.**

To the details concerning the waves leaking from the WGEI and comparison of the theoretical simulations to the

25  results of the corresponding observations, the special and separate paper will be devoted. And all the calculations for the field in the altitude range above the WGEI will be suitable for a direct inclusion only in this paper.

    **(5)  In Figure 4 why Ey is oscillating along Z ?**

**The following note is made in the present draft of the paper in the response to this question of the Reviewer,**

30  **see . P37, L18-29:**

"Let us make a note also on the dependences of the field components in the WGEI on the vertical coordinate (*z*) and the change of such a dependence during a propagation of the VLF beam along the WGEI (Figs. 4 (**P28, L5**), 6 (**P30, L1**), 7 (**P30, L6**), 13 (**P36, L3**)). The initial distribution of the electromagnetic field on *z* (Fig. 4a) is determined by the initial conditions on the beam, see relations (32). Such a field includes, naturally, higher

eigenmodes of the WGEI. The structure and behaviour of these eigenmodes in the WGEI will be a subject of a separate paper. Here we only note on this subject that the higher-order modes, in distinction to the lower ones, have quite large losses and practically disappear after a beam propagation for a distance of order 1000 km in the WGEI. This circumstance determines the change in the altitude ($z$) and transverse ($y$) distributions of the

5    field of the beam during its propagation along the WGEI. In particular at the distance $x$=600 km from the beam input (Figs. 4 b, c, **P28, L5**) the few lowest modes of WGEI along $z$ and $y$ coordinate are still survived. At $x$=1000 km (Figs. 4 d, c, 6 e, f, 7 a, b), practically, only main mode in $z$ direction is survived. Note that the field picture mentioned above concerns real WGEI with losses, and also gyrotropy and anisotropy cause both volume effects and surface impedance, in distinction to the ideal planar metallized waveguide with isotropic filling (Collin

10   2001). "

**Minor points:**

**The English is not fluent and there are many mistakes (or typos) which can be easily corrected with a word processor**

The English will be improved by our co-authors who are working in USA and UK in the final version of the paper,
15   if we will be allowed by the Editors and Referee to prepare it; the correction with a word processor has been done just now for the present draft of the paper and this procedure will be repeated as well when the final version of the paper will be prepared.

**Page 3 line 21 To Be Corrected**

Just in case I include below also the lines neighbouring to line 21 in P. 3 (in the previous version of the paper):

20   **"Some other details on the distinctions from the previously published models are given below in Sect. 3. The methods of effective boundary conditions, in particular effective impedance conditions (Tretyakov, 2003; Senior and Volakis, 1995; Kurushin and Nefedov, 1983) are well-known and can be used, in particular, for the layered metal-dielectric, metamaterial and gyrotropic active layered and waveguiding media of different types (Tretyakov, 2003;"**

25   As we see just now, this peace of text is correct. Please specify if something still should be improved, and in this case this will be improved.

**Page 3 line 25 Wait –** done, see **P14, L13**; hereafter such improvements are revealed in the text od the new draft using the red color font

**(I have not checked the references but I have seen that Ruibie & Tolutue is not correct) –** improved, in
30   accordance with the cited paper, the text of which we have, see **P13, L24, 26**; **P14, L12**.

**Page 4 line 10 – waves:** changed to "electromagnetic waves", **P14, L12**.

**Page 4 line 11 LAIM appear before and then must be explained before** – "LAIM" has been explained in the 1-2 lines of the Introduction – on the page 2 – before page 4 line 11, see **P12, L6, 7** .

**Legend of Figure 1 is too long. A part must be in the text (it is also true for other figures).**

As it was recommended, the caption to Figure 1 is reduced. Namely the phrases:

5 " $\theta$ Is the angle between the directions of the vertical axis $z$ and geomagnetic field $\vec{H}_0$ . Note that The coordinate system $x'y'z'$ included in the Fig. 1 is connected with the geomagnetic field; $\vec{H}_0$ is directed along axis $z'$, lies in the plane $xz$ , while the planes $x'z'$ and $xz$ coincide with each other "

are transferred from the end of the caption to Figure 1 to the text of the article before Figure 1, see **P15, L8-11**. The signature of the remaining figures includes only the data necessary to identify these figures, and to
10 distinguish among themselves the different curves in each figure. The information included in the captions is fundamentally necessary to provide readers with the opportunity to quickly find out what is the meaning of each of the Figures separately, as well as all the Figures in general. As for the description of the figures in the text of the article, it is included for each figure separately and when comparing the physical effect between themselves, illustrated by various figures or groups of figures. At the same time, a description is also given of
15 the corresponding figures in the necessary proportion while the basic physical effects, illustrated by the corresponding figures are described. These effects are mainly then included in the Conclusions.

**Of course I have not checked the correctness of all equations but I have seen an error in the first equation (equation (1)) for the ion plasma frequency.**

The formulas in Eqs. (1) (**P16, L13**) for electron and ion plasma concentrations in the accepted approximation of
20 the three-component plasma-like ionosphere (electron, effective one type ion and neutral components) and quasineutrality, are right.

**Page 6 line 5 and line 17 the sign inside exp() is different.**

Yes, this is right- tere are two parts of the argument of the phase multiplier ~exp*(iⵁt -ik$_x$x)*, which really have different signs, see also **P16. L7, 19** .

25 **Page 7 the values of BETAij are not clear. What parameters they contain ?**

Please look at the line placed 3 lines above the upper formula from Eqs. (1), see **P16, L10**. It is written there: $\hat{\beta} = \hat{\varepsilon}^{-1}$, or $\vec{E} = \hat{\beta}\vec{D}$ (the arguments of $\hat{\varepsilon}$, $\hat{\beta}$ are omitted here, but included in the paper. Therefore the tensor $\hat{\beta}$ , inverse respectively to $\hat{\varepsilon}$ and depends on the same parameters as $\hat{\varepsilon}$ , which is described in Sect. 3.1, in particular using formulas (1) and then four lines after formula (1). Therefore $\hat{\beta}$ depends on the same
30 parameters as the tensor $\hat{\varepsilon}'$, described by formulas (1), with components, the altitude distributions of which is illustrated in Figs. 3 (**P27, L 24**; **P28, L1**) and 5c, d (**P29, L11**) , and by the angle $\theta$ and corresponding rotation

matrices, mentioned in the four lines (**P16, L14-17**) after Eq. (1) (and not included explicitly). Please note that due to chosen subject- layered anisotropy an gyrotropy inhomogeneous plasma-like Earth-Atmosphere-Ionosphere media - and respectively to do "what is necessary and how it is necessary", we are forced to choice combined analytical-numerical approach. Namely, all what is possible, we are doing analytically and all other-
5  numerically. In particular the tensor $\hat{\beta}$ is obtained, using proper formulas, from the tensor $\hat{\varepsilon}$ .

**Page 9 line 10 respectively two time** – yes, this phrase is improved now as follows, see **P19, L11, 12**:

" $k_x^{'}$ and $k_z^{'}$ are the components of wave number, respectively, transverse and longitudinal relatively to geomagnetic field."

**Page 9 line 15 relation - The matrix at the end of equation (14) seems strange. The left lower element is not 1-**
10  **i ? –** thank you very much, the typos in this formula is improved, see **P19, L19**. Namely, the upper right element is (-1-i). Just in case, this matrix has been obtained analytically by means of few different approaches independently, with the same result.

**Page 13 another parameter DELTA appears here. Is the DELTA in equation (11) similar to the DELTA in**
15  **equation (24) ?**  - thank you, there are different values. To distinguish between them, the corresponding value in equation (24) is re-denoted now as $\Delta_0$ , see **P23, L13**.

**Title 3.5 too long.-** Yes, the title 3.5 is shortened as follows (**P26, L11**):
**"3.5 The Modes of the VLF Waveguide. Reflection from the Upper Effective Boundary of VLF Waveguide."**

20  **Page 16 a lot of typos, discharges, demonstrating, speaking, present, presentation. – improved, namely:**

The first two phrases in the beginning of Chapter 3.5, namely (**P26, L12-14**)

"Our model, in general, needs the consideration of the excitations of the waveguide modes by means of current sources such as dipole-like VLF radio source and lightning discahrges. Then, we will present the results of the reflection of the waves incident on the upper boundary (*z=Lz*) of the effective WGEI demonsrtrating that this
25  structure has indeed good enough waveguiding properties."

Are replaced by:

"Our model, in general, needs the consideration of the excitations of the waveguide modes by means of current sources such as dipole-like VLF radio source and lightning discharge. Then, the reflection of the waves incident on the upper boundary (*z=L$_z$*) of the effective WGEI can be considered. There will be possible to demonstrate
30  that this structure has indeed good enough waveguiding properties"

As a result  of this replacement:

**discharges –** replaced by "discharge" (**P26, L13**)
**demonsrtrating –**removed

5   **Then the following improvement are done in this text, in the first paragraph of Sect. 3.5, p. 16:**

**Shortly speaking -** removed
**Present; presentation –** are improved (**P26, L23**; **P26, L24**).
**Page 17 line 1 these  -** improved (now it is came to the last line of p. 26: **P26, L26**).

**Page 17 line 9 why Figures 3_2 and 4_3 –** (the same concerns line 8) – improved as Fig. 3, Fig. 4 – see in the
10   present draft **P27, L7, 8**

 **In Figure 3 it is difficult to understand the contain of the panels b) to g) - Page 19 line 13 figure 5 –** Improved. Namely, quality of all panels in Figs. 3 and Fig. 5 is improved (widths of the lines increased, sizes of the letters and numbers in the Figures increased). Therefore the contain of all panels in Figs. 3, 5 should become understandable now, see **P27, L24, 28, L1**; **P29, L11**.

15   **Besides of that, the sign in square root in the first of formula (9) is improved –** there was before mistakenly, line 17, p. 8:

$$\kappa_{1,2}{}^2 = \frac{\beta_{11} + \beta_{22}}{2} \pm \left( (\frac{\beta_{11} + \beta_{22}}{2})^2 + \beta_{12}\beta_{21} \right)^{1/2}$$

There became after the improvement now, **P18, L19**:

$$\kappa_{1,2}{}^2 = \frac{\beta_{11} + \beta_{22}}{2} \pm \left( (\frac{\beta_{11} - \beta_{22}}{2})^2 + \beta_{12}\beta_{21} \right)^{1/2}$$

20       **All other necessary improvements, such as polishing English, Adding the inclusions, corresponding to the newly obtained results into the Abstract and Introduction and others, will be done after getting the permission from the Referee and Editors to prepare the final version of the revised paper.**
       **We also add separately the new Figs. 8, 9 , 10 and Figs, 3, 5, which are improved with the requirement of the Referee 2.**

**Model of Propagation of VLF Beams in the Waveguide Earth-Ionosphere. Principles of Tensor Impedance Method in Multilayered Gyrotropic Waveguides.**

Yuriy Rapoport[#1], Vladimir Grimalsky[*2], Victor Fedun[~3], Oleksiy Agapitov[⊥4], John Bonnell[⊥5], Asen Grytsai[#6], Gennadi Milinevsky[#ε7], Alex Liashchuk[±8], Alexander Rozhnoi[φ9], Maria Solovieva[φ10], Andrey Gulin[#11]

[#]Taras Shevchenko National University of Kyiv, Ukraine
[*]Autonomous University of State Morelos (UAEM), Mexico
[~]The University of Sheffield, UK
[⊥]University of California, Berkeley, USA
[ε] College of Physics, International Center of Future Science, Jilin University, Changchun, China
[±]National Center for Control and Testing of Space Facilities of the State Agency of Ukraine
[φ]Institute of the Earth Physics, RAS, Moscow, Russia

[1]yuriy.rapoport@gmail.com, [2]v_grim@yahoo.com, [3]v.fedun@sheffield.ac.uk, [4]oleksiy.agapitov@gmail.com, [5]jwbonnell@berkeley.edu, [6]a.grytsai@gmail.com, [7]genmilinevsky@gmail.com, [8]alex.liashchukk@gmail.com, [9]rozhnoi@ifz.ru, [10]MCSolovieva@gmail.com, [11]mytimetosay@gmail.com

*Correspondence to*: Yuriy Rapoport (yuriy.rapoport@gmail.com)

**Abstract.** Modeling propagation of VLF electromagnetic beams in the waveguide earth-ionosphere (WGEI) is of a great importance because variation in the characteristics of these waves is an effective instrument for diagnostics the influences on the ionosphere "from above" (Sun-Solar Wind-Magnetosphere-Ionosphere), "from below" (the most powerful meteorological, seismogenic and other sources in the lower atmosphere and lithosphere/Earth, such as hurricanes, earthquakes, tsunamis etc.), from inside the ionosphere (strong thunderstorms and lightning discharges) and even from the far space (such as gamma-flashes, cosmic rays etc.). Thus, VLF became one of the universal instruments for monitoring the Space Weather in the direct sense of this term, i.e. the state of the Sun-Earth space and the ionosphere as it is, particularly determined by all possible relatively powerful sources, wherever they are placed. This paper is devoted mostly to modeling VLF electromagnetic beam propagation in the WGEI. We present a new tensor impedance method for modeling propagation of electromagnetic beams (TIMEB) in a multi-layered/inhomogeneous waveguide. Suppose that such a waveguide, i.e. WGEI, possesses the gyrotropy and inhomogeneity with a thick cover layer placed above the waveguide. Note a very useful and attractive feature of the proposed TIMEB method: in spite of a large thickness of the waveguide cover layer, the proposed effective impedance approach reflects an impact of such a cover on the electromagnetic (EM) waves, which propagate in the waveguide. This impedance approach can be applied for EM waves/beams in layered gyrotropic/anisotropic active media in very wide frequency range, from VLF to optics. Moreover, this approach can be applied to calculations of EM waves/beams propagation in the media of an artificial origin such as metamaterial microwave or optical waveguides. The results of the modeling the propagation of VLF beams in the WGEI are included. The qualitative comparison between the theory and experimental observation of increasing losses of VLF waves in the WGEI is discussed. The new proposed method and its further development allow the comparison with the results of the future rocket experiment. This method

allows to model (i) excitation of the VLF modes in the WGEI and their excitation by the typical VLF sources, such as radio wave transmitters and lightning discharges and (ii) leakage of VLF waves/beams into the upper ionosphere/magnetosphere.

*Keywords* — **ionosphere, atmosphere, VLF, tensor impedance, gyrotropy, layered waveguide, beam, electromagnetic wave, boundary conditions, ionospheric disturbances, vertical coupling processes**

**5    1 Introduction**

This paper is dedicated to the propagation in the system Lithosphere–Atmosphere–Ionosphere–Magnetosphere (LAIM) of electromagnetic (EM) waves /beams in the radio range, with particular applications to very low frequencies (VLF). This topic became very actual due to the following reasons. (1) Variation in the characteristics of these waves is now an effective instrument for the diagnostics of "ionospheric weather" as a part of the Space Weather (Hapgood 2017; Yigit et
10    al. 2016; Richmond 1996) in its direct meaning: the state of the Sun-Earth space and the ionosphere in particular determined by all possible sufficiently powerful sources, wherever they are placed. Change in the characteristics (amplitude and phase) of the VLF waves propagating in the waveguide earth-ionosphere (WGEI) reflects the corresponding variations in the ionospheric electrodynamics characteristics (complex dielectric permittivity) and respectively, the influences on the ionosphere "from  above" (Sun-Solar Wind-Magnetosphere Ionosphere (WINDMII) (Patra  et al., 2011; Koskinen, 2011;
15    Boudjada et al., 2012; Wu et al., 2016), "from below" (the most powerful meteorological, seismogenic and other sources in the lower atmosphere and lithosphere/Earth, such as cyclones and hurricanes (Nina et al., 2017; Rozhnoi et al., 2014; Chou et al., 2015), earthquakes (Hayakawa, 2015; Surkov and Hayakawa, 2014; Sanchez-Dulcet et al., 2015), tsunamis etc. or from inside the ionosphere (strong thunderstorms and lightning discharges, terrestrial gamma-ray flashes or sprite streamers (Cummer et al., 1998; Qin et al., 2012; Dwyer  2012; Dwyer and Uman, 2014; Cummer et al., 2014; Mezentsev et al., 2018).
20    Note that the VLF signals are very important for the merging of the atmospheric physics and space plasma physics with the astrophysics and high-energy physics. The corresponding "intersection area" for these two disciplines includes cosmic rays and very popular now objects of investigation – high-altitude discharges (sprites), anomalous X-ray bursts, powerful gamma-ray bursts etc. The key phenomena for the occurrence of all of these objects are runaway electrons with runaway breakdown, and one of the necessary conditions of them is the presence of cosmic rays, consequently these phenomena are intensified
25    during the air showers generating by cosmic particles (Gurevich and Zubin 2001; Gurevich et al. 2009).. The runaway breakdown and lightning discharges including high-latitude ones case radio emission both in HF range, which could be observed using LOFAR and other radio telescopes (Buitink et al., 2014; Scholten et al., 2017; Hare 2018), and in the VLF range. Corresponding experimental research include measurements of the VLF characteristics by the international measurement system of the pairs "transmitted-receiver" separated by a distance of a couple of thousand km (Biagi et al.,
30    2011; Biagi et al., 2015). Another international system is based on the measurements of VLF characteristics for the characterization of the thunderstorms with the lightning discharges/World Wide Lightning Location Network (WWLLN) (Lu et al. 2019).  (2) Intensification of the magnetospheric research, wave processes, particle distribution and wave-particle

interaction in the magnetosphere including radiation belts leads to the great interest to the VLF plasma waves, in particular whistlers (Artemyev et al., 2013; Agapitov et al., 2014; Agapitov et al., 2018)**.**

The differences of our proposed model from the known ones used for the simulation of the VLF waves in the WGEI are the following. (1) In distinction to the impedance invariant imbedding model (Shalashov and Gospodchikov, 2010; Kim and Kim, 2016), our model provides optimal, for definite class of problems, which we consider, balance between the analytical and numerical approaches and is, in fact combined analytical-numerical one, basing on matrix sweep, method (Samarskii, 2001). As a result, this model allows obtaining analytically the tensor impedance and at the same time, provides high effectiveness and stability of the modeling. (2) In distinction to the full-wave finite difference time domain (FDTD) models such as (Chevalier and Inan, 2006; Marshall et al., 2017; Yaxin et al., 2012; Azadifar et al., 2017), our method provides very physically clear lower and upper boundary conditions, in particular physically justified upper boundary conditions corresponding to the radiation of the waves propagation in the WGEI to the upper ionosphere/magnetosphere. This allows in a perspective to determine the leakage modes and to interpret not only ground-based, but also satellite measurements of the VLF beam characteristics. (3) In distinction to the models (Kuzichev and Shklyar, 2010; Kuzichev et al., 2018; Lehtinen and Inan, 2009; Lehtinen and Inan, 2008) based on the mode presentations and made in the frequency domain, we use the combined approach, This approach includes condition of the radiation at the altitudes of the F region, equivalent impedance conditions in the lower E region and at the lower boundary of the WGEI, mode approach, and finally, beam method. This combined approach, finally, creates the possibility to interpret adequately data of both ground and satellite detection on the EM wave/beam propagating in the WGEI and these which leakage from the WGEI into the upper ionosphere/magnetosphere. Some other details on the distinctions from the previously published models are given below in Sect. 3.

The methods of effective boundary conditions, in particular effective impedance conditions (Tretyakov, 2003; Senior and Volakis, 1995; Kurushin and Nefedov, 1983) are well-known and can be used, in particular, for the layered metal-dielectric, metamaterial and gyrotropic active layered and waveguiding media of different types (Tretyakov, 2003; Senior and Volakis, 1995; Kurushin and Nefedov, 1983; Collin, 2001; Wait, 1996) including plasma-like solid state (Ruibys, and Tolutis, 1983) and space plasma (Wait, 1996) media. The plasma wave processes in the waveguide structures metal-semiconductor-dielectric, placed into the external magnetic field, were widely investigated (Ruibys and Tolutis, 1983; Maier, 2007; Tarkhanyan and Uzunoglu, 2006) in various frequency ranges, from radio to optical ones. Corresponding waves are applied in modern plasmonics and in non-destructive testing of semiconductor interfaces. It is of interest to realize the resonant interactions of volume and surface electromagnetic waves in these structures, so the simulations of the wave spectrum there are important. To describe such complex layered structures, it is very convenient and effective to use impedance approach (Tretyakov, 2003; Senior and Volakis, 1995; Kurushin and Nefedov, 1983). As a rule, impedance boundary conditions are used, when the layer covering waveguide is thin (Senior and Volakis, 1995; Kurushin and Nefedov, 1983). One of the known exclusions is the impedance invariant imbedding model, the distinction to which of our new method has been mentioned above. Our new approach TIMEB, proposed in the present paper has the set of very attractive,

for practical purposes, features. These features are: (i) the surface impedance characterizes cover layer of finite thickness, and this impedance is expressed analytically; (ii) the method allows an effective modelling of 3D beam propagating in the gyrotropic waveguiding structure; (iii) finally, if the considered waveguide can be modified by any external influence such as bias magnetic or electric fields, or by any extra wave or energy beams (such as acoustic or quasistatic fields etc.), the

5   corresponding modification of the characteristics (phase and amplitude) of the electromagnetic VLF beam propagating in the waveguide structure can be modelled.

Our approach was targeting properly and is suitable for the farther important development which will allow to solve also the following problems, which continue the list presented above: (iv) the problem of the excitation of the waveguide by the waves incident on the considered structure from above could be solved as well with the slight modification of the

10   presented model, with inclusion also ingoing waves; (v) consider a plasma-like system placed into the external magnetic field, such as the LAIM system (Grimalsky et al., 1999 a, b) or dielectric-magnetized semiconductor structure; then the electromagnetic waves radiated outside the waveguiding structure (such as helicons (Ruibys and Tolutis, 1983) or whistlers (Wait, 1996)) and the waveguide modes could be considered altogether; (vi) adequate boundary radiation conditions on the upper boundary of the covering layer are derived; and, based on this (and absence of ingoing waves), the leakage modes

15   above the upper boundary of the structure (in other words, upper boundary of covering layer), will be searched with the farther development of the model, delivered in the present paper. Namely, the process of the leakage of the electromagnetic waves from the (opened) waveguide, then their transformation into magnetized plasma waves, propagating along magnetic field lines, and, possibly, excitation of the waveguiding modes by the waves incident on the system from external space (Walker, 1976), can be modeled as a whole. Such a modification can be measured, characterizing the external fields and

20   corresponding field sources, caused the above-mentioned waveguide modification. Combining with the proper measurements of the phases and amplitudes of the electromagnetic waves, propagating in the waveguiding structures and leakage waves, the model possessing the above mentioned features can be used for searching, and even monitoring the external influences on the layered gyrotropic active artificial or natural media, for example microwave or optical waveguides or the system LAIM and WGEI, respectively.

25   The structure of the paper is as follows. In Sect. 2, formulation of the problem is presented. In Sect. 3 the algorithm is presented, including the determination of the conditions of radiation of the VLF waves/beams into the upper ionosphere/magnetosphere at the upper boundary, placed in the F region at the altitude (250-400) km; effective tensor impedance boundary conditions at the upper boundary (~ 85 km) of the effective WGEI; and finally the 3D model of the propagation of the VLF beam in the WGEI, which we call TIMEB, because in fact the beam method is combined with tensor

30   impedance method. The questions on the mode presentation and leakage modes of VLF are discussed very briefly, because the corresponding details will be presented in the next papers. In Sect. 4, the results of numerical modeling are presented. In Sect. 5, the discussion is presented, including an example of the qualitative comparison between the results of our theory and an experiment; and the future rocket experiment on the measurements of the characteristics of VLF signal, radiated from the VLF transmitter and propagating in the WGEI and penetrating into the upper ionosphere. Finally, conclusions are presented.

**2 Formulation of the problem**

The VLF electromagnetic (EM) waves with frequencies $f$ = 10 - 100 kHz can propagate along the Earth's surface for long distances >1000 km. The Earth's surface of a high conductivity z = 0 ( $z$ is vertical coordinate) and the ionosphere F-layer $z$ = 300 km form the VLF waveguide, see Fig. 1. The propagation of the VLF electromagnetic radiation excited by a near-Earth antenna within the WGEI should be described by the full set of the Maxwell equations in the isotropic atmosphere 0 < $z$ < 60 km, the approximately isotropic ionosphere D-layer 60 km < $z$ < 75 km, and the anisotropic E- and F- layers of the ionosphere, due to the geomagnetic field $\vec{H}_0$, added by the boundary conditions at the Earth's surface and at the F-layer. $\theta$ is the angle between the directions of the vertical axis $z$ and geomagnetic field $\vec{H}_0$. Note that the coordinate system $x'y'z'$ included in the Fig. 1 is connected with the geomagnetic field; $\vec{H}_0$ is directed along $z'$ axis, lies in the plane $xz$, while the planes $x'z'$ and $xz$ coincide with each other.

[Figure]

**Figure 1. The geometry of the anisotropic/gyrotropic waveguide. EM waves propagate in OX direction. $\vec{H}_0$ is the external magnetic field. The (effective) WGEI for EM waves occupies the region $0 < z < L_z$. Isotropic media occupies the region $0 < z < L_{ISO}$ , $L_{ISO} < L_z$ . Anisotropic/gyrotropic media occupies the region $L_{ISO} < z < L_{max}$ . Covering layer occupies the region $L_z < z < L_{max}$. WG includes isotropic region $0 < z < L_{ISO}$ and a part of anisotropic region $L_z < z < L_{max}$. It is supposed that the anisotropic region is relatively small part of the WG, $(L_z-L_{ISO})/L_z$ ~ (0.1-0.2). At the upper boundary of covering layer ($z = L_{max}$) the radiation of EM to the external region ($z > L_{max}$) is accounted for with the proper boundary conditions. Integration of the equations describing the EM field propagation allows to obtain effective impedance boundary conditions at the upper boundary of effective WG ($z = L_z$). These boundary conditions effectively includes all the effect on the wave propagation of the covering layer and the radiation (at $z = L_{max}$) to the external region ($z > L_{max}$).**

**3. Algorithm**

We present here the algorithm of the new proposed method, staying in details only on the main subject of the present paper, in particular on the boundary conditions, impedance method and the method for the beam propagation in the WGEI. The other parts of the method, connected with the mode presentation of the excitation of WGEI by a given current source and the

reflection of the EM waves from the effective upper boundary of the WGEI and leakage of the EM waves from the WGEI to the upper ionosphere/magnetosphere, will be outlined here only very briefly and will become the subjects of the next papers.

**3.1 Direct and inverse tensors characterizing the ionosphere**

In the next subsections we will derive the formulas describing the transfer of the boundary conditions at the upper boundary ($z=L_{max}$), Fig. 1, resulting in the tensor impedance conditions at the upper boundary of the effective WGEI ($z=L_i$). To make this, we need, firstly, to describe the tensors, characterizing the ionosphere. The monochromatic EM field is considered with the components of EM field ~ $exp(i\omega t)$. The main goal is to transfer the EM boundary conditions from the upper ionosphere at the height $L_z \sim 250 - 400$ km to the lower ionosphere $L_z \sim 70 - 90$ km. The vertical axis is $OZ$, the inclination angle of the geomagnetic field is $\Theta_I$ (Fig. 1). The anisotropic medium is inhomogeneous along $OZ$ axis only and is characterized by the tensor permittivity $\hat{\varepsilon}(\omega, z)$ or by the inverse tensor $\hat{\beta}(\omega, z) = \hat{\varepsilon}^{-1}(\omega, z)$: $\vec{E} = \hat{\beta}(\omega, z) \cdot \vec{D}$, where $\vec{D}$ is the electric induction. Below the absolute units are utilized. The expressions for the components of the effective permittivity of the ionosphere are in the coordinate frame $X'YZ'$ where $OZ'$ axis is aligned along the geomagnetic field $\vec{H}_0$:

$$\hat{\varepsilon}' = \begin{pmatrix} \varepsilon_1 & \varepsilon_h & 0 \\ -\varepsilon_h & \varepsilon_1 & 0 \\ 0 & 0 & \varepsilon_3 \end{pmatrix}, \quad \varepsilon_1 = 1 - \frac{\omega_{pe}^2 \cdot (\omega - i\nu_e)}{((\omega - i\nu_e)^2 - \omega_{He}^2) \cdot \omega} - \frac{\omega_{pi}^2 \cdot (\omega - i\nu_i)}{((\omega - i\nu_i)^2 - \omega_{Hi}^2) \cdot \omega}, \quad \varepsilon_h \equiv ig;$$

$$g = -\frac{\omega_{pe}^2 \cdot \omega_{He}}{((\omega - i\nu_e)^2 - \omega_{He}^2) \cdot \omega} + \frac{\omega_{pi}^2 \cdot \omega_{Hi}}{((\omega - i\nu_i)^2 - \omega_{Hi}^2) \cdot \omega}, \quad \varepsilon_3 = 1 - \frac{\omega_{pe}^2}{(\omega - i\nu_e) \cdot \omega} - \frac{\omega_{pi}^2}{(\omega - i\nu_i) \cdot \omega}; \tag{1}$$

$$\omega_{pe}^2 = \frac{4\pi e^2 n_0}{m_e}, \quad \omega_{pi}^2 = \frac{4\pi e^2 n_0}{m_i}, \quad \omega_{He} = \frac{eH_0}{m_e c}, \quad \omega_{Hi} = \frac{eH_0}{m_i c}$$

Here $\omega_{pe}, \omega_{pi}, \omega_{He}, \omega_{Hi}$ are plasma and cyclotron frequencies for electrons and ions respectively; $m_e$, $m_i$, $\nu_e$, $\nu_i$ are the masses and the collision frequencies. The expressions of the components of $\hat{\varepsilon}(\omega, z)$ are obtained from (1) by means of multiplication with the rotation matrices (Spiegel, 1959). In the case of a medium with a scalar conductivity $\sigma$, like the lower ionosphere or atmosphere, the effective permittivity (1) reduces to the scalar: $\varepsilon = 1 - 4\pi i \sigma/\omega$

**3.2 The equations for the EM field and upper boundary conditions**

The EM field depends on the horizontal coordinate $x$ as ~$exp(-ik_x x)$. Generally, $k_x \leq k_0$, where $k_0 = \omega/c$. In simulations of VLF beam propagation, we put $k_x = k_0$. In the case of searching VLF waveguide modes $k_x$ is slightly complex and should be calculated from boundary conditions at the Earth's surface and upper surface of the effective WGEI.

The Maxwell equations are:

$$-\frac{\partial H_y}{\partial z} = ik_0 D_x, \quad \frac{\partial H_x}{\partial z} + ik_x H_z = ik_0 D_y, \quad -ik_x H_y = ik_0 D_z$$

$$-\frac{\partial E_y}{\partial z} = -ik_0 H_x, \quad \frac{\partial E_x}{\partial z} + ik_x E_z = -ik_0 H_y, \quad -ik_x E_y = -ik_0 H_z \tag{2}$$

In eq. (2), $E_x = \beta_{11} D_x + \beta_{12} D_y + \beta_{13} D_z$ etc. All the components of the EM field can be represented through the horizontal components of the magnetic field $H_x$, $H_y$, and the following equations for these components have been derived:

$$\frac{\partial}{\partial z}\left(\frac{\beta_{22}}{1-\beta_{22}\frac{k_x^2}{k_0^2}}\frac{\partial H_x}{\partial z}\right) - \frac{\partial}{\partial z}\left(\frac{\beta_{21}}{1-\beta_{22}\frac{k_x^2}{k_0^2}}\frac{\partial H_y}{\partial z}\right) - ik_x\frac{\partial}{\partial z}\left(\frac{\beta_{23}}{1-\beta_{22}\frac{k_x^2}{k_0^2}}H_y\right) + k_0^2 H_x = 0 \tag{3a}$$

$$\frac{\partial}{\partial z}\left((\beta_{11} + \frac{k_x^2}{k_0^2}\frac{\beta_{12}\cdot\beta_{21}}{1-\beta_{22}\frac{k_x^2}{k_0^2}})\frac{\partial H_y}{\partial z}\right) - \frac{\partial}{\partial z}\left(\frac{\beta_{12}}{1-\beta_{22}\frac{k_x^2}{k_0^2}}\frac{\partial H_x}{\partial z}\right) +$$

$$+ik_x\frac{\partial}{\partial z}\left((\beta_{13} + \frac{k_x^2}{k_0^2}\frac{\beta_{12}\cdot\beta_{23}}{1-\beta_{22}\frac{k_x^2}{k_0^2}})H_y\right) + ik_x(\beta_{31} + \frac{k_x^2}{k_0^2}\frac{\beta_{32}\cdot\beta_{21}}{1-\beta_{22}\frac{k_x^2}{k_0^2}})\frac{\partial H_y}{\partial z} - \tag{3b}$$

$$-ik_x\frac{\beta_{32}}{1-\beta_{22}\frac{k_x^2}{k_0^2}}\frac{\partial H_x}{\partial z} + k_0^2(1-\beta_{33}\frac{k_x^2}{k_0^2} - \frac{k_x^4}{k_0^4}\frac{\beta_{23}\cdot\beta_{32}}{1-\beta_{22}\frac{k_x^2}{k_0^2}})H_y = 0$$

The expressions for the horizontal components of the electric field $E_x$, $E_y$ are:

$$E_x = \frac{i}{k_0}\left((\beta_{11} + \frac{k_x^2}{k_0^2}\frac{\beta_{12}\cdot\beta_{21}}{1-\beta_{22}\frac{k_x^2}{k_0^2}})\frac{\partial H_y}{\partial z} - \frac{\beta_{12}}{1-\beta_{22}\frac{k_x^2}{k_0^2}}\frac{\partial H_x}{\partial z}\right) - \frac{k_x}{k_0}(\beta_{13} + \frac{k_x^2}{k_0^2}\frac{\beta_{12}\cdot\beta_{23}}{1-\beta_{22}\frac{k_x^2}{k_0^2}})H_y$$

$$E_y = \frac{i}{k_0}\left(-\frac{\beta_{22}}{1-\beta_{22}\frac{k_x^2}{k_0^2}}\frac{\partial H_x}{\partial z} + \frac{\beta_{21}}{1-\beta_{22}\frac{k_x^2}{k_0^2}}\frac{\partial H_y}{\partial z}\right) - \frac{k_x}{k_0}\frac{\beta_{23}}{1-\beta_{22}\frac{k_x^2}{k_0^2}}H_y \tag{4}$$

In the region $z \geq L_{max}$ the upper ionosphere is assumed weakly inhomogeneous, and the geometric optics approximation is valid in the VLF range there. Note that such an approximation is invalid at the upper boundary of the effective VLF WGEI at 80 – 90 km because of the great inhomogeneity of the ionosphere in the vertical direction within $E$-layer. These circumstances determine the choice of the upper boundary $z = L_{max} \sim$ (250-400) km, where the conditions of the radiation are

formulated. The dispersion equation connected the wave numbers and the frequency of the outgoing waves has been got from eqs. (3), where $H_{x,y} \sim e^{-ik_z \tilde{z}}$, while the derivatives like $\partial \beta_{11}/\partial z$ and the inhomogeneity of the media are neglected:

$$\left( \beta_{22} k_z^2 - k_0^2 (1 - \beta_{22} \frac{k_x^2}{k_0^2}) \right) \cdot \left( (\beta_{11}(1 - \beta_{22} \frac{k_x^2}{k_0^2}) + \frac{k_x^2}{k_0^2} \beta_{12} \cdot \beta_{21}) k_z^2 + ((\beta_{13} + \beta_{31})(1 - \beta_{22} \frac{k_x^2}{k_0^2}) + \right.$$
$$+ \frac{k_x^2}{k_0^2}(\beta_{12} \cdot \beta_{23} + \beta_{32} \cdot \beta_{21}) k_x k_z - k_0^2 ((1 - \beta_{33} \frac{k_x^2}{k_0^2})(1 - \beta_{22} \frac{k_x^2}{k_0^2}) - \frac{k_x^4}{k_0^4} \beta_{23} \cdot \beta_{32}) \bigg) -$$
$$- \left( \beta_{21} k_z^2 + \beta_{23} k_x k_z \right) \cdot \left( \beta_{12} k_z^2 - \beta_{32} k_x k_z \right) = 0$$

(5)

Thus, generally Eq. (5) which determines the wave numbers for the outgoing waves is of the fourth order (Wait 1996). The boundary conditions at the upper boundary $z = L_{max}$ within the ionosphere $F$-layer are the absence of the ingoing waves, i.e. the outgoing (radiated) waves are present only. Two roots should be selected that possess the negative imaginary parts $Im(k_{z1, z2}) < 0$, i.e. the outgoing waves dissipate upwards. However, in the case of VLF waves some simplification can be used. Namely, the expressions for the wave numbers $k_{1,2}$ are obtained from eqs. (3), where the dependence on $x$ is neglected: $|k_{1,2}| >> k_0$. This approximation is valid within $F$-layer where the first outgoing wave corresponds to the whistler of small dissipation, the second one to the highly dissipating slow wave. The EM field components, which are necessary to formulate the boundary conditions for eqs. (3a, b) at $z \geq L_{max}$, can be presented as:

$$H_x = A_1 e^{-ik_{z1} \tilde{z}} + \alpha_2 A_2 e^{-ik_{z2} \tilde{z}}, \quad H_y = \alpha_1 A_1 e^{-ik_{z1} \tilde{z}} + A_2 e^{-ik_{z2} \tilde{z}}$$

(6)

In the relations (6), $\tilde{z} = z - L_z$. eqs. (3) are simplified there in the approximation described above:

$$\beta_{22} \frac{\partial^2 H_x}{\partial z^2} - \beta_{21} \frac{\partial^2 H_y}{\partial z^2} + k_0^2 H_x = 0, \quad \beta_{11} \frac{\partial^2 H_y}{\partial z^2} - \beta_{12} \frac{\partial^2 H_x}{\partial z^2} + k_0^2 H_y = 0$$

(7)

Again, the solution of eqs. (7) is searched as: $H_{x,y} \sim e^{-ik_z \tilde{z}}$. The following equation has been obtained to get the wave numbers $k_{z1, z2}$ from eqs. (7):

$$\kappa^4 - (\beta_{22} + \beta_{11})\kappa^2 + \beta_{11}\beta_{22} - \beta_{12}\beta_{21} = 0, \quad \kappa^2 = \frac{k_0^2}{k_z^2}$$

(8)

Therefore, as it follows from eq. (8),

$$\kappa_{1,2}^2 = \frac{\beta_{11} + \beta_{22}}{2} \pm \left( (\frac{\beta_{11} - \beta_{22}}{2})^2 + \beta_{12}\beta_{21} \right)^{1/2}; \alpha_1 = \frac{\beta_{22} - \kappa_1^2}{\beta_{21}} = \frac{\beta_{12}}{\beta_{11} - \kappa_1^2}; \alpha_2 = \frac{\beta_{11} - \kappa_2^2}{\beta_{12}} = \frac{\beta_{21}}{\beta_{22} - \kappa_2^2}; k_{z1,z2}^2 = \frac{k_0^2}{\kappa_{1,2}^2}$$

(9)

The signs of $k_{z1, z2}$ have been chosen from the condition $Im(k_{z1, z2}) < 0$. From Eqs. (5) at the upper boundary $z = L_{max}$ the following relations are valid:

$$H_x = A_1 + \alpha_2 A_2, \quad H_y = \alpha_1 A_1 + A_2$$

(10)

As it follows from eq. (10),

$$A_1 = \Delta^{-1}(H_x - \alpha_2 H_y); A_2 = \Delta^{-1}(H_y - \alpha_1 H_x); \Delta = 1 - \alpha_1 \alpha_2$$

(11)

Thus, it is possible to exclude the amplitudes of the outgoing waves $A_{1,2}$ from Eqs. (9). As a result, at $z = L_{max}$ the boundary conditions are rewritten in terms of $H_x$, $H_y$ only:

$$\frac{\partial H_x}{\partial z} = -i(k_{z1}A_1 + k_{z2}\alpha_2 A_2) = -\frac{i}{\Delta}\left((k_{1z} - \alpha_1\alpha_2 k_{z2})H_x + \alpha_2(k_{z2} - k_{z1})H_y\right)$$

$$\frac{\partial H_y}{\partial z} = -i(k_{z1}\alpha_1 A_1 + k_{z2}A_2) = -\frac{i}{\Delta}\left((k_{z2} - \alpha_1\alpha_2 k_{z1})H_y + \alpha_1(k_{z1} - k_{z2})H_x\right)$$

(12)

The relations (12) are the upper boundary conditions of the radiation for the boundary $z=L_{max}\sim(250\text{-}400)$ km. Then these conditions will be transformed/recalculated using the analytical-numerical recurrent procedure into equivalent impedance boundary conditions at $z=L_z\sim (70\text{-}90)$ km.

Note that in the "whistler/VLF approximation", valid at frequencies ~ 10 kHz, one can get for the F region of the ionosphere. In this approximation and accounting for that $k_x \approx 0$, we find, using eqs. (5), (8), (9) that dispersion equation takes the form

$$k_z'^2 k^2 = k_0^2 g^2$$

(13)

where $k^2 = k_x^2 + k_z^2 = k_x'^2 + k_z'^2$; $k_x'$ and $k_z'$ are the components of wave number, respectively, transverse and longitudinal relatively to geomagnetic field. For the F region of the ionosphere, where $v_e \ll \omega \ll \omega_{He}$, eq. (13) reduces to the standard form of whistler dispersion equation $|k_z'| k = k_0 |g|$; $g \approx -\omega_{pe}^2/(\omega\omega_{He})$; $\omega = c^2 k |k_z'| (\omega_{He}/\omega_{pe}^2)$; in a special case of the waves, propagating exactly along geomagnetic field, $k_x' = 0$ one obtain, for the propagating whistler waves, well-known dispersion dependence (Artcimovich and Sagdeev, 1979) $\omega = c^2 k_z'^2 (\omega_{He}/\omega_{pe}^2)$. Coming back to our problem and accounting for that in our case we can reasonably put $k_x \approx 0$, eq. (13) reduces to $k_z^4 \cos^2\theta = k_0^4 g^2$. As a result, we get $k_{z1} = \sqrt{g/\cos\theta}k_0$, $k_{z2} = -i\sqrt{g/\cos\theta}k_0$, and then, similarly to the relations (12), the boundary conditions can be presented, in terms of the tangential components of electric field, particularly in the form:

$$\frac{\partial \vec{U}}{\partial z} + \hat{B}\vec{U} = 0 \; ; \vec{U} = \begin{bmatrix} E_x \\ E_y \end{bmatrix}; \hat{B} = \frac{1}{2}\sqrt{\frac{g}{\cos\theta}}k_0 \begin{bmatrix} 1+i & -1-i \\ 1+i & 1+i \end{bmatrix}$$

(14)

Conditions (12) or (14) are the conditions of radiation (absence of ingoing waves) formulated at the upper boundary $z=L_{max}$ and suitable for the determination of the energy of the wave leaking from the WGEI into the upper ionosphere/magnetosphere. Let us emphasize again that the formulas expressing the boundary conditions of the radiation (more accurately speaking, an absence of incoming waves, what is the consequence to the causality principle) (12), (14) are obtained as a result of limiting pass by the small parameter $k_x/k_0 |k_x/k_z| \to 0$ in eq. (5). Note that in spite of disappearance of the dependence of these boundary conditions explicitly on $k_x$, the dependence of the characteristics of the wave propagation process on $k_x$, as a whole, is accounted for, and all results are still valid for the description of the wave beam propagation in the WGEI along the horizontal axis $x$ with finite $k_x \sim k_0$.

**3.3 Equivalent Tensor Impedance Boundary Conditions at the Upper Boundary $z=L_z$ of the Effective WGEI**

The tensor impedance at the upper boundary of the effective WGEI $z=L_z$, Fig. 1, is obtained by means of recalculating to the level $z=L_z\sim 80-90$ km of the conditions of radiation (12) or (14), formulated at the upper boundary, placed in the F region of the ionosphere, at $z=L_{max}\sim (250\text{-}400)$ km.

The main idea of the effective tensor impedance method is the unification of the analytical and numerical approaches and the derivation of the proper impedance boundary conditions without any approximation of the "thin cover layer", used in the majority of an effective impedance approaches previously, applied either for artificial or natural layered gyrotropic structures, see, f. e. (Tretyakov, 2003; Senior and Volakis, 1995; Kurushin and Nefedov, 1983; Alperovich and Fedorov, 2007). There is one known exception, namely invariant imbedding impedance method (Shalashov and Gospodchikov, 2010; Kim and Kim, 2016). The comparison of our method with the invariant imbedding impedance method will be presented in the end of this subsection. Eqs. (3), jointly with the boundary conditions (12), have been solved by finite differences. Outline here the main ideas and the steps of the derivations of the corresponding formulas.

The derivatives in Eqs. (3) are approximated as

$$\frac{\partial}{\partial z}\left(C(z)\frac{\partial H_x}{\partial z}\right) \approx \frac{1}{h}\left(C(z_{j+1/2})\frac{(H_x)_{j+1}-(H_x)_j}{h}-C(z_{j-1/2})\frac{(H_x)_j-(H_x)_{j-1}}{h}\right),$$

$$\frac{\partial}{\partial z}\left(F(z)H_x\right) \approx \frac{1}{2h}\left(F(z_{j+1})(H_x)_{j+1}-F(z_{j-1})(H_x)_{j-1}\right) \quad \text{etc.}$$

(15)

In eq. (15), $z_{j+1/2}=h\cdot(j+0.5)$. In eqs. (10) the approximation is $\partial H_x/\partial z \approx [(H_x)_N-(H_x)_{N-1}]/h$. Here $h$ is the discretization step along $OZ$ axis; $N$ is the total number of the nodes. At each step $j$ the difference approximations of Eqs. (3) take the form:

$$\hat{\alpha}_j^{(-)}\cdot\vec{H}_{j-1}+\hat{\alpha}_j^{(0)}\cdot\vec{H}_j+\hat{\alpha}_j^{(+)}\cdot\vec{H}_{j+1}=0$$

(16)

where $\vec{H}_j=\begin{pmatrix}H_x\\H_y\end{pmatrix}$, $j=N-1,N-2,...,1$, $z_j=h\cdot j$, $L_z=h\cdot N$. The expressions for the matrix coefficients in Eq. (16) are complicated; they are given in Appendix. The set of the matrix eqs. (16) has been solved by the method called factorization, or elimination, or matrix sweep (method) [Samarskii, 2001]. Namely, it is possible to write down:

$$\vec{H}_j=\hat{b}_j\cdot\vec{H}_{j-1}, \quad j=N,...,1$$

(17a)

$$H_{xj+1}=b_{11\,j+1}H_1+b_{12\,j+1}H_2; \quad H_{yj+1}=b_{21\,j+1}H_1+b_{22\,j+1}H_2; \quad H_1 \equiv H_{xj}; \quad H_2 \equiv H_{yj}$$

(17b)

This method is in fact a variant of the Gauss elimination method for the matrix 3-diagonal set of the Eqs. (16). The value of $\hat{b}_N$ has been obtained from the boundary conditions (12). Namely, they can be rewritten as:

$$\hat{\alpha}_N^{(-)} \cdot \vec{H}_{N-1} + \hat{\alpha}_N^{(0)} \cdot \vec{H}_N = 0 \tag{18}$$

Therefore $\hat{b}_N = -(\hat{\alpha}_N^{(0)})^{-1} \cdot \hat{\alpha}_N^{(-)}$. Then the matrices $\hat{b}_j$ have been computed sequentially down until the desired value of z

= $=L_z = h \cdot N_z$, where the impedance boundary conditions are assumed to be applied. At each step the formulas for $\hat{b}_j$ follow

from (16), (17) and take the form

$$(\hat{\alpha}_j^{(0)} + \hat{\alpha}_j^{(+)} \cdot \hat{b}_{j+1}) \cdot \vec{H}_j = -\hat{\alpha}_j^{(-)} \cdot \vec{H}_{j-1} \tag{19}$$

Therefore, accounting for (17), we obtain $\hat{b}_j = -(\hat{\alpha}_j^{(0)} + \hat{\alpha}_j^{(+)} \cdot \hat{b}_{j+1})^{-1} \cdot \hat{\alpha}_j^{(-)}$. The derivatives in eqs. (4) have been

approximated as:

$$\left(\frac{\partial H_x}{\partial z}\right)_{N_z} \approx \frac{(H_x)_{N_z+1} - (H_x)_{N_z}}{h} = \frac{(b_{N_z+1\ 11} - 1) \cdot (H_x)_{N_z} + b_{N_z+1\ 12} \cdot (H_y)_{N_z}}{h} \text{ ; analogously for } \left(\frac{\partial H_y}{\partial z}\right)_{N_z} \tag{20}$$

Note that a result of this discretization, only the values at the grid level $N_z$ are included into the numerical approximation of

the derivatives $\partial H_{x,y} / \partial z$ at $z = L_z$. We determine tensor impedance $\hat{Z}$ in particular at $z=L_z \sim 85$ km such as the tensor

value containing in the following relations, all of which are related to the corresponding altitude (in other words, to the grid

with number $N_z$, corresponding to this altitude):

$$\vec{n} \times \vec{E} = \hat{Z} \cdot \vec{H}, \ \vec{n} = (0,0,1); \text{ or } E_x = Z_{21} H_x + Z_{22} H_y; \ E_y = -Z_{11} H_x - Z_{12} H_y \tag{21}$$

The equivalent tensor impedance is obtained, in fact, using two-step procedure. (1) We obtain the matrix $\hat{b}_j$ using the set of

equations (3a, b) with the boundary conditions (12) and the procedure (17)-(19) described above. (2) Put the expressions

(21) with tensor impedance into the left parts and the derivatives $\partial H_{x,y} / \partial z$ in the form (20) into the right parts of eqs. (4).

Equating in the left and right parts of the two obtained equations coefficients at $H_x$, $H_y$ respectively, we obtain the analytical

expressions for the components of the tensor impedance at $z = L_z$:

$$Z_{11} = -\frac{i}{k_0 h}\left(\frac{\beta_{21}}{1 - \beta_{22}\frac{k_x^2}{k_0^2}} \cdot b_{21} - \frac{\beta_{22}}{1 - \beta_{22}\frac{k_x^2}{k_0^2}} \cdot (b_{11} - 1)\right), \ Z_{12} = -\frac{i}{k_0 h}\left(\frac{\beta_{21}}{1 - \beta_{22}\frac{k_x^2}{k_0^2}}\frac{\partial H_y}{\partial z} \cdot (b_{22} - 1) - \frac{\beta_{22}}{1 - \beta_{22}\frac{k_x^2}{k_0^2}} \cdot b_{12} - k_x h \cdot \frac{\beta_{23}}{1 - \beta_{22}\frac{k_x^2}{k_0^2}}\right),$$

$$Z_{21} = \frac{i}{k_0 h}\left((\beta_{11} + \frac{k_x^2}{k_0^2}\frac{\beta_{12} \cdot \beta_{21}}{1 - \beta_{22}\frac{k_x^2}{k_0^2}}) \cdot b_{21} - \frac{\beta_{12}}{1 - \beta_{22}\frac{k_x^2}{k_0^2}} \cdot (b_{11} - 1)\right),$$

$$Z_{22} = \frac{i}{k_0 h}\left( (\beta_{11} + \frac{k_x^2}{k_0^2}\frac{\beta_{12}\cdot\beta_{21}}{1-\beta_{22}\frac{k_x^2}{k_0^2}})\cdot(b_{22}-1) - k_x h\cdot(\beta_{13} + \frac{k_x^2}{k_0^2}\frac{\beta_{12}\cdot\beta_{23}}{1-\beta_{22}\frac{k_x^2}{k_0^2}}) - \frac{\beta_{12}}{1-\beta_{22}\frac{k_x^2}{k_0^2}}\cdot b_{12}\right) \qquad (22)$$

The proposed method of the transfer of the boundary conditions from the ionosphere $F$-layer $L_{max} = 250 – 400$ km into the lower part of the $E$-layer $L_z = 80 – 90$ km is stable and easily realizable, when compared with some alternative approaches based on the invariant imbedding methods (Shalashov and Gospodchikov, 2010; Kim and Kim, 2016). The stability of our method is due to the stability of the Gauss elimination method when the coefficients at the central diagonal are dominating; the last is valid for the ionosphere with electromagnetic losses where the absolute values of the permittivity tensor are big. The application of the proposed method of the matrix sweep in the media without losses may require the utilization of the Gauss method with the choice of the maximum element, to ensure the stability. However, as our simulations (not presented here) demonstrated, for the electromagnetic problems in the frequency domain the simple Gauss elimination and one with the choice of the maximal element gives the same results. The accumulation of errors may occur in evolutional problems in the time domain, when the Gauss method should be applied sequentially many times. The use of the independent functions $H_x$, $H_y$ in Eqs. (3) seems natural, as well as the transfer (17a), because the impedance conditions are the expressions of the electric $E_x$, $E_y$ through these magnetic components $H_x$, $H_y$ at the upper boundary of the VLF waveguide $80 – 90$ km. The naturally chosen direction of the recalculation of the upper boundary conditions from $z=L_{max}$ to $z=L_z$, i.e. from upper layer with large impedance value to lower altitude layer with relatively small impedance value, provides, at the same time, the stability of the simulation procedure. The obtained components of the tensor impedance are small, $|Z_{\alpha\beta}| \leq 0.1$. This determines the choice of the upper boundary $z=L_z$ of the effective WGEI. Due to small enough impedance, EM waves incident from below on this boundary reflect effectively back. Therefore, the region $0 \leq z \leq L_z$ indeed can be presented as an effective WGEI. Then such a waveguide includes not only lower up to $L_{ISO} \sim (65-75)$ km with rather small losses, but also thin dissipative and anisotropic/gyrotropic layer between 75 and 85-90 km.

Finally, the main differences and at the same time advantages of the proposed tensor impedance method from the known method of the impedance recalculating, in particular invariant imbedding methods (Shalashov and Gospodchikov, 2010; Kim and Kim, 2016) are the following. (i) In distinction to invariant imbedding method, our method is a direct method of the recalculation of tensor impedance, and the corresponding tensor impedance is determined analytically, see eqs. (22). (ii) Our method, for the media without non-locality, does not need a solution of integral equation(s), as in invariant imbedding method. (iii) The proposed tensor impedance method does not need the revealing of forward and reflected waves. Moreover, even the conditions of the radiation (12) at the upper boundary $z=L_{max}$ are determined through the total field components $H_{x,y}$, what makes the proposed procedure technically much less cumbersome and practically much more convenient. (iv) At the same time, the procedure is very effective and computationally stable, as it is explained above in this subsection. As it is already mentioned, for the very low-loss systems, the required level of stability can be achieved with the modification based on the choice of the maximal element for matrix inversion.

**3.4 Propagation of the Electromagnetic Waves in the Gyrotropic Waveguide and the TIMEB Method**

Use, as the independent functions, the transverse components $E_y$, $H_y$. The goal is to derive the equations for the slowly varying amplitudes $A(x,y,z)$, $B(x,y,z)$ of the VLF beams included corresponding field components:

$$E_y = \frac{1}{2} A(x,y,z) \cdot e^{i\omega t - ik_0 x} + c.c., \quad H_y = \frac{1}{2} B(x,y,z) \cdot e^{i\omega t - ik_0 x} + c.c. \tag{23}$$

5 Note that in this case it should be $k_x = k_0$, because the beam propagates in the WGEI, the main part of which is occupied by the atmosphere and lower ionosphere (D region), rather closed to free space by its electromagnetic parameters. The presence of a thin anisotropic and dissipative layer belonging to the E region of the ionosphere causes, altogether with the impedance boundary condition the proper $z$ dependence of $B(x,y,z)$. Using (21), (22), it is possible to write down the boundary conditions at the height $z = L_z$ for the slowly varying amplitudes $A(x,y,z)$, $B(x,y,z)$ of the transverse components $E_y$, $H_y$.

10 Namely, from the Maxwell equations in the method of beams it is possible to express the components $E_x$ and $H_x$ through $E_y$, $H_y$:

$$H_x \approx -\frac{i}{k_0} \frac{\partial E_y}{\partial z}, \quad E_x \approx \gamma_{12} E_y + i \frac{\tilde{\beta}_{33}}{k_0} \frac{\partial H_y}{\partial z} + \tilde{\beta}_{13} H_y \tag{24}$$

where $\gamma_{12} = \Delta_0^{-1}(\varepsilon_{13}\varepsilon_{32} - \varepsilon_{12}\varepsilon_{33})$, $\tilde{\beta}_{13} = \Delta_0^{-1}\varepsilon_{13}$, $\tilde{\beta}_{33} = \Delta_0^{-1}\varepsilon_{33}$; $\Delta_0 = \varepsilon_{11}\varepsilon_{33} - \varepsilon_{13}\varepsilon_{31}$. The using of Eqs. (21) and (24) leads to the following form of the boundary conditions for $A$, $B$:

$$A - \frac{i}{k_0} Z_{11} \cdot \frac{\partial A}{\partial z} + Z_{12} \cdot B \approx 0, \quad \gamma_{12} \cdot A + \frac{i}{k_0} Z_{21} \cdot \frac{\partial A}{\partial z} + (\tilde{\beta}_{13} - Z_{22}) \cdot B + \frac{i}{k_0} \tilde{\beta}_{33} \cdot \frac{\partial B}{\partial z} \approx 0 \tag{25}$$

Let us derive the evolution equations for the slowly varying amplitudes $A(x,y,z)$, $B(x,y,z)$ of the VLF beams. Below the monochromatic beams are considered, so the frequency $\omega$ is fixed and the amplitudes do not depend on time $t$. Search the solutions for the EM field as $\vec{E}, \vec{H} \sim \exp(i\omega t - ik_x x - ik_y y)$. The Maxwell equations are written down as

$$-ik_y H_z - \frac{\partial H_y}{\partial z} = ik_0 D_x, \quad \frac{\partial H_x}{\partial z} + ik_x H_z = ik_0 D_y, \quad -ik_x H_y + ik_y H_x = ik_0 D_z$$

$$-ik_y E_z - \frac{\partial E_y}{\partial z} = -ik_0 H_x, \quad \frac{\partial E_x}{\partial z} + ik_x E_z = -ik_0 H_y, \quad -ik_x E_y + ik_y E_x = -ik_0 H_z \tag{26}$$

20 Here $D_x = \varepsilon_{11} E_x + \varepsilon_{12} E_y + \varepsilon_{13} E_z$. From Eqs. (21), it is possible to get the expressions for $E_x$, $E_z$ through $E_y$, $H_y$:

$$E_x = \frac{1}{\Delta}\left\{[\varepsilon_{13} \cdot \varepsilon_{32} - (\varepsilon_{12} + \frac{k_x k_y}{k_0^2}) \cdot (\varepsilon_{33} - \frac{k_y^2}{k_0^2})]E_y + \frac{i}{k_0}(\varepsilon_{33} - \frac{k_y^2}{k_0^2})\frac{\partial H_y}{\partial z} + \frac{k_x}{k_0}\varepsilon_{13} \cdot H_y + \frac{ik_y}{k_0^2}\varepsilon_{13}\frac{\partial E_y}{\partial z}\right\}$$

$$E_z = \frac{1}{\Delta}\left\{[\varepsilon_{31} \cdot (\varepsilon_{12} + \frac{k_x k_y}{k_0^2}) - \varepsilon_{32} \cdot (\varepsilon_{11} - \frac{k_y^2}{k_0^2})]E_y - \frac{i}{k_0}\varepsilon_{31}\frac{\partial H_y}{\partial z} - \frac{k_x}{k_0} \cdot (\varepsilon_{11} - \frac{k_y^2}{k_0^2})H_y - \frac{ik_y}{k_0^2} \cdot (\varepsilon_{11} - \frac{k_y^2}{k_0^2})\frac{\partial E_y}{\partial z}\right\} \tag{27}$$

In eq. (27), $\Delta \equiv (\varepsilon_{11} - \frac{k_y^2}{k_0^2}) \cdot (\varepsilon_{33} - \frac{k_y^2}{k_0^2}) - \varepsilon_{31} \cdot \varepsilon_{13}$. The equations for $E_y$, $H_y$ obtained from the Maxwell equations are:

$$\left(\frac{\partial^2}{\partial z^2} - k_x^{\,2} - k_y^{\,2}\right)E_y + ik_y\left(\frac{\partial E_z}{\partial z} - ik_x E_x - ik_y E_y\right) + k_0^{\,2}D_y = 0; \quad -ik_0\frac{\partial E_x}{\partial z} + k_x k_0 E_z + k_0^{\,2}H_y = 0 \tag{28}$$

After substitution of expressions (27) for for $E_x$, $E_z$ into Eqs. (28) the coupled equations for $E_y$, $H_y$ only can be got. Namely, the expansion should be used: $k_x = k_0 + \delta k_x$, $/\delta k_x/ \ll k_0$, also $/k_y/ \ll k_0$. Then the correspondence should be applied (Weiland and Wilhelmsson 1977):

$$-i\cdot\delta k_x \to \frac{\partial}{\partial x}, \quad -i\cdot k_y \to \frac{\partial}{\partial y} \tag{29}$$

The expansions should be until the quadratic terms with respect to $k_y$ and the linear terms with respect to $\delta k_x$. As a result, the parabolic equations (Levy 2000) for the slowly varying amplitudes $A$ and $B$ have been derived. In the atmosphere and the lower ionosphere, where the effective permittivity reduces to a scalar $\varepsilon(\omega, z)$, they are independent:

$$\frac{\partial A}{\partial x} + \frac{i}{2k_0}\left(\frac{\partial^2 A}{\partial y^2} + \frac{\partial^2 A}{\partial z^2}\right) + \frac{ik_0}{2}\cdot(\varepsilon - 1)A = 0$$
$$\frac{\partial B}{\partial x} + \frac{i}{2k_0}\left(\frac{1}{\beta}\frac{\partial}{\partial z}(\beta\frac{\partial B}{\partial z}) + \frac{\partial^2 B}{\partial y^2}\right) + \frac{ik_0}{2}\cdot(\varepsilon - 1)B = 0 \tag{30a}$$

Here $\beta \equiv \varepsilon^{-1}$. Accounting for the presence of gyrotropic layer near the and the presence of tensor impedance boundary conditions at the upper boundary $z = L_z$ of the VLF waveguide, the equations for the slowly varying amplitudes in general case are coupled and possess a complicated form:

$$\frac{\partial A}{\partial x} + \frac{i}{2k_0}\left(\frac{\partial^2 A}{\partial y^2} + \frac{\partial^2 A}{\partial z^2}\right) + \frac{ik_0}{2}\cdot(\tilde{\varepsilon}_{22} - 1)\cdot A + \frac{\gamma_{21}}{2}\frac{\partial B}{\partial z} + \frac{ik_0}{2}\cdot\gamma_{23}B = 0$$
$$\frac{\partial B}{\partial x} + \frac{i}{2k_0}\left(\frac{1}{\tilde{\beta}_{11}}\frac{\partial}{\partial z}(\tilde{\beta}_{33}\frac{\partial B}{\partial z}) + \frac{\partial^2 B}{\partial y^2}\right) + \frac{i}{2\tilde{\beta}_{11}}\frac{\partial}{\partial z}(\gamma_{12}A) + \frac{1}{2\tilde{\beta}_{11}}\frac{\partial}{\partial z}(\tilde{\beta}_{13}B) + \frac{ik_0}{2\tilde{\beta}_{11}}\gamma_{32}A + \frac{\tilde{\beta}_{31}}{2\tilde{\beta}_{11}}\frac{\partial B}{\partial z} + \frac{ik_0}{2}\cdot(\frac{1}{\tilde{\beta}_{11}} - 1)\cdot B = 0 \tag{30b}$$

In Eq. (30b),

$$\gamma_{12} \equiv \frac{\varepsilon_{13}\cdot\varepsilon_{32} - \varepsilon_{12}\cdot\varepsilon_{33}}{\Delta}, \gamma_{21} \equiv \frac{\varepsilon_{23}\cdot\varepsilon_{31} - \varepsilon_{21}\cdot\varepsilon_{33}}{\Delta}, \gamma_{23} \equiv \frac{\varepsilon_{21}\cdot\varepsilon_{13} - \varepsilon_{23}\cdot\varepsilon_{11}}{\Delta}, \gamma_{32} \equiv \frac{\varepsilon_{31}\cdot\varepsilon_{12} - \varepsilon_{32}\cdot\varepsilon_{11}}{\Delta}, \tilde{\beta}_{11} \equiv \frac{\varepsilon_{11}}{\Delta}, \tilde{\beta}_{13} \equiv \frac{\varepsilon_{13}}{\Delta},$$

$$\tilde{\beta}_{31} \equiv \frac{\varepsilon_{31}}{\Delta}, \tilde{\beta}_{33} \equiv \frac{\varepsilon_{33}}{\Delta}; \Delta \equiv \varepsilon_{11}\cdot\varepsilon_{33} - \varepsilon_{13}\cdot\varepsilon_{31}.$$

Eqs. (30b) reduce to Eqs. (30a) when the effective permittivity is scalar. At the Earth's surface $z = 0$ the impedance conditions reduce, due to a finite conductivity of the Earth, to the form:

$$E_y = Z_E H_x, \quad E_x = -Z_E H_y, \quad Z_E \equiv \left(\frac{i\omega}{4\pi\sigma_E}\right)^{1/2} \tag{31a}$$

Here $\sigma_E \sim 10^8$ s$^{-1}$ is the Earth's conductivity. The boundary conditions (31a) at the Earth's surface, where $Z_{22} = Z_{21} \equiv Z_E$, $Z_{12} = Z_{21} = 0$, $\beta_{33} = \varepsilon(z = 0)^{-1}$, $\gamma_{12} = 0$, $\tilde{\beta}_{13} = 0$, can be rewritten as

$$E_y + \frac{i}{k_0} Z_E \frac{\partial E_y}{\partial z} = 0, \quad \frac{i}{\varepsilon(z=0)k_0} \frac{\partial H_y}{\partial z} + \frac{i}{k_0} Z_E H_y = 0 \tag{31b}$$

5   The Eqs. (30) added by the boundary conditions (25) at the upper boundary of the VLF waveguide $z=L_z$ and by the boundary conditions at the Earth's surface (31b) are used below to simulate the VLF wave propagation. The surface impedance of the Earth has been calculated from the Earth's conductivity. The initial conditions to the problem (30), (25), (31b) are

$$A(x=0, y, z) = 0, \ B(x=0, y, z) = B_0 exp\left(-\left((y-0.5L_y)/y_0\right)^{2n}\right) exp\left(-\left((z-z_1)/z_0\right)^{2n}\right), \ n = 2 \tag{32}$$

10   The size of the computing region along $OY$ axis is $L_y \sim 1000$ km. Because the gyrotropic layer is relatively thin and is placed at the upper part of the VLF waveguide, whereas the beams are excited near the Earth's surface, the wave diffraction in this gyrotropic layer along $OY$ axis is quite small, i.e. the terms with $\partial^2 A/dy^2$, $\partial^2 B/dy^2$ are small there. Contrary to this, the wave diffraction is very important in the atmosphere in the lower part of the VLF waveguide near the Earth's surface. To solve the problem of the beam propagation, the method of splitting with respect to physical factors has been applied (Samarskii 2001).

15   Namely, the problem has been approximated by the finite differences:

$$\vec{C} \equiv \begin{pmatrix} A \\ B \end{pmatrix}, \quad \frac{\partial \vec{C}}{\partial x} + \hat{L}_y \vec{C} + \hat{L}_z \vec{C} = 0 \tag{33}$$

In the terms $\hat{L}_y \vec{C}$, the derivatives with respect to $y$ are included, whereas all another terms are included into $\hat{L}_z \vec{C}$. Then the following fractional steps have been applied, the first one is along y, the second one is along $z$:

20   $$\frac{\vec{C}^{p+1/2} - \vec{C}^p}{h_x} + \hat{L}_y \vec{C}^{p+1/2} = 0, \quad \frac{\vec{C}^{p+1} - \vec{C}^{p+1/2}}{h_x} + \hat{L}_z \vec{C}^{p+1} = 0 \tag{34}$$

The region of simulation is $0 < x < Lx = 1000 - 2000$ km, $0 < y < L_y = 2000 - 3000$ km, $0 < z < L_z = 80 - 90$ km. The numerical scheme (34) is absolutely stable. Here $h_x$ is the step along $OX$ axis, $x_p = p\,h_x$, $p = 0, 1, 2,\ldots$. This step has been chosen from the condition of the independence of the simulation results on diminishing $h_x$.

[Figure]

**Figure 2. The rotation of the local Cartesian coordinate frame at each step along the Earth's surface $h_x$. on a small angle $\delta\varphi \approx$ $\approx \Delta x/R_E$, radians, while $\Delta x = h_x$. The following strong inequalities are valid $h_x << L_z << R_E$. The Earth's surface is at $Z = 0$.**

Under the simulations at each step along $OX$ axis, the correction due to the Earth's curvature has been inserted in adiabatic manner, namely the rotation of the local coordinate frame $XOZ$. Because the step along $x$ is small $h_x \sim 1$ km $<< L_i$, this correction of the $\vec{C}$ results in the multiplier $exp(-ik_0 \cdot \delta x)$, where $\delta x = z \cdot (h_x/R_E)$, $R_E >> L_i$ is the Earth's radius, see Fig. 2 and the capture to this figure. At the distances $x \leq 1000$ km, the results of simulations do not depend on the insertion of this correction, whereas at higher distances some quantitative differences occur. Namely, the VLF beam propagates more closely to the upper boundary of the waveguide.

**3.5 The Modes of the VLF Waveguide. Reflection from the Upper Effective Boundary of VLF Waveguide.**

Our model, in general, needs the consideration of the excitations of the waveguide modes by means of current sources such as dipole-like VLF radio source and lightning discharge. Then, the reflection of the waves incident on the upper boundary ($z=L_z$) of the effective WGEI can be considered. There will be possible to demonstrate that this structure has indeed good enough waveguiding properties. Then, in the model described in the present paper, the VLF beam is postulated already on the input of the system. To understand, how such a beam is excited by the, say, dipole antenna near the lower boundary $z=0$ of the WGEI, the formation of the beam structure based on the mode presentation should be searched. Then the conditions of the radiation (absence of ingoing waves) (12) can be used as the boundary conditions for the VLF beam radiated to the upper ionosphere/magnetosphere. Due to relatively large scale of the inhomogeneity in this region, the complex geometrical optics (Rapoport et al. 2014) would be quite suitable for the modeling a beam propagation, even accounting for the wave dispersion in magnetized plasma. The proper effective boundary condition, similarly to (Rapoport et al. 2014) would allow to make relatively accurate matching between the regions, described by means of full wave electromagnetic approach (with Maxwell equations) and complex geometrical optics (FWEM-CGO approach). All of these material is not included into the **present** paper, but will be delivered in the two future papers, one of which will be dedicated to the mode **presentation** of the VLF propagation in the WGEI, and the other one - to the leakage of VLF beams from the WGEI into upper ionosphere and magnetosphere and the propagation in these media. However, we should mention in the present paper only one result, which

concerns the mode excitation in the WGEI, because this result is principally important for the justification of TIMEB method. Namely, it is shown that the $\geq 5$ lowest modes of the WGEI are strongly localized in the atmosphere-lower ionosphere. Their longitudinal wavenumbers are close to the corresponding wavenumbers of EM waves in the atmosphere. This fact convinces that the TIMEB method can be applied to the propagation of VLF electromagnetic waves in the WGEI.

**4. The Results of Modeling**

The dependencies of the components of the permittivity $\varepsilon_l$, $\varepsilon_3$, $\varepsilon_h$ in the coordinate frame associated with the geomagnetic field $\vec{H}_0$ are given in Fig. 3. The typical results of simulations are presented in Fig. 4. The parameters of the ionosphere correspond to Fig. 3. The inclination of the geomagnetic field is 45°. The VLF frequency is $\omega = 10^5$ s$^{-1}$, $f = \omega/2\pi \approx 15.9$ kHz. The Earth's surface is assumed as ideally conductive here: $Z = 0$. The values of EM field are given in absolute units, i.e. the magnetic field is measured in Oersteds (Oe), or Gauss (Gs), 1 Gs = $10^{-4}$ T, whereas the electric field is also in Gs, 1 Gs = 300 V/cm there.

Note that in the absolute (Gaussian) units the magnitudes of the magnetic field component $/H_y/$ are the same as ones of the electric field component $|E_z|$ in the atmosphere region where the permittivity is $\varepsilon \approx 1$. Below in the figure captions the correspondence between the absolute units and practical SI ones is given.

It is seen that the absolute values of the components of the permittivity increase sharply just above $z = 75$ km. The behavior of the components of the permittivity is step-like, as seen from Fig. 3_2, a. Due to this, the results of simulations are tolerant to the choice of the position of the upper wall of the waveguide the Earth's surface – ionosphere. The computed components of the tensor impedance at $z = 85$ km are: $Z_{11} = 0.087 + i0.097$, $Z_{21} = 0.085 + i0.063$, $Z_{12} = -0.083 - i0.094$, $Z_{22} = 0.093 + i0.98$. So, a condition $/Z_{\alpha\beta}/ \leq 0.15$ is satisfied there, which is necessary for applicability of the boundary conditions (3).

The maximum value of the $H_y$ component is 0.1 Oe = $10^{-5}$ T in Fig. 3, a) for the initial VLF beam at $x = 0$. This corresponds to the value of $E_z$ component of 0.1 Gs = 30 V/cm. At the distance $x = 1000$ km the magnitudes of the magnetic field $H_y$ are of about $3 \cdot 10^{-5}$ Oe = 3 nT, whereas ones of the electric field $E_y$ are of about $3 \cdot 10^{-6}$ Gs $\approx 1$ mV/cm.

[Figure]

[Figure]

| e | f | g |

**Figure 3. Part a) is the vertical dependencies of the components of modules of components of the permittivity in the frame associated with the geomagnetic field $|\varepsilon_l|$, $|\varepsilon_3|$, $|\varepsilon_h|$, the curves 1, 2, 3 correspondingly. Parts b) – g) are the real and imaginary parts of the components $\varepsilon_l$, $\varepsilon_3$, $\varepsilon_h$, general and detailed views.**

[Figure]

**Figure 4. Part a) is the initial distribution of $|H_y|$ at x = 0. Parts b), c) are $|E_y|$ and $|H_y|$ at x = 600 km. Parts d), e) are $|E_y|$ and $|H_y|$ at x = 1000 km. For the electric field it is $3 \cdot 10^{-6}$ Gs $\approx$ 1 mV/cm, for the magnetic field it is $3 \cdot 10^{-5}$ Gs $\approx$ 3 nT. At the altitudes z < 75 km it is $|E_z| \approx |H_y|$, so $3 \cdot 10^{-5}$ Gs $\approx$ 10 mV/cm there. $\omega = 1. \cdot 10^5 c^{-1}$; $\theta = 45°$**

It is seen from Fig. 4, b) - e), that the wave beams are localized within the waveguide the Earth's surface – ionosphere $0 < z < 75$ km mainly in the regions with the isotropic permittivity. The mutual transformations of the beams of different polarizations occur near the upper boundary, due to the anisotropy of the ionosphere within the thin layer 75 km $< z <$ 85 km, Fig. 4, b), d). These transformations depend on the values of the components of the permittivity of the ionosphere at the altitudes $z > 80$ km and the components of the tensor impedance, so the measurements of the phase and amplitude

modulations of different EM components near the Earth's surface can yield the information on the properties of the lower and even middle ionosphere.

The qualitative effect is changing the polarization, i.e. an occurrence of $E_y$ component of the electric field at small altitudes $z$ ~ 5 – 10 km near the Earth's surface, due to these mutual transformations of EM beams, Fig. 4, parts b), de). Note that the case of the ideal conductivity of the Earth's surface is considered here, $Z = 0$, so at $z = 0$ the component is $E_y = 0$. If this impedance is $Z \neq 0$, then $E_y$ component occurs also at the Earth's surface.

The magnitudes of $E_y$ component depend essentially on the values of the electron concentration at the altitudes $z = 75 - 100$ km. In Fig. 5, parts a), b), there are different dependencies of the electron concentration $n(z)$, three curves, solid (1), dash (2) and dot (3) ones. The corresponding dependencies of the absolute values of the components of the permittivity are in Fig. 4, parts c), d).

[Figure]

| a | b | c | d |

**Figure 5. Different profiles of the electron concentrations n used in simulations. The solid, dash, and dot curves correspond to these different profiles. Part a) is the detailed view; b) is a general view. The corresponding profiles of the modules of the components of the permittivity $|\varepsilon_3|$ and $|\varepsilon_h|$ are given in parts c), d).**

The distributions of $|E_y|$, $|H_y|$ on $z$, $y$ at $x = 1000$ km are given in Fig. 6. Parts a), b) correspond to the solid (1) curve $n(z)$ in Fig. 5; parts c), d) are for the dash (2) curve; parts e), f) are for the dot (3) curve in Fig. 5. The initial beams of $H_y$ are the same and are given in Fig. 4, a). The values of the tensor impedance for these three cases are presented in Table 1

**Table 1. The values of the tensor impedance corresponding to the data shown in Fig. 5. Impedances presented in the lines 1,2 and 3 in Table one correspond to the solid (1), dash (2) and dot (3) curves in Figs. 5 a)-d), respectively.**

| $Z_{11}$ | $Z_{21}$ | $Z_{12}$ | $Z_{22}$ |
|---|---|---|---|
| 0.088 + i0.098 | 0.085 + i0.063 | -0.083- i0.094 | 0.093 + i0.098 |
| 0.114+i0.127 | 0.107+i0.079 | -0.105-i0.127 | 0.125+i0.125 |
| 0.067+i0.0715 | 0.061+i0.051 | -0.060-i0.070 | 0.069+i0.072 |

[Figure]

**Figure 6. Parts a), c), e) are dependencies of $|E_y|$, parts b), d), f) are dependencies of $|H_y|$ at x = 1000 km; $\omega = 1. \cdot 10^5 c^{-1}$; $\theta = 45°$. The initial beams are the same as in Fig. 4, a). Parts a), b) corresponds to the solid (1) curves in Fig. 5; parts c), d) are for the dash (2) curves; parts e), f) correspond to the dot (3) curves there. For the electric field it is $3 \cdot 10^{-6}$ Gs ≈ 1 mV/cm, for the magnetic field it is $3 \cdot 10^{-5}$ Gs ≈ 3 nT. At the altitudes z < 75 km it is $|E_z| \approx |H_y|$, so $3 \cdot 10^{-5}$ Gs ≈ 10 mV/cm there.**

[Figure]

**Figure 7. The dependencies of EM components on the altitude z in the center of the waveguide y = 1500 km for the different profiles of the electron concentration. The solid (1), dash (2), and dot (3) curves correspond to the different profiles of the electron concentration in Fig. 5, a), b), the same kinds of curves. For the electric field it is $3 \cdot 10^{-6}$ Gs ≈ 1 mV/cm, for the magnetic field it is $3 \cdot 10^{-5}$ Gs ≈ 3 nT. At the altitudes z < 75 km it is $|E_z| \approx |H_y|$, so $3 \cdot 10^{-5}$ Gs ≈ 10 mV/cm there. $\omega = 1. \cdot 10^5 c^{-1}$; $\theta = 45°$**

The distributions of $|E_y|$, $|H_y|$ on z at x = 1000 km in the center of the waveguide y = 1500 km are given in Fig. 7. This and other (not presented here) simulations show that change in complex tensors of both volume dielectric permittivity and impedances at the lower and upper boundaries of effective WGEI influence remarkably on the VLF losses in the WGEI. The modulation of the electron concentration at the altitudes above z= 120 km affects weakly the excitation of $E_y$ component within the waveguide.

**5. An influence of the parameters on the polarization transformation and losses in the WGEI**

An important effect of the gyrotropy and anisotropy is the corresponding transformation of the field polarization during the propagation in the WGEI, absent in the ideal metal planar waveguide without gyrotropy and anisotropy. We will show that such an effect is quite sensitive to the carrier frequency of the beam, propagating in the WGEI, inclination of the geomagnetic field and perturbations in the electron concentration, which can vary under the influences of the powerful enough sources placed "below", "above" and "inside" the ionosphere. Let us remind that in the real WGEI, the anisotropy and gyrotropy are connected both with the volume effect and effective surface tensor impedances at both the lower and upper surfaces of the effective WGEI, namely $z=0$ and $z=L_z$ (Fig. 1). The corresponding transformation of the field polarization is determined a follows. We introduce the characteristic polarization relation $|E_y / H_y|(z; y = L_y / 2; x = x_0)$, taken at the central plane of the beam ($y=L_y$) at some characteristic distance ($x=x_0$) from the beam input/VLF transmitter. The following arguments justify such a choice of the characteristic polarization parameter ($|E_y/H_y|$) and its dependence on the vertical coordinate ($z$). (1) The WGEI is rather similar to the ideal planar metallized waveguide. This is connected with two circumstances. First, tensor $\hat{\varepsilon}$ is different remarkably from the isotropic one $\hat{I}$ only in the relatively small (upper) part of the WGEI, namely in the altitude range $\Delta z \sim (5 \div 10)$ km (from (75-80) to 85 km, Fig. 1). Second, both the Earth and ionosphere conductivity are quite high and corresponding impedances are quite low. In particular, the elements of the effective tensor impedance at the upper boundary of WGEI are small, $|Z_{\alpha\beta}| \leq 0.1$ (see, for example, the table 1). (2) Respectively, the carrier modes of the VLF beam are closed to the modes of the ideal metallized planar waveguide. These modes are subdivided into the sets of uncoupled ($E_x,H_y,E_z$) and ($H_x,E_y,H_z$) modes. The detail search of the propagation of the separate eigenmodes of the WGEI is not a goal of the present paper, and respectively, will not be presented here and will became a subject of the special paper. (3) Because we have adopted for the initial beam(s) the input boundary conditions in the form (23) (with $H_y \neq 0$, $E_y = 0$), the above mentioned value $|E_y / H_y|(z; y = L_y / 2; x = x_0)$ characterizes the mode coupling and corresponding transformation of the polarization at the distance $x_0$ from the beam input due to the presence of the volume and surface gyrotropy and anisotropy in the real WGEI. The results presented below are obtained for $x_0 = 1000$ km, what is, by the order of value, a typical distance, for example, between the VLF transmitter and receiver of the European VLF/LF radio network (Biagi et al. 2015). Another parameter characterizing the propagation of the beam in the WGEI, Namely effective total loss parameter, is $|H_{ymax}(x=x_0)/ H_{ymax}(x=0)|$. Note that this parameter characterizes both dissipative and diffraction losses connected with beam spreading in the transverse ($y$) direction during the propagation in the WGEI.

In Fig. 8, dependences of the introduced above polarization and loss characteristics are presented and the influences on these parameters of the carrier beam frequency and the inclination of magnetic field (instead of inclination angle we use the angle $\theta$ between the direction of the geomagnetic field and the vertical direction, see Fig. 1).

[Figure]

5 **Figure 8.** The modeling is done for the altitude dependence of the electron concentration shown in Fig. 5b, curve 1. Characteristics of the polarization transformation parameter $|E_y/H_y|$ (a-c) and effective coefficient of the total losses at the distance $x_0=1000\ km$ from the beam input ( d ); (a, b) and ( d ) - dependences of the polarization parameter (a, b) and total losses (d) on the vertical coordinate for different angles $\theta$ between geomagnetic field, respectively; black (1), red (2), green (3) and blue (4) curves in Figs. a, b and d correspond to $\theta$ equals to $5°, 30°, 45°$ and $60°$, respectively; Figs. a and b correspond to frequencies $\omega = 0.86.\cdot10^5 c^{-1}$ and

10 $\omega = 1.14\cdot10^5 c^{-1}$, respectively; ( c ) – dependence of the polarization parameter on the vertical coordinate for the different frequencies; black (1), red (2) and green (3) curves correspond to the frequencies equal to $0.86\cdot10^5$, $1.\cdot10^5$ and $1.14\cdot10^5\ c^{-1}$, respectively and $\theta = 45°$.

As it is seen from Figs. 8 a-c, the altitude dependence of the polarization parameter $|E_y/H_y|$ has two main maxima in the WGEI, the higher of which lies in the gyrotropic region above 70 km, while the other in the isotropic region of the WGEI.

15 As it is seen from Fig. a, b, the value of the lager second maximum increases, while the position of the second maximum shifts to the lower altitudes with increasing frequency. At the higher frequency ( $\omega = 1.14\cdot10^5 c^{-1}$ ), the larger maximum of the polarization parameter corresponds to the intermediate value of the angle $\theta = 45°$ (Fig. 8 b); for the lower frequency ( $\omega = 0.86.\cdot10^5 c^{-1}$ ), the largest value of the first (higher) maximum corresponds to the almost vertical direction of the geomagnetic field ( $\theta = 5°$, Fig. 8 a). For the intermediate value of the angle ( $\theta = 45°$ ), the largest value of the main

20 maximum corresponds to the higher frequency ( $\omega = 1.14\cdot10^5 c^{-1}$ ) in the considered frequency range (Fig. 8 c). The total losses increase monotonically with increasing frequency and depend weakly on the value of $\theta$ (Fig. 8 d).

To model the effect of increasing and decreasing the electron concentration $n_e$ in the lower ionosphere on the polarization parameter, we have used the following parameterization for the change $\Delta n_e = n_e(z) - n_{0e}(z)$ of the electron concentration, where $n_{0e}(z)$ is the unperturbed altitude distribution of the electron concentration:

$$\Delta n_e(z) = n_{0e}(z)\Phi(z)\,;\ \ \Phi(z) = [F(z)] - \frac{(z-z_2)^2}{\Delta z_{12}^2}[F(z_1)] - \frac{(z-z_1)^2}{\Delta z_{12}^2}[F(z_2)]\,;\ \ F(z) = f\cdot ch^{-2}\{[z - (\frac{z_1+z_2}{2})]/\Delta z\} \qquad (35)$$

5  In relations (35), $\Delta z_{12} \equiv z_2 - z_1$ ; $\Delta z$ is the effective width of the altitude distribution of the perturbation of electron concentration; the perturbation $\Delta n_e$ is concentrated in the range of altitudes $z_1 \le z \le z_2$ (and is equal to zero outside this region); $\Delta n_e(z_1) = \Delta n_e(z_2) = 0$, while $\Phi(z_1) = \Phi(z_2) = 0$.

[Figure]

10      a              b              c

**Figure 9.** Electron concentration (a), decreased and increased (curves 2 (red) and 3 (blue), respectively) relatively to reference concentration (curve 1 in Fig. 1b, repeated, for the convenience, as curve 1 (black) in Fig. 9_new a) and corresponding altitude distributions of the polarization parameter $|E_y/H_y|$ (Figs b and c, respectively). Curves 1 (black), 2 (red), and 3 (green) in Figs. b, c correspond to $\omega$ equal to $0.86\cdot10^5 c^{-1}$, $1\cdot10^5 c^{-1}$ and $1.14\cdot10^5 c^{-1}$, respectively;  decreasing concentration (curve 2, Fig. a) and
15  **increasing concentration (curve 3 in Fig. a) correspond to the following parameters determined the parameterization of concentration variation (see formulas (35)):** $z_1$=50 km, $z_2$=90 km, $\Delta z = 20$ km ; **for decreasing concentration (curve 1, Fig. a)  and increasing concentration (curve 3, Fig. a) -** $f = -1.25$ **and** $f = 250$, **respectively. Angle** $\theta$ **is equal to** $45°$ .

As it is seen from Fig. 9, the change in the concentration in the lower ionosphere causes rather nontrivial effect on the parameter of the polarization transformation $|E_y/H_y|$. Note that either increase or decrease in the ionosphere plasma
20  concentration have been reported as a result of seismogenic phenomena, tsunamis, particle precipitation in the ionosphere due to wave-particle interaction in the radiation belts (Pulinets et al. 2005; Shinagawa et al. 2013; Arnoldy et al. 1989; Glukhov et al. 1992; Tolstoy et al. 1986) etc. Namely, this effect does not reduces only to increase (Fig. 9 b) or decrease (Fig. 9 c) of the maximum value of the parameter of the polarization transformation $|E_y/H_y|$. Note also that the corresponding change of this parameter has rather remarkable absolute values from dozens to thousands percent, as it is seen from the
25  comparison between Figs. 9 b, c and Fig. 8 c, curve 3. The last curve corresponds to the unperturbed distribution of the ionospheric electron concentration (see curve 1, Fig. 5b and curve 1 in Fig. 9 a). It is even more interesting that in the case of decreasing  (Fig. 9 a, curve 2 ) electron concentration, the main maximum of $|E_y/H_y|$ occurs in the lower atmosphere (at the altitude around 20 km, Fig. 9 b, curve 3, which corresponds to $\omega = 1.14\cdot10^5 c^{-1}$ ). In the case of increasing electron

concentration (Fig. 9 a, curve 3) the main maximum  of $|E_y/H_y|$ occurs near the E region of the ionosphere (at the altitude around 77 km). The secondary maximum, which is placed, in the absence of the perturbation of the electron concentration, in the lower atmosphere (Fig. 8 c, curves 2, 3), or mesosphere/ionosphere D region ((Fig. 8 c, curve 1), practically disappears or just is not seen in the present scale, in the case under consideration (Fig. 9 c, curves 1-3).

[Figure]

**a**                **b**                **c**

**Figure 10. The effects connected with surface impedances at the upper and lower boundaries of the WGEI. (a, b) - Frequency dependences of the real (a) and imaginary (b) parts of the element Z11 of the effective tensor impedance at the upper boundary (z=Lz, see Fig. 1) of the WGEI. Curves 1 (black), 2 (red), 3 (blue) and 4 (green) correspond to angle $\theta$ equal to $5°, 30°, 45°$ and $60°$ ,**

10 **respectively. The Earth conductivity is supposed to be infinite (the surface impedance at the lower boundary of WGEI is zero). ( c ) – The vertical coordinate dependencies of transformation polarization parameter |Ey/Hy| at the frequency $\omega = 0.86 \cdot 10^5$ c$^{-1}$ and angle $\theta = 45°$ for the (scalar/isotropic) surface impedance Z at the lower surface of the WGEI equal to 10-4 (Earth conductivity $\sigma$ equal to 109 c-1) , curve 1 and Z =10-2 ( $\sigma = 10^7 \text{c}^{-1}$ ), curve 2.**

15       As it is seen from Fig. 10, the real (a) and imaginary (b) parts of the surface impedance at the upper boundary of the WGEI have a quasiperiodical character with the amplitude of "oscillations" occurring around some effective average values (not shown explicitly in Figs. 10 a, b) decreases with increasing the angle $\theta$ . Even without the determination of the exact average values for each of the curves 1-4 in Figs. 10 a, b, it is seen that corresponding average values of $Re(Z_{11})$ and $Im(Z_{11})$, in general, decrease with increasing angle $\theta$ . It is also seen that average values of $Re(Z_{11})$ for $\theta$ equal to $5°, 30°, 45°$ and $60°$ (curves 1-4

20 in Fig. 10 a) and $Im(Z_{11})$ corresponding to $\theta$ equal to $45°$ and $60°$ (curves 3, 4 in Fig. 10 b),  increase with increasing frequency in the considered frequency range $(0.86-1.14) \cdot 10^5$ c$^{-1}$. The average values of $Im(Z_{11})$ corresponding to $\theta$ equal to $5°$ and $30°$ , change in the frequency range $(0.86-1.14) \cdot 10^5$ c$^{-1}$ non-monotonically, having maximum values around frequency $(1-1.1) \cdot 10^5$ c$^{-1}$.   It is interesting to note that the value of finite impedance at the lower (Earth-atmosphere) boundary of the WGEI make a quite observable influence on the polarization transformation parameter minimum near the E region of the ionosphere (curves 1, 2  in the Fig. 10 c). Namely, the decrease of

25 surface impedance Z at the lower boundary (Earth-atmosphere) of the WGEI in two orders causes the increase of the corresponding minimum value of $|E_y/H_y|$ in ~ 100% (compare minima in the curves 1 and 2 around $z$=70 km).

**6. Discussion**

The observations presented in (Roznoi et al. 2015) demonstrated a possibility for seismogenic increasing losses of VLF waves in the WGEI (Fig. 8; see the details in (Roznoi et al. 2015)). We will discuss the correspondence to these experimental results qualitatively.

[Figure]

**Figure 11. Averaged through night residual VLF/LF signals in the ground observation for the wave paths: JJY-Moshiri, JJI-Kamchatka, JJY-Kamchatka, NWC-Kamchatka, and NPM-Kamchatka. Horizontal dotted lines show the 2σ level. The color filled zones highlight values exceeding the -2σ level. Two panels below are Dst variations and earthquakes magnitude values (from Rozhnoi et al., 2015, but not including the DEMETER data). See other details in (Rozhnoi et al. 2015).**

[Figure]

**Figure 12. Modification of the ionosphere by electric field of seismogenic origin. (a) – Geometry of the electrostatic problem on the penetration into the ionosphere of the electric field excited by near-ground seismogenic current source; I and II - isotropic and anisotropic regions of the system "atmosphere-ionosphere". (b) Electric field in the mesosphere; in the presence of the seismogenic current sources only in the mesosphere (curve 1); only in the lower atmosphere (curve 2); both in the mesosphere and in the lower atmosphere (curve 3); current sources in the mesosphere and lower atmosphere are of the same sign and coincide by the sign with fair weather current (directed vertically downward). ( c ) Relative perturbations (normalized on the corresponding steady-state**

values in the absence of perturbing electric field, denoted by the index "0") of electron temperature ($T_e / T_{e0}$), electron concentration ($N_e / N_{e0}$), and electron collision frequency ($\nu_e / \nu_{e0}$).

[Figure]

Figure 13. Altitude distributions of the normalized tangential (*y*) electric (a) and magnetic (b) VLF beam field components in the central plane of the transverse beam distribution (*y=0*) at the distance *z=1000* km from the input of the system. Curves 1 in Fig. a, b correspond qualitatively to the presence of only mesospheric electric current source (with relatively smaller value of $N_e$ and larger $\nu_e$) and curves 2 – to the presence of both mesospheric and near-ground seismogenic electric current sources (with relatively larger value of $N_e$ and smaller $\nu_e$); curves 1 and 2 correspond to the identical input beams at *z=0* (not shown here). Curves 1 and 2 in Fig. 10 a, b correspond qualitatively to the curves 1 and 3 in Fig. 9 b, respectively (see also Fig. 9 c and the caption to that figure). $\omega = 1.5 \cdot 10^5 c^{-1}$; $\theta = 45°$

To do this, account for the modification of the ionosphere due to electric field excited by the near-ground seismogenic current source. In the model (Rapoport et al. 2006), the presence of the mesospheric current source, which reflects the observations (Martynenko 2004; Meek 2001; Bragin 1974) is taken into account, and curve 1 in Fig.12 b the corresponding vertical field distribution in the mesosphere. It is supposed that the mesospheric current has only *Z* component and is positive, what means that it is directed vertically downward, as well as fair-weather current (curve 1, Fig. 11). Then suppose that near-ground seismogenic current is directed in the same way, as mesospheric current. If the mesospheric current is equal to zero and only corresponding seismogenic near-ground current is present. Corresponding mesospheric electric field, under the condition of given difference of the potentials between the Earth and the ionosphere (curve 2, Fig. 12 b), is directed oppositely to those excited by the corresponding mesospheric current (curve 1, Fig. 12 b). As a result, in the presence of both mesospheric and seismogenic near-ground current, the total mesospheric electric field (curve 3, Fig. 11 b) is less by the absolute value, than those in the presence of *only* mesospheric current (curve 1, Fig. 12 b). As it is shown in (Rapoport et al. 2006), the decrement of losses |*k''*| for VLF waves in the WGEI is proportional to $| k'' | \sim | \varepsilon'' | \sim N_e / \nu_e$. Accounting for that in the external electric field in the mesosphere $N_e$ and $\nu_e$ decreases and increases, respectively, due to the appearance of seismogenic near-ground electric current, in addition to the mesospheric current (curve 3, Fig. 12 b), losses increase comparatively to the case, when the seismogenic current is absent and electric field is larger by absolute value (curve 1, Fig. 12_new_9 b). An increasing in VLF beam losses, shown in Fig 13 corresponds to increasing losses with increasing absolute

value of imaginary part of the dielectric permittivity when near-ground seismogenic current source (curves 2 in Fig. 12 a, b) appear, additionally to the pre-existing mesospheric current source (curves 1 in Fig. 12 a, b, see also caption to the Fig. 12). This corresponds qualitatively to the results, presented in (Roznoi et al. 2015), see also Figs. 11_new_8. Note that the above mentioned estimations concern only volume losses in the WGEI. Losses connected with the modification of effective

5 impedance are not included in the course of these elementary estimations, and the more detailed consideration of both the modification of the ionosphere by means of electric-photochemistry mechanism and the variation of losses due to all mechanisms including volume and effective impedance effects will be a subject of the subsequent papers.

Note the following. The present paper is devoted to the new method of modeling characteristics of the WGEI, namely TIMEB and to the illustration of this method by the examples of the beam propagation in WGEI presented above.

10 Respectively the field shown in Figs. 4, 6, 7, 13_new include the range of altitudes inside the WGEI. Nevertheless the present method, in particular the application of the formulas (30), (24), (23), (27) and (15)-(19) allows to determine all the field components in the range of altitudes $0 \leq z \leq L_{max}$, where $L_{max} = 300$ km. We will present here only the final qualitative result of such simulations. Namely, it is shown that in the range $L_z \leq z \leq L_{max}$, where $L_z = 85$ km is the upper boundary of the effective WGEI, all the field components are (1) at least one order of altitude less than the corresponding

15 maximal value in the WGEI and (2) field components have the oscillating character (along $z$ coordinate) and describes the modes, leaking from the WGEI. The detail consideration of the electromagnetic waves leaking from the WGEI will be presented in the special paper.

Let us make a note also on the dependences of the field components in the WGEI on the vertical coordinate ($z$) and the change of such a dependence during a propagation of the VLF beam along the WGEI (Figs. 4, 6, 7, 13_new). The initial

20 distribution of the electromagnetic field on $z$ (Fig. 4a) is determined by the initial conditions on the beam, see relations (32). Such a field includes, naturally, higher eigenmodes of the WGEI. The structure and behavior of these eigenmodes in the WGEI will be a subject of a separate paper. Here we only note on this subject that the higher-order modes, in distinction to the lower ones, have quite large losses and practically disappear after beam propagation for a distance of order 1000 km in the WGEI. This circumstance determines the change in the altitude ($z$) and transverse ($y$) distributions of the field of the

25 beam during its propagation along the WGEI. In particular at the distance $x$=600 km from the beam input (Figs. 4 b, c) the few lowest modes of WGEI along $z$ and $y$ coordinate are still survived. At $x$=1000 km (Figs. 4 d, c, 6 e, f, 7 a, b), practically, only main mode in z direction is survived. Note that the field picture mentioned above concerns real WGEI with losses, and also gyrotropy and anisotropy cause both 
[revised manuscript text omitted]
|$, but does not reduce only to this. The corresponding change of this parameter has rather remarkable absolute values from dozens to thousands percent. Even more interesting is that in the case of decreasing electron concentration, the main maximum of |Ey/Hy | occurs in the lower atmosphere (at the altitude around 20 km. In the case of increasing electron concentration, the main maximum of $|E_y/H_y|$ occurs near the E region of the ionosphere (at the altitude around 77 km), while the secondary maximum practically disappears.

(iii) The real and imaginary parts of the surface impedance at the upper boundary of the WGEI have a quasiperiodical character with the amplitude of "oscillations" occurring around some effective average values decreases with increasing the angle $\theta$. Corresponding average values of $Re(Z_{11})$ and $Im(Z_{11})$, in general, decrease with increasing angle $\theta$. Average values of $Re(Z_{11})$ for $\theta$ equal to $5°, 30°, 45°$ and $60°$ and $Im(Z_{11})$ corresponding to $\theta$ equal to $45°$ and $60°$, increase with increasing frequency in the considered frequency range $(0.86\text{-}1.14) \cdot 10^5$ c$^{-1}$. The average values of $Im(Z_{11})$ corresponding to $\theta$ equal to $5°$ and $30°$, change in the frequency range $(0.86\text{-}1.14) \cdot 10^5$ c$^{-1}$ non-monotonically, having maximum values around frequency $(1\text{-}1.1) \cdot 10^5$ c$^{-1}$.

(iv) The value of finite impedance at the lower (Earth-atmosphere) boundary of the WGEI make a quite observable influence on the polarization transformation parameter minimum near the E region of the ionosphere. Namely, the decrease of surface impedance Z at the lower boundary (Earth-atmosphere) of the WGEI in two orders causes the increase of the corresponding minimum value of $|E_y/H_y|$ in ~ 100%.

(8) In the range $L_z \leq z \leq L_{max}$, where $L_z = 85$ km is the upper boundary of the effective WGEI, all the field components are (1) at least one order of altitude less than the corresponding maximal value in the WGEI and (2) field components have the oscillating character (along $z$ coordinate) and describes the modes, leaking from the WGEI. The detail consideration of the electromagnetic waves leaking from the WGEI will be presented in the special paper. The initial distribution of the electromagnetic field on $z$ (vertical direction) is determined by the initial conditions on the beam. Such a field includes higher eigenmodes of the WGEI. The higher-order modes, in distinction to the lower ones, have quite large losses and practically disappear after a beam propagation for a distance of order 1000 km in the WGEI. This circumstance determines the change in the altitude ($z$) distribution of the field of the beam during its propagation along the WGEI. In particular at the distance $x$=600 km from the beam

input, the few lowest modes of WGEI along z coordinate are still survived. At $x=1000$ km, practically, only main mode in z direction is survived. 
[revised manuscript text omitted]

---

## Author Response (AR1)

- 5 We are very grateful to the Anonymous Referee #2 and are answering directly to each of the main his comments and will answer to each of the minor comments in the final text of the revised paper as well. We believe that the paper becomes really better and more clear, informative and useful for the readers due to the improvements inspired by the Referee.
- The section "5. Influence of the parameters of WGEI on the polarization transformation and losses of the propagating VLF waves" (P21-24 L485-581) (added especially in response to comments (2), (3) of the Reviewer) and the Figures 8-10 are new, and the corresponding Conclusions are added. In our answers to the Referee's comment, we refer to the improvements in the paper, noting corresponding pages and lines such as f.e. "P29 L696-703", respectively, what means: page 31, lines 696-703. The questions of the Referee are revealed below by "black bold" font.

**(1) The interesting goal in a model is to determine what are the main parameters for the increase/decrease of the EM field. Here in this paper we only have a variation of one parameter: the electron density.**

Concerning the perturbations in the electron concentration: we include in the revised paper, besides increase or decrease in electron concentration, Fig. 5, P19 L460-463, which has been present already in the

- 20 previous version of the paper, also parametrization of the perturbation in electron concentration. This parameterization is made of the basis of such parameters, as maximum of the perturbation, region, where it is concentrated and effective width of this distribution by the vertical coordinate, see P23 L.533-540 and formula (35), P22 L536. Then in response to this note of the Referee 2, we added also an influence on the electrodynamics characteristics of the VLF beams propagating in the waveguide Earth-Ion ionosphere (WGEI) of
- the angle of the inclination of geomagnetic field and carrier frequency on the the beam (P22-23 L510-531; P 23-24 L538-580). Then besides just field spatial distributions, as an object of the influence of the parameters, we include two other very interesting and important parameters. One of them is a parameter of the polarization transformation  $|E_y/H_y|$ , described in new Sect. 5, P21 L489-506. This value can be measured and characterizes the effect of the gyrotropy and anisotropy, both volume and surface, described by the tensor surface
- 30 impedances at the lower and upper boundaries of the waveguide Earth-Ionosphere (WGEI). The other value, the influence on which we analyze, is the complex tensor impedance, in particular real and imaginary parts of its diagonal element Z011. In particular, an influence on it of the inclination angle of the geomagnetic field and carrier beam frequency are included into the revised paper. Then we analyze also losses of the VLF beam (P21 L506-509). The details concerning these extra simulations with the influences on the propagation of a beam in
- 35 the WGEI of different parameters are presented in the Fig. 8(P22 L512-522), Fig. 9(P23 L541-548), Fig. 10(P24

**L563-569)** and the texts with explanations included after these Figures in the (new) Sect. 5, revealed in the revised paper by red color.

Note that the Figures 11-13 (P 25-26) in the paper after the revision are the same, as Figures 8-10 in the paper before the revision (while the Figures 8-11 (P22-24) in the revised paper are new, as it was already

- 5 mentioned). The Figures 11-13 describe experimentally and (qualitatively) theoretically a common effect of seismogenic changing electron concentration and collision frequency. The last changes are described in our previous model (Rapoport et al. 2006) of the influence on the ionosphere of the seismogenic electrostatic field including the corresponding heating-photochemistry effects. This is rather complicated mechanism which includes a set of parameters in the ionosphere (the number of which is >> 1) such as characteristics of external
- 10 electrostatic sources placed in the lower atmosphere and mesosphere, photochemistry parameters, electron concentration distribution, photochemistry parameters etc., effect of which is reflected finally in Figs. 12 a- c (P25-26 L587-598). In the modeling beam propagation in the WGEI, we used only qualitatively the fact, that as a result of mentioned above seismogenic mechanism, electron concentration  $N_e$  and collision frequency  $v_e$ change simultaneously ( $N_e$  and  $v_e$  increase and decrease, respectively). As a result, the curve 2, comparatively to
- 15 the curve 1 in Fig. 13 a,b (P26, L599-606), reflects an effect of additional losses of VLF beam, which occurs due to simultaneous change of not *one*, but *two* parameters ( $N_e$  and  $v_e$ ), and, moreover, these parameters change their signs consistently. See also the text in Sect. 6, P25 L582-584; P26-27 608-626.

20

35

**(2) Here in this paper we only have a variation of one parameter: the electron density. It means that you show something which is evident: when the density increases the electric field decreases.**

As it is already mentioned in item (1), a variations of other parameters are included as well, such as geomagnetic field inclination angle, beam carrier frequency, value of surface impedance/conductivity of the lower boundary of the WGEI (atmosphere-Earth boundary). Corresponding results are described in more details below (see item (3)). But even effects from electron variations are *not* evident! Namely:

- (i) The simultaneous effect from change of electron concentration Ne and collision frequency ve is non-trivial. First, as described above, this effect is caused by the consistent change of two different parameters, and consistent change in their signs is described by rather complicated model of seismogenic electrostatic-heating-photochemistry effects, described in particular in (Rapoport et al. 2006; Grimalsky et al. 2003). The consistent change of signs of both Ne and ve causes the proper change in VLF losses/amplitude of the electromagnetic field (curve 2 in Figs. 13 a, b, P26 L599-606), which qualitatively corresponds to the effect observed before strong earthquake (Fig. 11, P25 L585-590). These results are summarized in the Conclusion (3), P29 L688-690.
  - (ii) As it is seen from Fig. 9 (P23 L541-548), the change in the concentration in the lower ionosphere causes rather nontrivial effect on the parameter of polarization transformation |Ey/Hy|. Note that either increase or decrease in the ionosphere plasma concentration have been reported as a result of seismogenic phenomena, tsunamis, particle precipitation in the ionosphere due to wave-particle

interaction in the radiation belts (Pulinets et al. 2005; Shinagawa et al. 2013; Arnoldy et al. 1989; Glukhov et al. 1992; Tolstoy et al. 1986) etc. Namely, this effect does not reduces only to increase (Fig. 9 b) or decrease (Fig. 9 c) of the maximum value of the polarization transformation parameter |Ey/Hy|. Note also that the corresponding change of this parameter has rather remarkable absolute values from dozens to thousands percent, as it is seen from the Figs. 9 b, c, Fig. 8 c. It is even more interesting that in the case of decreasing (Fig. 9 a, curve 2) electron concentration, the main maximum of |Ey/Hy| appears in the lower atmosphere (at the altitude around 20 km, Fig. 9 b, curve 3, which corresponds to  $\omega = 1.14 \cdot 10^5 s^{-1}$ ). In the case of increasing electron concentration (Fig. 9a, curve 3) the main maximum of |Ey/Hy| appears near the E region of the ionosphere (at the altitude around 77 km, Fig. 9c). The secondary maximum which is placed, in the absence of the perturbation of the electron concentration, in the lower atmosphere (Fig. 8 c, curves 2, 3, P22 L512-522), or mesosphere/ionospheric D region ((Fig. 8\_c, curve 1), practically disappears or just is not

seen in the present scale, if electron concentration increases (Fig. 9 c, curves 1-3, P23, L541-548).

The new effects mentioned in item (ii) above is described in the revised paper in P23-24 L549-562. These results are summarized in the Conclusion (6), item (ii), P29 L708-714.

5

10

(3) What is the effect of other parameters as the magnetic field inclination for example? The plasma frequency?

As it was already mentioned above in the paragraph (1), in response to this comment, we have demonstrated an influence of the change in the angle of the inclination of geomagnetic field and carrier frequency on the

- 20 electromagnetic characteristics of propagating EM VLF beam. Then besides just field spatial distributions, as an object of the influence of the parameters, we include two other very interesting and important parameters. One of them is polarization transformation. This value can be measured and characterizes the effect of the gyrotropy and anisotropy, both volume and surface, described by the tensor surface impedances at the lower and upper boundaries of the waveguide Earth-Ionosphere (WGEI). The other value, an influence on which we
- 25 analyze, is the complex tensor impedance, in particular real and imaginary parts of its diagonal element Z011. In particular an influence on it of the inclination angle of the geomagnetic field and carrier beam frequency are included into the revised paper. The influence on the losses of the VLF beam carrier frequency is analyzed as well. The following results are obtained, in particular, (and reflected in the new Figs. 8 -10, P22-24) basing on this new modeling, performed in the response to the present Reviewer's note.
- 30 (i) As it is seen from Figs. 8 a-c (P22 L512-522), the altitude dependence of the polarization parameter  $|E_y/H_y|$  has two main maxima in the WGEI, the higher of which lies in the gyrotropic region above 70 km, while the other in the isotropic region of the WGEI. As it is seen from Fig. 8 a, b, the value of the larger second maximum increases, while the value of the second maximum decreases and its position shifts to the lower altitudes with increasing frequency. At the higher frequency ( $\omega = 1.14 \cdot 10^5 c^{-1}$ ), the larger maximum of the polarization parameter corresponds to the intermediate value of the angle  $\theta = 45^{\circ}$  (Fig. 8b); for the lower frequency ( $\omega = 0.86. \cdot 10^5 c^{-1}$ ), the largest value of the first (higher) maximum corresponds to the almost vertical direction of the
  - 3

geomagnetic field ( $\theta = 5^{\circ}$ , Fig. 8a). For the intermediate value of the angle ( $\theta = 45^{\circ}$ ), the largest value of the main maximum corresponds to the higher frequency ( $\omega = 1.14 \cdot 10^5 c^{-1}$ ) in the considered frequency range (Fig. 8c). The total losses increase monotonically with increasing frequency and depend weakly on the value of  $\theta$  (Fig. 8d).

These results are included in the paper after Figure 8 (P22 L512-522) in (P22 L523-531) and summarized in Conclusion (6), item (i), P29 L700-707.

5

(ii) Again, as it was already mentioned, as it is seen from Fig. 9 (P23 L541-548), the change in the concentration (which determines plasma frequency) in the lower ionosphere causes rather nontrivial effect on the polarization transformation parameter |Ey/Hy |. Note that either increase 10 or decrease in the ionosphere plasma concentration have been reported as a result of seismogenic phenomena, tsunamis, particle precipitation in the ionosphere due to wave-particle interaction in the radiation belts (Pulinets et al. 2005; Shinagawa et al. 2013; Arnoldy et al. 1989; Glukhov et al. 1992; Tolstoy et al. 1986) etc. Namely, this effect does not reduces only to increase (Fig. 9 b) or decrease (Fig. 9 c) of the maximum value of the polarization transformation parameter [Ey/Hy]. 15 Note also that the corresponding change of this parameter has rather remarkable absolute values from dozens to thousands percent, as it is seen from the comparison between Figs. 9 b, c (P23 L541-548) and Fig. 8 c, curve 3, P22, L512-522. The last curve corresponds to the unperturbed distribution of the ionospheric electron concentration (see curve 1, Fig. 5b, P19 L460-463, and curve 1 in Fig. 9). It is even more interesting that in the case of decreasing (Fig. 9 a, curve 2) electron concentration, the main maximum of |Ey/Hy | appears in the lower atmosphere (at the altitude 20 around 20 km, Fig. 9 b, curve 3, which corresponds to  $\omega = 1.14 \cdot 10^5 c^{-1}$ ). In the case of increasing electron concentration (Fig. 9 a, curve 3) the main maximum of |Ey/Hy | appears near the E region of the ionosphere (at the altitude around 77 km), while the secondary maximum placed, in the absence of the perturbation of the electron concentration, in the lower atmosphere (Fig. 8, curves 25 2, 3, P22, L512-522), or mesosphere/ionospheric D region ((Fig. 8\_new c, curve 1), practically disappears or just is not seen in the present scale, in the case under consideration (Fig. 9 c, curves 1-3, P23, L541-548).

These results are presented in the revised paper after Fig. 9, in P23-24 L549-562, and summarized in Conclusion (6), item (ii), P29-30 L708-725.

30 (iii) As it is seen from Fig. 10 (P24 L563-569), the real (a) and imaginary (b) parts of the surface impedance at the upper boundary of the WGEI have a quasiperiodical character with the amplitude of "oscillations" occurring around some effective average values (not shown explicitly in Figs. 10 a, b) decreases with increasing the angle θ. Even without the determination of the exact average values for each of the curves 1-4 in Figs. 10 a, b, it is seen that corresponding average values of Re(Z011) and Im(Z011), in general, decrease with increasing angle θ. It is also seen that average values of Re(Z011) for θ equal to 5°, 30°, 45° and 60° (curves 1-4 in Fig. 10 a) and Im(Z011) corresponding to θ equal to 45° and 60° (curves 3, 4 in Fig. 10 b), increase with increasing frequency in the

considered frequency range  $(0.86-1.14)\cdot 10^5 \text{ s}^{-1}$ . The average values of  $\text{Im}(Z_{011})$  corresponding to  $\theta$  equal to 5° and 30°, change in the frequency range  $(0.86-1.14)\cdot 10^5 \text{ c}^{-1}$  non-monotonically, having maximum values around frequency  $(1-1.1)\cdot 10^5 \text{ s}^{-1}$ .

**These results are included in the revised paper after Figure 10 (P24 L563-569), in (L24 L571-580) and summarized in Conclusion (6), item (iii), P29-30 L713-723.**

(iv) It is interesting to note that the value of finite impedance at the lower (Earth-atmosphere) boundary of the WGEI make a quite observable influence on the minimum value of the polarization transformation parameter near the E region of the ionosphere (curves 1, 2 in the Fig. 10 c, P24 L562-568). Namely, the decrease of surface impedance Z at the lower boundary (Earth-atmosphere) of the WGEI in two orders causes the increase of the corresponding minimum value of |Ey/Hy| in ~ 100% (compare minima in the curves 1 and 2 around z=70 km).

**These results are presented in the end of the text after the caption to Figure 10, P34, L22-26, and summarized in Conclusion (7), item (iv), P30 L722-725.**

Concerning the plasma frequency ( $\omega_{pe,i}$ ) it is proportional to the square root of the electron concentration  $N_e$

- and included into complex conductivity and complex tensor  $\hat{\varepsilon}$ . These values determine also effective tensor of surface impedance. Therefore all of the electrodynamics characteristics both presented in the initial version of the paper and added and reflected in the new Figs. 8-10, P22-24, made in the response to the Reviewer's notes, are influenced by these volume tensor  $\hat{\varepsilon}$  and surface impedance  $Z_{0ij}(i, j = 1, 2)$  and therefore by the altitude distributions of electron concentration n. For example, change in the altitude distribution n (Fig. 5b, P19
- 20 L460-463) causes change in  $\hat{\varepsilon}$  (Figs. 5 c, d) and surface tensor impedance  $\hat{Z}_0$  (Table 1, P19-20 L470-471). As a result, spatial distributions of the electromagnetic field change (Fig. 6, P20 L472-475; Fig. 7, P20 L476-479). Due to changes in electron concentration (and therefore  $\omega_{pe,i}$ ), transformation polarization parameter |Ey/Hy| and its altitude distribution change very non-trivially (Fig. 9, P23 L541-548). There are only couple of examples, but in any of the other results obtained in the paper, an influence of *n* and, respectively,  $\omega_n$  are reflected. We do
- not emphasize separately  $\omega_{pe,i}$  because they are determined unambiguously by n, the influence of which on the electrodynamics characteristics of the VLF beam is investigated rather in details, and the masses of the corresponding particles, while the latter are assumed to be given within the framework of the approximations adopted in this article.

**(4) Why the calculation is stopped at 80 km?**

30 The proposed new tensor impedance method for modeling propagation of electromagnetic beams (TIMEB) and the developed model allows and we really did the simulations of (all) the electromagnetic field components both inside the WGEI ( $0 < z < L_z$ ,  $L_z$ =85 km) and above the WGEI ( $L_z < z < L_{max, z}$ ,  $L_{max}$ =300 km). Nevertheless the

5

detailed calculations of the field distribution outside the WGEI will be not presented in this paper and will be published elsewhere. This paper is devoted, besides the new method in general, also to the propagation of the beam *inside* the WGEI, i.e. in the range of altitudes .  $0 < z < L_z$ ,  $L_z$ =85 km. The calculations of the field above the WGEI are performed in this context only to establish /confirm the present approximation of the propagation of

- 5 the beam *inside* the WGEI. In other words the results of the field calculations above the WGEI, in the region  $L_z < z < L_{max,,} L_{max} = 300$  km confirms that the region  $0 < z < L_z$ ,  $L_z = 85$  km is really a good waveguide for the VLF field accounting for the effects of the gyrotropy and anisotropy for the plasma-like media placed in the inclined geomagnetic field. This fact is really confirmed by the calculations of (all) field components performed for the range of the altitudes from 0 to 300 km. Nevertheless because, as it was already mentioned above, this paper is
- 10 devoted to the beam propagation in the WGEI, only final qualitative conclusion based on the calculation above the WGEI is presented in the paper. Namely, in the response to the present Reviewer's question, the following text is added into the paper (P27 L627-634):

"The TIMEB is a new method of modeling characteristics of the WGEI. The results of beam propagation in WGEI modeling presented above include the range of altitudes inside the WGEI (see Figs. 4-7). Nevertheless, the TIMEB method

15 described by Eqs. (15)-(19), (22-24), (27), (30) and allows to determine all field components in the range of altitudes  $0 \le z \le L_{\max}$ , where  $L_{\max} = 300$  km. The structure and behavior of these eigenmodes in the WGEI and leakage waves will be a subject of separate papers. We present here only the final qualitative result of the simulations. In the range  $L_z \le z \le L_{\max}$ , where  $L_z = 85$  km is the upper boundary of the effective WGEI, all field components are (1) at least one order of altitude less than the corresponding maximal field value in the WGEI and (2) field components have the oscillating character along 20 z coordinate and describe the modes, leaking from the WGEI."

**These results are summarized in the paragraph (7) of the Conclusions, P30 L726-728.**

To the details concerning the waves leaking from the WGEI and comparison of the theoretical simulations to the results of the corresponding observations, the special and separate paper will be devoted. And all the calculations for the field in the altitude range above the WGEI will be suitable for a direct inclusion only in this paper.

**(5) In Figure 4 why Ey is oscillating along Z?**

25

**The following note is made in the revised paper in the response to this question of the Reviewer, see . P27 L635-643:**

"Let us make a note also on the dependences of the field components in the WGEI on the vertical coordinate (z). 30 The initial distribution of the electromagnetic field with altitude z (Fig. 4a) is determined by the boundary conditions of the beam (see Eq. (32)). The field component includes higher eigenmodes of the WGEI. The higher-order modes experienced quite large losses and practically disappear after beam propagation on 1000 km distance. This determines the change in altitude (z) and transverse (y) distributions of the beam field during propagation along the WGEI. In particular, at the distance x=600 km from the beam input (Figs. 4b, c), the few lowest modes of the WGEI along z and y coordinates still persist. At distance x=1000 km (Fig. 4d, c; Fig. 6e, f; and Fig. 7a, b), only the main mode persists in the z direction. Note, the described field structure correspond to real WGEI with losses. The gyrotropy and anisotropy causes the volume effects and

5 surface impedance, in distinction to the ideal planar metallized waveguide with isotropic filling (Collin, 2001). "

**Minor points:**

15

**The English is not fluent and there are many mistakes (or typos) which can be easily corrected with a word processor**

10 The English language is polished with many relevant grammatical and stylistic improvements that did not stand out in the text of the reviewed article due to their significant number. The collective of authors in particular those who are working in USA and UK and working abroad for a long time did this.

**Page 3 line 21 To Be Corrected**

This line is written in the revised paper (P3 L75-76) in the form: "Some other details on the distinctions from the previously published models are given below in Sect. 3."

Just in case I include below also the lines neighboring to the corresponding line in the revised paper (P3 L73-81):

"This combined approach, finally, creates the possibility to interpret adequately a data of both groundbased and satellite detection on the EM wave/beam propagating in the WGEI and those, which experienced a leakage from the WGEI into the upper ionosphere and magnetosphere. Some other details on the distinctions

20 from the previously published models are given below in Sect. 3.

The methods of effective boundary conditions such as effective impedance conditions (Tretyakov, 2003; Senior and Volakis, 1995; Kurushin and Nefedov, 1983) are well-known and can be used, in particular, for the layered metal-dielectric, metamaterial and gyrotropic active layered and waveguiding media of different types (Tretyakov, 2003; Senior and Volakis, 1995; Kurushin and Nefedov, 1983; Collin, 2001; Wait, 1996) including

25 plasma-like solid state (Ruibys, and Tolutis, 1983) and space plasma (Wait, 1996) media.. "

As we see now, this peace of this text (incl. line (P3 L75-76)) in the revised paper is correct.

Page 3 line 25 Wait – done, and the name "Wait" is written right and revealed using the red color font everywhere in the revised paper, see P3 L80, 81, 102; P8 L196; P35 L911.

(I have not checked the references but I have seen that Ruibie & Tolutue is not correct) – this reference is improved, as "Ruibys, and Tolutis" everywhere in the revised paper, see P3 L80, 82, 102; P35 L878.

Page 4 line 10 – waves: changed to "electromagnetic waves", P3 L101.

5

10

**Page 4 line 11 LAIM appear before and then must be explained before –** "LAIM" is explained now in the second line of the Introduction, see P1 L33.

**Legend of Figure 1 is too long. A part must be in the text (it is also true for other figures).**

As it was recommended, the caption to Figure 1 is reduced. Namely the (modified for the revised paper) phrases: "In Fig. 1,  $\theta$  is the angle between the directions of the vertical axis z and geomagnetic field  $\vec{H}_0$ . Note that theta  $\theta$  angle is complementary to the angle of inclination of the geomagnetic field. Geomagnetic field  $\vec{H}_0$  is directed along z'axis, lies in the plane xz, while the planes x'z' and xz coincide with each other."

are transferred from the end of the caption to Figure 1 to the text of the article before Figure 1, see P4-5 L135-138. The captions to the remaining figures includes only the data necessary to identify these figures, and to distinguish among themselves the different curves in each figure and are necessary. The information included in the captions is fundamentally necessary to provide readers with the opportunity to quickly find out what the

15 meaning of each of the Figures separately is, as well as all the Figures in general. As for the description of the figures in the text of the article, it is included for each figure separately and when comparing the physical effect between themselves, illustrated by various figures or groups of figures. At the same time, a description is also given for the corresponding figures in the necessary proportion while the basic physical effects, illustrated by the corresponding figures are described. These physical effects are mainly then included in the Conclusions.

**20 Of course I have not checked the correctness of all equations but I have seen an error in the first equation (equation (1)) for the ion plasma frequency.**

The mass of electrons  $m_e$  (present in the formula (1) (P6, L167) for the square of plasma frequency) is replaced by the mass of ions  $m_i$ . Now the formulas (1) for electron and ion plasma concentrations in the accepted approximation of the three-component plasma-like ionosphere (including electron, effective one type of ion

25 and effective one type of neutral components) and quasineutrality, are right. The phrase describing this approximation is included into the text of the paper, Subsection 3.1.1, see P6 L170-175.

**Page 6 line 5 and line 17 the sign inside exp() is different.**

Yes, this is right- there are two parts of the argument of the phase multiplier  $\sim \exp(i\omega t - ik_x x)$ , which really have different signs, see P5 L162, P6 L179 in revised paper.

8

**30 Page 7 the values of BETAij are not clear. What parameters they contain ?**

Please look at the line placed 3 lines above the upper formula from Eqs. (1), namely (P5 L163). It is written there:  $\hat{\beta} = \hat{\varepsilon}^{-1}$ , or  $\vec{E} = \hat{\beta}\vec{D}$  (the arguments of  $\hat{\varepsilon}$ ,  $\hat{\beta}$  are omitted here, but included in the paper. Therefore the tensor  $\hat{\beta}$ , inverse respectively to  $\hat{\varepsilon}$  and depends on the same parameters as  $\hat{\varepsilon}$ , which is described in Sect. 3.1, in particular using formulas (1). Therefore  $\hat{\beta}$  depends on the same parameters as the tensor  $\hat{\varepsilon}'$ , described by

- 5 formulas (1), with components, the altitude distributions of which is illustrated in Figs. 3 (P17-18 L435-439) and 5c, d (P19 L460-463), and by the angle θ and corresponding standard rotation matrices, mentioned in (P6 L173), the fifth line after Eq. (1) (and not included explicitly). Please note that due to chosen subject- layered anisotropy and gyrotropy inhomogeneous plasma-like Earth-Atmosphere-Ionosphere media - and respectively to do "what is necessary and how it is necessary", we are forced to choice combined analytical-numerical
- approach. Namely, all what is possible, we are doing analytically and all other-numerically. In particular the tensor  $\hat{\beta}$  is obtained, using proper formulas for reversal tensors (they are standard and not included in the paper), from the tensor  $\hat{\varepsilon}$ .

Page 9 line 10 respectively two time – yes, this phrase is improved now as follows, see P9 230-231:

" $k_x$  and  $k_z$  are the transverse and longitudinal components of wave number relative to geomagnetic field."

- 15 Page 9 line 15 relation The matrix at the end of equation (14) seems strange. The left lower element is not 1-? – thank you very much, the typos in this formula is improved, see P9 L238 (eq. (14)). Namely, the upper right element is (-1-i). Just in case, this matrix has been obtained analytically by means of few different approaches independently, with the same result.
- Page 13 another parameter DELTA appears here. Is the DELTA in equation (11) similar to the DELTA in equation (24) ? thank you, there are different values. To distinguish between them, the corresponding value in equation (24) is re-denoted now as  $\Delta_0$ , see P13 L336.

**Title 3.5 too long.-** Yes, the title of the Subsect. 3.5 is shortened as follows (P16 L397): **"3.5 VLF Waveguide Modes and Reflection from the VLF Waveguide Upper Effective Boundary ."**

**25**

**Page 16 a lot of typos, discharges, demonstrating, speaking, present, presentation. – improved, namely:**

The first two phrases in the beginning of Chapter 3.5, namely

"Our model, in general, needs the consideration of the excitations of the waveguide modes by means of current sources such as dipole-like VLF radio source and lightning discahrges. Then, we will present the results of the

9

30 reflection of the waves incident on the upper boundary (*z=Lz*) of the effective WGEI demonstrating that this structure has indeed good enough waveguiding properties."

Are replaced by (P16 L398-401):

"In general, our model needs the consideration of the waveguide modes excitations by a current source such as dipole-like VLF radio source and lightning discharge. Then, the reflection of the waves incident on the upper boundary  $(z=L_z)$  of the effective WGEI can be considered. There will be possible to demonstrate that this structure has indeed good enough

5 waveguiding properties"

As a result of this replacement:

"discharges" – replaced by "discharge" (P16, L399)

"demonsrtrating" – removed and the word "demonstrate" (P16 L400) is used in this piece of text in the revised paper.

Then the following improvement are done in this text, in the first paragraph of Sect. 3.5, p. 16: 10

Shortly speaking - removed Present; presentation – are improved (P16 L401,403). Page 17 line 1 these - corresponding piece of text (P16, 17 L409-413) is rephrased and the word ("these") is included in P17 L413.

Page 17 line 9 why Figures 3\_2 and 4\_3 – (the same concerns line 8) – improved as Fig. 3, Fig. 4 – see in the 15 revised paper P17 L420, 421.

In Figure 3 it is difficult to understand the contain of the panels b) to g) - Page 19 line 13 figure 5 – Improved. Namely, quality of all panels in Figs. 3 and Fig. 5 is improved (widths of the lines increased, sizes of the letters and numbers in the Figures increased). Therefore the contain of all panels in Figs. 3, 5 should become

understandable now, see P17-18, L435-436; P19 L460. 20

Besides of that, after extra cross-checking the set of improvements have been done which only clarified the content of the paper and has not changed its physical sense. In particular the following extra improvements have been included in the revised paper.

Ssign in square root in the first of formula (9) is improved – there was before mistakenly, line 17, p. 8:

$$\kappa_{1,2}^{2} = \frac{\beta_{11} + \beta_{22}}{2} \pm \left( \left( \frac{\beta_{11} + \beta_{22}}{2} \right)^{2} + \beta_{12} \beta_{21} \right)^{2}$$

There became after the improvement in the revised paper, P8 L214:

$$\kappa_{1,2}^{2} = \frac{\beta_{11} + \beta_{22}}{2} \pm \left( \left( \frac{\beta_{11} - \beta_{22}}{2} \right)^{2} + \beta_{12} \beta_{21} \right)^{1/2}$$

Note that besides improving Figs. 3 (P17-18 L435-439), 5 (P19 L460-463), In accordance with Reviewer's requirements, the new Figs. 8, 9, 10 (P22 L512-522; P23 L541-548; P24 L563-569) are prepared, and then Figs. 11-14 from the revised paper got now the new numbers (11-14 mentioned above). The numbers of these figs. in the previous version of the paper were 8-10.

5

In Eq. 27 (P14 L345) the value  $\Delta$  is re-denoted as  $\Delta_y$  (in P14 L346, placed after Eq. (27)) to distinguish it from the similar value  $\Delta$  included in Eq. 30b (P14 L358) or, more accurately speaking in the Eqs. for  $\gamma_{12},...,\tilde{\beta}_{11}...\Delta$  written in (P14 L360) *after* Eq. (30b).

The value included into the first of Eqs. (30b) (P14 L358) has not been determined in the previous version of 10 the paper. In the revised paper, the value  $\tilde{\varepsilon}_{22}$  is included into the set of Eqs. in the (P14 L360) placed after formulas (30b).

The impedance Z and corresponding impedance elements  $Z_{ij}$  (I,j=1,2) are re-denouted as  $Z_0$  and  $Z_{0ij}$ , respectively everywhere in the paper, to distinguish impedance from the vertical coordinate Z. This concerns, f.e. formulas 22 (P11-12 L 289-292) for the elements  $Z_{ij}$  (I,j=1,2) of the impedance tensor; caption to Fig. 10

15 (P24 L568, 569); in the Table 1 (P19-20 L470-471) with the values of tensor impedance components etc.; extra column is added in the Table 1 with the proper titles of the corresponding lines of this table.

The Reference to Yaxin et al., 2012 in previous version of the paper is rewritten as (Yu et al. 2012) in (P2 L66) in the revised paper; the reference to (Artemyev et al., 2015), missed previously, is added in the revised paper in (P2 L59); the reference to (Yigit et al., 2016), missed in the previous version of the paper is added in the revised paper in (P2 L38).

11

20 Abstact and Conclusion 1 are improved stylistically.

---

## Author Response (AR2)

Dear Editor and Reviewer!

Thank you very much for the comments, which lead to the improvement of the quality of the paper, again!

The following is improved in accordance with your comments revealing the new improvements in the text by blue and in the formulas - by red colors; the revealing the improvements done on the previous stages of the paper's revision using the red color is saved in the present version of the paper as well.

**Editor's comment:**

**(1) There appeared some unclearness in the text. Most probably, some of them are due to conversion to pdf. format, e.g. "Ltym hf,jnf. C jndtnjv htwtyptynfv b ghb `njv enjxyz. B ntrcn" (lines 68-69)**
Now it is written in **P2 L68-69**:
This allows to determine the leakage modes and to interpret not only ground-based, but also satellite measurements of the VLF beam characteristics.

**Referee's comments:**

**(2) Line 73 remove a in "adequately a data"**
Now it is written in **P3 L74**:
…adequately data…

**(3) Line 430 Zc11 does not exist**
Now it is written in **P17 L436**:
…$Z_{011} =$…

**(4) There is a problem with the legend of Figure 3. We do not know where are the real and the imaginary parts of the components.**
Now it is written in **P18 L444-445**:
**(b) – (g) The real (corresponding lines with the values denoted by one prime) and imaginary (corresponding lines with the values denoted by two primes) parts of the components** $\varepsilon_1$, $\varepsilon_3$, $\varepsilon_h$, **general and detailed views.**

**(5) Line 443 –5 to be exponent**
Now it is written in **P18 L453**:
…$3 \cdot 10^{-5}$ Gs…

**(6) In Figure 6, 6b looks like 6f please check**
Yes, thank you! Fig. 6b is now replaced by the proper one, **P20 L482**

**(7) Line 531 please check because it seems not related to fig 8d**
No, this line (now: **P22 L544-545**) does relate to Fig. 8d; curves 1-4 at this Figure correspond to the different values of $\theta$ (see **P22 L528, 533-534**).

**(8) Line 536 meaning of f here ? f is already used as the frequency !!**

Thank you!  Yes, now the non-dimensional coefficient " $f$ " is replaced by " $f_{00}$ " in formula (35) **(P23 L549)**; relations with " $f_{00}$ " included in **(P23 L557; P23 L558).**

**(9) However I recommend to the authors to carefully read again their paper. I have seen some problems but I am not sure that I have seen all the small points concerning for example the equations and the figures**

Yes, we have rechecked the text of the paper, figures, formulas. As a result, the following additional improvements have been done.

- The reference concerning the properties of E-region is added: in the text - (Guglielmi and Pokhotelov, 1996), **P13 L332**; in the list of references- in **P33 L 853**

- The determination of $\beta_{33}$ is rewritten more accurately, namely the previous formula $\beta_{33} = \varepsilon(z=0)^{-1}$ is replaced now by $\beta_{33} = \varepsilon^{-1}(z=0)$ in **P 15 L367**; in the same line, in the formulas for the components of the impedance the index "0" missed previously, namely $Z_{012} = Z_{021} = 0$

- The misprinting in second formula from (31b) is improved, namely the previous formula $\dfrac{i}{\varepsilon(z=0)k_0}\dfrac{\partial H_y}{\partial z} + \dfrac{i}{k_0}Z_{0E}H_y = 0$ (containing the second term with a wrong dimensionality) is replaced now by $\dfrac{i}{\varepsilon(z=0)k_0}\dfrac{\partial H_y}{\partial z} + Z_{0E}H_y = 0$ , see **P 15 L 368**.

- The phrase "…for the magnetic field maximum value (Fig. e) is $3\cdot10^{-5}$ Gs $\approx 3$ nT$\approx 10$ mV/cm." is replaced by "for the magnetic field maximum value (Fig. e) is $3\cdot10^{-5}$ Gs $\approx 3$ nT.", see **P18 L453**.

- Axes labels in Fig. 4 and Fig. 6 are corrected to be clearly visible

- The references to the books, from which ionospheric data has been taken, are added in in the text: The parameters of the ionosphere used for modeling are taken from (Al'pert 1972; Alperovich and Fedorov 2007; Kelley 2009; Schunk and Nagy, 2010; Jursa 1985) -

In **P 17 L424-425** in the text of the paper and in (**P32 L798; P31 L796-797; P34 L869; P36 L943; P34 L868**) – in the list of references.

- The acknowledgements concerning OA, JB, VF are improved/detalized – in **P25 L784-788**

- The caption of the Figure 11 is improved with adding the proper license of the paper Rozhnoi et al. (2015) mentioned in this caption, see **P25** L 603-604

- Fig. 12b is rebuilt, to illustrate more clearly the main information presented in this Figure, see **P25 L605** .

- The Sections AUTHOR CONTRIBUTION, COMPETING INTERESTS, DATA AVAILABILITY are added, see **P30-31 L759-781**

- The set of other minor clarifications: improvements of misprintings and small stylistic and grammar improvements, not described here in details, have been added as well.